# Better by Default: Strong Pre-Tuned MLPs and Boosted Trees on Tabular Data

**David Holzmüller**[*]
SIERRA Team, Inria Paris
Ecole Normale Superieure
PSL University

**Léo Grinsztajn**
SODA Team, Inria Saclay

**Ingo Steinwart**
University of Stuttgart
Faculty of Mathematics and Physics
Institute for Stochastics and Applications

## Abstract

For classification and regression on tabular data, the dominance of gradient-boosted decision trees (GBDTs) has recently been challenged by often much slower deep learning methods with extensive hyperparameter tuning. We address this discrepancy by introducing (a) RealMLP, an improved multilayer perceptron (MLP), and (b) strong meta-tuned default parameters for GBDTs and RealMLP. We tune RealMLP and the default parameters on a meta-train benchmark with 118 datasets and compare them to hyperparameter-optimized versions on a disjoint meta-test benchmark with 90 datasets, as well as the GBDT-friendly benchmark by Grinsztajn et al. (2022). Our benchmark results on medium-to-large tabular datasets (1K–500K samples) show that RealMLP offers a favorable time-accuracy tradeoff compared to other neural baselines and is competitive with GBDTs in terms of benchmark scores. Moreover, a combination of RealMLP and GBDTs with improved default parameters can achieve excellent results without hyperparameter tuning. Finally, we demonstrate that some of RealMLP's improvements can also considerably improve the performance of TabR with default parameters.

## 1 Introduction

Perhaps the most common type of data in practical machine learning (ML) is tabular data, characterized by a fixed number of features (columns) that can take different types such as numerical or categorical, as well as a lack of the spatiotemporal structure found in image or text data. The moderate dimension and lack of symmetries make tabular data accessible to a wide variety of machine learning methods. Although tabular data is very diverse and no method is dominant on all datasets, gradient-boosted decision trees (GBDTs) exhibit excellent results on benchmarks [18, 43, 58, 69], although their superiority has been challenged by a variety of deep learning methods [3].

While many architectures for neural networks (NNs) have been proposed [3], variants of the simple multilayer perceptron (MLP) have repeatedly been shown to be good baselines for tabular NNs [15, 16, 30, 54]. Moreover, in terms of training time, MLPs are often slower than GBDTs but still considerably faster than many other architectures [18, 43]. Therefore, we study how MLPs can be improved in terms of architecture, training, preprocessing, hyperparameters, and initialization. We also demonstrate that at least some of these improvements can successfully improve TabR [17].

---

[*]Work done partially while still at University of Stuttgart.

38th Conference on Neural Information Processing Systems (NeurIPS 2024).

Even with fast and accurate NNs, the cost of extensive hyperparameter optimization can be problematic and hinder the adoption of new methods. To address this issue, we investigate the potential of better dataset-independent default parameters for MLPs and GBDTs. Specifically, we compare the library defaults (D) to our tuned defaults (TD) and (dataset-dependent) hyperparameter optimization (HPO). Unlike McElfresh et al. [43], who argue in favor HPO on GBDTs over trying NNs, our results show a better time-accuracy trade-off for trying different (tuned) default models, as is done by modern AutoML systems [10, 11].

## 1.1 Contribution

The problem of finding better default parameters can be seen as a meta-learning problem [64]. We employ a meta-train benchmark consisting of 118 datasets on which the default hyperparameters are optimized, and a disjoint meta-test benchmark consisting of 90 datasets on which they are evaluated. We consider separate default parameters for classification, optimized for classification error, and for regression, optimized for RMSE. Our benchmarks do not contain missing numerical values, and we restrict ourselves to sizes between 1K and 500K samples, cf. Section 2.

In Section 3, we introduce **RealMLP**, which improves on standard MLPs through a **bag of tricks** and **better default parameters**, tuned entirely on the meta-train benchmark. We introduce many **novel or nonstandard components**, such as preprocessing using robust scaling and smooth clipping, a new numerical embedding variant, a diagonal weight layer, new schedules, different initialization methods, etc. Our benchmark results demonstrate that it often outperforms other comparably fast NNs from the literature and can be competitive with GBDTs. To demonstrate that our bag of tricks is useful for other models, we introduce **RealTabR-D**, a version of TabR [17] including some of our tricks that, despite less extensive tuning, achieves excellent benchmark results.

In Section 4, we provide **new default parameters**, tuned on the meta-train benchmark, for XGBoost [9], LightGBM [31], and CatBoost [51]. While they cannot match HPO on average, they outperform the library defaults on the meta-test benchmark.

In Section 5, we evaluate these and other models on the meta-test **benchmark** and the benchmark by Grinsztajn et al. [18]. We also investigate several possibilities for algorithm selection and ensembling, demonstrating that algorithm selection over default methods provides a better time-performance tradeoff than HPO, thanks to our new improved default parameters and MLP.

The code for our benchmarks, including scikit-learn interfaces for the models, is available at

https://github.com/dholzmueller/pytabkit

Our code and data are archived at https://doi.org/10.18419/darus-4555.

## 1.2 Related Work

**Neural networks** Borisov et al. [3] review deep learning on tabular data and identify three main classes of methods: Data transformation methods, specialized architectures, and regularization models. In particular, recent research has mainly focused on specialized architectures based on attention [1, 7, 15, 27], including attention between datapoints [17, 37, 53, 56, 60]. However, these methods are usually significantly slower than MLPs or even GBDTs [17, 18, 43]. Our research instead expands on improvements to MLPs for tabular data such as the SELU activation function [35], bias initialization methods [61], regularization methods [30], categorical embedding layers [19], and numerical embedding layers [16].

**Benchmarks** Shwartz-Ziv and Armon [58] benchmarked three deep learning methods and noticed that they performed better on the datasets from their own papers than on other datasets. We address this issue by using more datasets and evaluating our methods on datasets that they were not tuned on. Grinsztajn et al. [18], McElfresh et al. [43], and Ye et al. [69] propose larger benchmarks and find that GBDTs still outperform deep learning methods on average, analyzing why and when this is the case. Kohli et al. [36] also emphasize the need for large benchmarks. We evaluate our methods on the benchmark by Grinsztajn et al. [18] as well as datasets from the AutoML benchmark [13] and the OpenML-CTR23 regression benchmark [12].

**Better defaults** Probst et al. [50] study the tunability of ML methods, i.e., the difference in benchmark scores between the best fixed hyperparameters and tuned hyperparameters. While their

Table 1: Characteristics of the meta-train and meta-test sets.

| | $\mathcal{B}_{\text{class}}^{\text{train}}$ | $\mathcal{B}_{\text{class}}^{\text{test}}$ | $\mathcal{B}_{\text{class}}^{\text{Grinsztajn}}$ | $\mathcal{B}_{\text{reg}}^{\text{train}}$ | $\mathcal{B}_{\text{reg}}^{\text{test}}$ | $\mathcal{B}_{\text{reg}}^{\text{Grinsztajn}}$ |
|---|---|---|---|---|---|---|
| #datasets | 71 | 48 | 18 | 47 | 42 | 28 |
| #dataset groups | 46 | 48 | 18 | 26 | 42 | 28 |
| min #samples | 1847 | 1000 | 3434 | 3338 | 1030 | 4052 |
| max #samples | 45222 | 500000 | 500000 | 48204 | 500000 | 500000 |
| max #classes | 26 | 355 | 2 | 0 | 0 | 0 |
| max #features | 561 | 10000 | 419 | 520 | 4991 | 359 |
| max #categories | 41 | 7019 | 14 | 38 | 359 | 20 |

approach involves finding better defaults, they do not evaluate them on a separate meta-test benchmark, only consider classification, and do not provide defaults for LightGBM, CatBoost, and NNs.

**Meta-learning** The problem of finding the best fixed hyperparameters is a meta-learning problem [4, 64]. Although we do not introduce or employ a fully automated method to find good defaults, we use a meta-learning benchmark setup to properly evaluate them. Wistuba et al. [66] and Pfisterer et al. [49] learn portfolios of configurations and van Rijn et al. [63] learn symbolic defaults, but neither of these papers considers GBDTs or NNs. Salinas and Erickson [55] learn large portfolios of configurations on an extensive benchmark, without studying the best defaults for individual model families. Such portfolios are successfully applied in modern AutoML methods [10, 11]. At the other end of the meta-learning spectrum, TabPFN [23] meta-learns a (tuning-free) learning method on small synthetic datasets. Unlike TabPFN, we only meta-learn hyperparameters and can therefore use fewer but larger and more realistic meta-train datasets, resulting in methods that scale to larger datasets.

## 2 Methodology

To evaluate a fixed hyperparameter configuration $\mathcal{H}$, we need a collection $\mathcal{B}^{\text{train}}$ of benchmark datasets and a scoring function that computes a benchmark score $\mathcal{S}(\mathcal{B}^{\text{train}}, \mathcal{H})$ by aggregating the errors attained by the method with hyperparameters $\mathcal{H}$ on each dataset. However, when optimizing $\mathcal{H}$ on $\mathcal{B}^{\text{train}}$, we might overfit to the benchmark and therefore ideally need a second benchmark $\mathcal{B}^{\text{test}}$ to get an unbiased score for $\mathcal{H}$. We refer to $\mathcal{B}^{\text{train}}, \mathcal{B}^{\text{test}}$ as meta-train and meta-test benchmarks and subdivide them into classification and regression benchmarks $\mathcal{B}_{\text{class}}^{\text{train}}, \mathcal{B}_{\text{reg}}^{\text{train}}, \mathcal{B}_{\text{class}}^{\text{test}}$, and $\mathcal{B}_{\text{reg}}^{\text{test}}$. We also use the Grinsztajn et al. [18] benchmark $\mathcal{B}^{\text{Grinsztajn}}$, which allows us to run more expensive baselines, since it limits training set sizes to 10K samples and contains fewer datasets due to more strict dataset inclusion criteria. Since $\mathcal{B}^{\text{train}}$ contains groups of datasets that are variants of the same dataset, for example by using different columns as targets, we use weighting factors inversely proportional to the group size.

Table 1 shows some characteristics of the considered benchmarks. The meta-test benchmark includes datasets that are more extreme in several dimensions, allowing us to test whether our default parameters generalize "out of distribution". For all datasets, we remove rows with missing numerical values and encode missing categorical values as a separate category.

### 2.1 Benchmark Data Selection

The meta-train set consists of medium-sized datasets from the UCI Repository [32], adapted from Steinwart [61]. The meta-test set consists of the datasets from the AutoML Benchmark [13] as well as the OpenML-CTR23 regression benchmark [12] with a few modifications: we subsample some large datasets and remove datasets that are already contained in the meta-train set, are too small, or have categories with too large cardinality. More details on the datasets and preprocessing can be found in Appendix C.3.

## 2.2 Aggregate Benchmark Score

To optimize the default parameters, we need to define a single benchmark score. To this end, we evaluate a method on $N_{\text{splits}} = 10$ random training-validation-test splits (60%-20%-20%) on each dataset. As metrics on individual dataset splits, we use classification error ($100\% - \text{accuracy}$) or 1-AUROC(one-vs-rest) for classification and

$$\text{nRMSE} := \frac{\text{RMSE}}{\text{standard deviation of targets}} = \sqrt{1 - R^2}$$

for regression. There are various options to aggregate these errors into a single score. Some, such as average rank or mean normalized error, depend on which other methods are included in the evaluation, hindering an independent optimization. We would like to use the geometric mean error because arguably, an error reduction from $0.02$ to $0.01$ is more valuable than an error reduction from $0.42$ to $0.41$. However, since the geometric mean error is too sensitive to cases with zero error (especially for classification error), we instead use a *shifted geometric mean error*, where a small value $\varepsilon := 0.01$ is added to the errors $\text{err}_{ij}$ before taking the geometric mean:

$$\text{SGM}_\varepsilon := \exp\left( \sum_{i=1}^{N_{\text{datasets}}} \frac{w_i}{N_{\text{splits}}} \sum_{j=1}^{N_{\text{splits}}} \log(\text{err}_{ij} + \varepsilon) \right).$$

Here, we use weights $w_i = 1/N_{\text{datasets}}$ on the meta-test set and Grinsztajn et al. [18] benchmark. On the meta-train set, we make the $w_i$ dependent on the number of related datasets, cf. Appendix C.3. In Appendix B.10, we present results for other aggregation strategies.

## 3 Improving Neural Networks

The following section presents RealMLP-TD, our improved MLP with tuned defaults, which was designed based on experiments on the meta-train benchmark. A simplified version called RealMLP-TD-S is also described. To demonstrate that our improvements can be useful for other architectures, we introduce RealTabR-D, a version of TabR that includes some of our improvements but has not been tuned as extensively as RealMLP-TD.

**Data preprocessing** In the first step of RealMLP, we apply one-hot encoding to categorical columns with at most eight distinct values (not counting missing values). Binary categories are encoded to a single feature with values $\{-1, 1\}$. Missing values in categorical columns are encoded to zero. After that, all numerical columns, including the one-hot encoded ones, are preprocessed independently as follows: Let $x_1, \ldots, x_n \in \mathbb{R}$ be the values in column $i$, and let $q_p$ be the $p$-quantile of $(x_1, \ldots, x_n)$ for $p \in [0, 1]$. Then,

$$x_{j,\text{processed}} := f(s_j \cdot (x_j - q_{1/2})), \quad f(x) := \frac{x}{\sqrt{1 + (\frac{x}{3})^2}},$$

$$s_j := \begin{cases} \frac{1}{q_{3/4} - q_{1/4}} & \text{, if } q_{3/4} \neq q_{1/4} \\ \frac{2}{q_1 - q_0} & \text{, if } q_{3/4} = q_{1/4} \text{ and } q_1 \neq q_0 \\ 0 & \text{, otherwise.} \end{cases}$$

In scikit-learn [48], this corresponds to applying a `RobustScaler` (first case) or `MinMaxScaler` (second case), and then the function $f$, which smoothly clips its input to the range $(-3, 3)$. Smooth clipping functions like $f$ have been used by, e.g., Holzmüller et al. [24] and Hafner et al. [20]. Intuitively, when features have large outliers, smooth clipping prevents the outliers from affecting the result too strongly, while robust scaling prevents the outliers from affecting the inlier scaling.

**NN architecture** Our architecture, visualized in Figure 1 (a), is a multilayer perceptron (MLP) with three hidden layers containing 256 neurons each, except for the following additions and modifications:

- RealMLP-TD employs categorical embedding layers [19] to embed the remaining categorical features with cardinality $> 8$.
- For numerical features, excluding the one-hot encoded ones, we introduce PBLD (periodic bias linear DenseNet) embeddings, which concatenate the original value to the PL embeddings proposed by Gorishniy et al. [16] and use a different periodic embedding with biases,

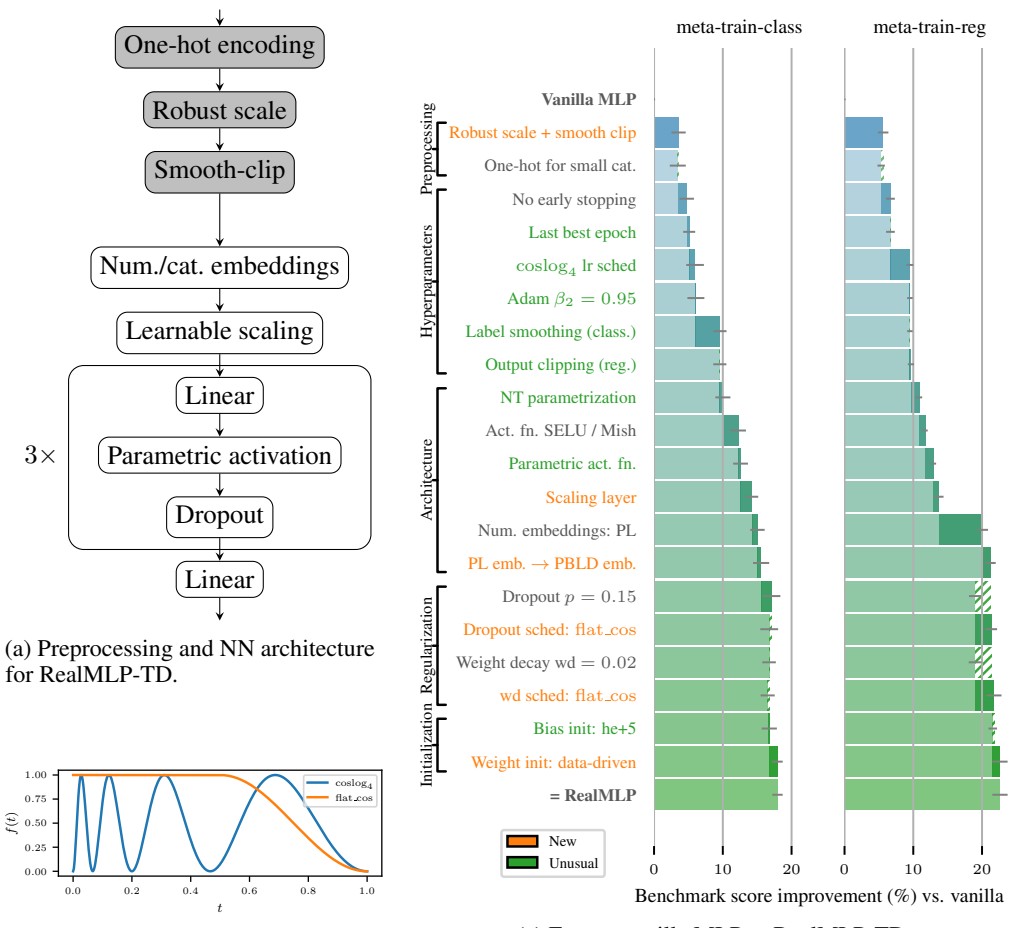

(a) Preprocessing and NN architecture for RealMLP-TD.

(b) The $\mathrm{coslog}_4$ and flat_cos schedules.

(c) From a vanilla MLP to RealMLP-TD.

Figure 1: **Components of RealMLP-TD.** Part (c) shows the result of adding one component in each step, where the best default learning rate is found separately for each step. The vanilla MLP uses categorical embeddings, a quantile transform to preprocess numerical features, default PyTorch initialization, ReLU activation, early stopping, and is optimized with Adam with default parameters. For more details, see Appendix A.4. The error bars are approximate 95% confidence intervals for the limit #splits $\to \infty$, see Appendix C.6.

inspired by Huang et al. [26] and Rahimi and Recht [52], respectively. PBLD embeddings apply separate small two-layer MLPs to each feature $x_i$ as

$$\left(x_i, \boldsymbol{W}_{\mathrm{emb}}^{(2,i)}\cos(2\pi\boldsymbol{w}_{\mathrm{emb}}^{(1,i)}x_i + \boldsymbol{b}_{\mathrm{emb}}^{(1,i)}) + \boldsymbol{b}_{\mathrm{emb}}^{(2,i)}\right) \in \mathbb{R}^4.$$

For efficiency reasons, we use 4-dimensional embeddings with $\boldsymbol{w}_{\mathrm{emb}}^{(1,i)}, \boldsymbol{b}_{\mathrm{emb}}^{(1,i)} \in \mathbb{R}^{16}, \boldsymbol{b}_{\mathrm{emb}}^{(2,i)} \in \mathbb{R}^3, \boldsymbol{W}_{\mathrm{emb}}^{(2,i)} \in \mathbb{R}^{3\times 16}$.

- To encourage (soft) feature selection, we introduce a scaling layer before the first linear layer, which is simply a matrix-vector product with a diagonal weight matrix. In other words, it computes $x_{i,\mathrm{out}} = s_i \cdot x_{i,\mathrm{in}}$, with a learnable scaling factor $s_i$ for each feature $i$. We found it beneficial to use a larger learning rate for this layer.
- Our linear layers use the neural tangent parametrization (NTP) as proposed by Jacot et al. [28], i.e., they compute $\boldsymbol{z}^{(l+1)} = d_l^{-1/2}\boldsymbol{W}^{(l)}\boldsymbol{x}^{(l)} + \boldsymbol{b}^{(l)}$, where $d_l$ is the dimension of the layer input $\boldsymbol{x}^{(l)}$. The motivation behind the use of the NTP here is that it effectively modifies the learning rate for the weight matrices depending on the input dimension $d_l$, hopefully preventing too large steps whenever the number of columns is large. We did not observe improvements when using the Adam version of the maximal update parametrization [68].

- RealMLP-TD uses parametric activation functions inspired by PReLU [21]. In general, for an activation function $\sigma$, we define a parametric version with separate learnable $\alpha_i$ for each neuron $i$:

$$\sigma_{\alpha_i}(x_i) = (1 - \alpha_i)x_i + \alpha_i\sigma(x_i) .$$

  When $\alpha_i = 1$, this recovers $\sigma$, and when $\alpha_i = 0$, the activation function is linear. As activation functions, we use SELU [35] for classification and Mish [45] for regression.
- We use dropout after each activation function. We do not use the Alpha-dropout variant originally proposed for SELU [35], as we were not able to obtain good results with it.
- For regression, at test time, the MLP outputs are clipped to the observed range during training. (We observed that this is mainly helpful for suboptimal hyperparameters.)

**Initialization**   The parameters $s_i$ of the scaling layer are initialized to 1, making it an identity function at initialization. Similarly, the parameters $\alpha_i$ of the parametric activation functions are initialized to 1, recovering the standard activation functions at initialization. We initialize weights and biases in a data-dependent fashion during a forward pass on the (possibly subsampled) training set. We rescale rows of standard-normal-initialized weight matrices to scale the variance of the output pre-activations over the dataset to one. For the biases, we use the data-dependent `he+5` initialization method [called hull+5 in 61].

**Training**   Like Gorishniy et al. [15], we use the AdamW optimizer [34, 40]. We set its momentum hyperparameters to $\beta_1 = 0.9$ and $\beta_2 = 0.95$ instead of the default $\beta_2 = 0.999$. The idea to use a smaller value for $\beta_2$ is adopted from the fastai tabular MLP [25]. RealMLP is optimized for 256 epochs with a batch size of 256. As a loss function for classification, we use softmax + cross-entropy with label smoothing [62] with parameter $\varepsilon = 0.1$. For regression, we use the MSE loss and affinely transform the targets to have zero mean and unit variance on the training and validation set.

**Hyperparameters**   We allow parameter-specific scheduled hyperparameters computed in each iteration using a base value, optional parameter-specific factors, and a schedule, as

$$\text{base\_value} \cdot \text{param\_factor} \cdot \text{schedule}\left(\frac{\text{iteration}}{\#\text{iterations}}\right),$$

allowing us, for example, to use a high learning rate factor for scaling layer parameters. Because we do not tune the number of epochs separately on each dataset, we use a multi-cycle learning rate schedule, providing multiple valleys that are usually preferable for stopping the training, while allowing high learning rates in between. Our schedule is similar to Loshchilov and Hutter [39] and Smith [59], but with a simpler analytical expression:

$$\text{coslog}_k(t) := \frac{1}{2}(1 - \cos(2\pi \log_2(1 + (2^k - 1)t))) .$$

We set $k = 4$ to obtain four cycles as shown in Figure 1 (b). To allow stopping at different levels of regularization, we schedule dropout and weight decay using the following schedule, cf. Figure 1 (b):[2]

$$\text{flat\_cos}(t) := \frac{1}{2}(1 + \cos(\pi(\max\{1, 2t\} - 1))).$$

The detailed hyperparameters can be found in Table A.1.

**Best-epoch selection**   Due to the multi-cycle learning rate schedule, we do not perform classical early stopping. Instead, we always train for the full 256 epochs and then revert the model to the epoch with the lowest validation error, which in this paper is based on classification error, or RMSE for regression. In case of a tie, we found it beneficial to use the last of the tied best epochs.

**RealMLP-TD-S**   Since certain aspects of RealMLP-TD are somewhat complex to implement, we introduce a simplified (and faster) variant called RealMLP-TD-S in Appendix A. Among the simplifications are: omitting embedding layers, using non-parametric activations, using a simpler initialization method, and omitting dropout and weight decay.

---

[2]inspired      by      a      similar      schedule      in      `https://github.com/lessw2020/Ranger-Deep-Learning-Optimizer`

**RealTabR-D**   For RealTabR-D, we adapt TabR-S-D by using our numerical preprocessing, setting Adam's $\beta_2$ to 0.95, using our scaling layer with a modification to amplify the effective learning rate by a factor of 96, adding PBLD embeddings for numerical features, and adding label smoothing for classification. More details can be found in Appendix A.3.

## 4   Gradient-Boosted Decision Trees

To find better default hyperparameters for GBDTs, we employ a semi-automatic approach: We use hyperparameter optimization libraries like hyperopt [2] and SMAC3 [38] to explore a reasonably large hyperparameter space, evaluating the benchmark score of each configuration on the meta-train benchmarks, and then perform some small manual adjustments like rounding the best obtained hyperparameters. To balance efficiency and accuracy, we fix the number of estimators to 1000 and use the `hist` method for XGBoost. We only consider the libraries' default tree-building strategies since it is one of their main differences. The tuned defaults (TD) for LightGBM (LGBM), XGBoost (XGB), and CatBoost can be found in Table C.1, C.2, and C.3, respectively.

While some of the obtained hyperparameter values might be sensitive to the tuning and benchmark setup, we observe some general trends. First, row subsampling is used in all tuned defaults, while column subsampling is rarely applied. Second, trees are generally allowed to be deeper for regression than for classification. Third, the Bernoulli bootstrap in CatBoost is competitive with the Bayesian bootstrap while also being faster.

## 5   Experiments

In the following, we evaluate different methods with library defaults (D), tuned defaults (TD), and hyperparameter optimization (HPO). Recall that TD uses fixed parameters optimized on the meta-train benchmarks, while HPO tunes hyperparameters on each dataset split independently. All methods except random forests select the best iteration/epoch on the validation set of the respective dataset split based on accuracy / RMSE. All NN-based regression methods standardize the labels for training.

### 5.1   Methods

We provide methods in the following variants:

- **D**: Default parameters, taken from the original library if possible (Appendix C.1).
- **TD**: Tuned default parameters from Section 3 and Section 4.
- **HPO**: Hyperparameters optimized separately for every train-test split on every dataset, using 50 steps of random search. Search spaces are specified in Appendix C.2 and are usually adapted from original or popular papers.

As tree-based methods, we use XGBoost (**XGB**), LightGBM (**LGBM**), and **CatBoost** from the respective libraries, as well as random forest (**RF**) from scikit-learn. The variant **XGB-PBB-D** uses meta-learned default parameters from Probst et al. [50]. For neural methods, we compare to **MLP**, **ResNet**, and FT-Transformer (**FTT**) from Gorishniy et al. [15], **MLP-PLR** from Gorishniy et al. [16], as well as **TabR** and **TabR-S** (without numerical embeddings) from Gorishniy et al. [17]. We compare these methods to **RealMLP** and **RealTabR** from Section 3. In addition, we investigate **Best**, which on each dataset split selects the method with the best validation score out of XGB, LGBM, CatBoost, and MLP-PLR (for Best-D) or RealMLP (for Best-TD and Best-HPO). **Ensemble** builds a weighted ensemble out of the same methods as Best, using the method of Caruana et al. [5] with 40 greedy selection steps as in Salinas and Erickson [55].

We do not run FTT, RF-HPO, and TabR-HPO on all benchmarks since some benchmarks (especially meta-test) are more expensive to run and these methods may run into out-of-memory errors.

### 5.2   Results

Figure 2 shows the results of the aforementioned methods on all benchmarks, along with their runtimes on a CPU. *Note that XGB results on some (mainly meta-test) datasets are affected by a bug in handling rare categories, see Appendix B.*

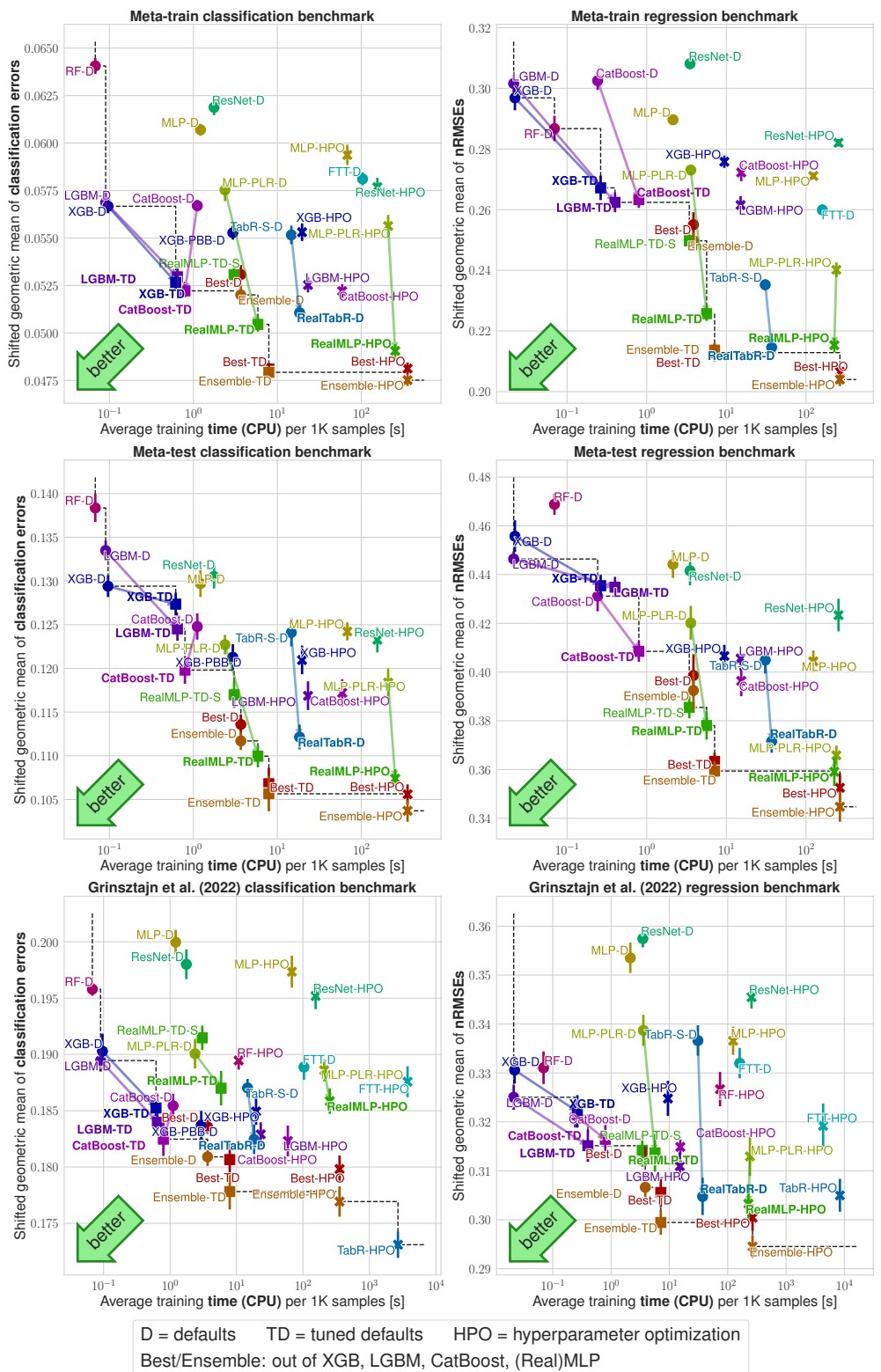

Figure 2: **Benchmark scores on all benchmarks vs. average training time.** The $y$-axis shows the shifted geometric mean ($\mathrm{SGM}_\varepsilon$) classification error (left) or nRMSE (right) as explained in Section 2.2. The $x$-axis shows average training times per 1000 samples (measured on $\mathcal{B}^{\mathrm{train}}$ for efficiency reasons), see Appendix C.7. The error bars are approximate 95% confidence intervals for the limit #splits $\to \infty$, see Appendix C.6. *Note that XGB results on some (mainly meta-test) datasets are affected by a bug in handling rare categories, see Appendix B.*

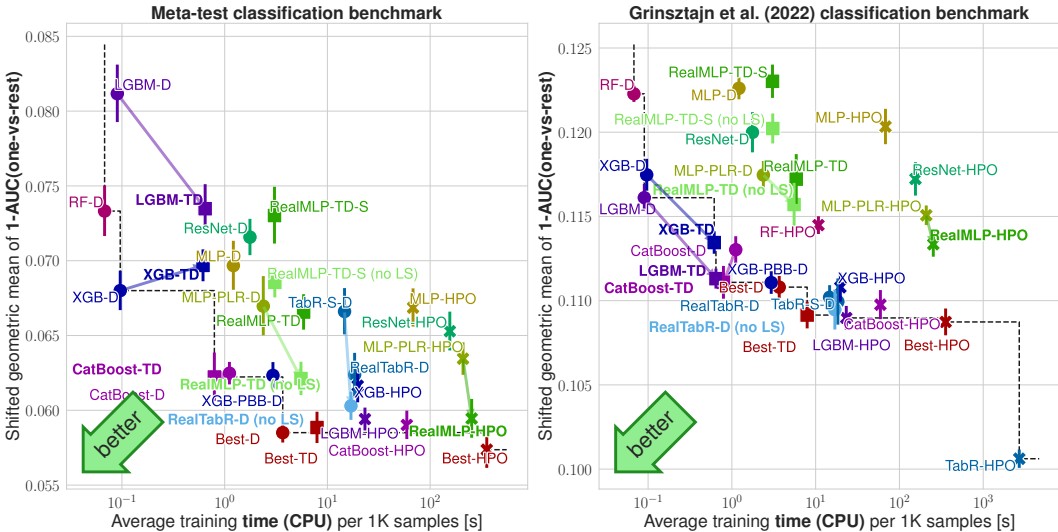

Figure 3: **Benchmark scores vs. average training time for AUC.** Methods labeled "no LS" deactivate label smoothing. Stopping and best-epoch selection are performed on accuracy, while HPO is performed on AUC. See Figure B.3 for stopping on cross-entropy. The $y$-axis shows the shifted geometric mean $(\mathrm{SGM}_\varepsilon)$ $1 - \mathrm{AUC}$ as explained in Section 2.2. The $x$-axis shows average training times per 1000 samples (measured on $\mathcal{B}^{\mathrm{train}}$ for efficiency reasons), see Appendix C.7. The error bars are approximate 95% confidence intervals for the limit #splits $\to \infty$, see Appendix C.6.

**How good are tuned defaults on new datasets?**     To answer this question, we compare the relative gaps between TD and HPO benchmark scores on the meta-test benchmarks to those on the meta-train benchmarks. The gap between RealMLP-HPO and RealMLP-TD is not much larger on the meta-test benchmarks, indicating that the tuned defaults transfer very well to the meta-test benchmark. For GBDTs, tuned defaults are competitive with HPO on the meta-train set, but not as good on the meta-test set. Still, they are considerably better than the untuned defaults on the meta-test set. Note that we did not limit the TD parameters to the literature search spaces for the HPO models (cf. Appendix C.2); for example, XGB-TD uses a smaller value of min_child_weight for classification and CatBoost-TD uses deeper trees and Bernoulli boosting. The XGBoost defaults XGB-PBB-D from Probst et al. [50] outperform XGB-TD on $\mathcal{B}^{\mathrm{test}}_{\mathrm{class}}$, perhaps because their benchmark is more similar to $\mathcal{B}^{\mathrm{test}}_{\mathrm{class}}$ or because XGB-PBB-D uses more estimators (4168) and deeper trees.

**RealMLP and RealTabR perform strongly among NNs.**     On most benchmarks, RealMLP-TD and RealTabR-D bring considerable improvements over MLP-PLR-D and TabR-S-D, at slightly larger runtimes, respectively. Similarly, RealMLP-HPO improves the results of MLP-PLR-HPO. TabR and FTT are notably slower than MLP-based methods on CPUs, while the difference is less pronounced on GPUs (Figure C.2). While RealMLP-TD beats TabR-S-D on many benchmarks, RealTabR-D performs even better on four out of six benchmarks, especially all regression benchmarks. On the Grinsztajn et al. [18] benchmark where we can afford to run more baselines, TabR-HPO performs best according to many aggregation metrics. It performs especially well on the *electricity* dataset, where MLPs struggle to learn high-frequency patterns [18].

**RealMLP and RealTabR are competitive with tree-based models.**     On the meta-train and meta-test benchmarks, RealMLP and RealTabR perform better than GBDTs in terms of shifted geometric mean error, while also being comparable or slightly better in terms of other aggregations like mean normalized error (Appendix B.10) or win-rates (Appendix B.12). On the Grinsztajn et al. [18] benchmark, RealMLP performs worse than CatBoost for classification and comparably for regression, while RealTabR-D performs comparably to CatBoost-TD for classification and better for regression.

**Among GBDTs, CatBoost defaults are better and slower.**     Several papers have found CatBoost to perform favorably among GBDTs while being more computationally expensive to train [8, 33, 43, 51, 69]. We observe the same for our tuned defaults on most benchmarks.

**Simply trying all default algorithms is faster and very often better than (naive) single-algorithm HPO.** When comparing Best-TD to 50-step HPO on RealMLP or GBDTs, we notice that Best-TD is faster on average, while also being competitive with the best of the HPO models. In comparison, Best-D is often outperformed by RealMLP-HPO. We also note that ensemble selection [5] usually gives 0–3% improvement on the benchmark score compared to selecting the best model, and can potentially be further improved [6]. Unlike McElfresh et al. [43], who argue in favor of CatBoost-HPO over trying NNs, our results favor model portfolios as used in modern AutoML systems [10].

**Analyzing NN improvements** Figure 1 (c) shows how adding the proposed RealMLP components to a simple MLP improves the meta-train benchmark performance. However, these results depend on the order in which components are added, which is addressed by a separate ablation study in Appendix B. For example, the large weight decay value makes RealMLP-TD sensitive to changes in some other hyperparameters like $\beta_2$. We also show in Appendix B.8 that our architectural improvements alone are beneficial when applied to MLP-D directly, although non-architectural aspects are at least as important. In particular, our numerical preprocessing is easy to adopt and often beneficial for other NNs as well (Appendix B.7). The scaling layer and PBLD embeddings are easy to use and turned out to be effective within RealTabR-D as well. If affordable, larger stopping patiences and the use of (cyclic) learning rate schedules can be useful, while label smoothing is influential but can be detrimental for metrics like AUROC (Figure 3, Appendix B.5).

**Dependence on benchmark choices** We observe that choices in benchmark design can affect the interpretation of the results. The use of different aggregation metrics than the shifted geometric mean reduces the advantage of TD methods (Appendix B.10). For classification, using AUROC instead of classification error (Figure 3, Appendix B.5) favors GBDTs. Different dataset selection and preprocessing criteria on different benchmarks lead to large differences between benchmarks in the average errors, as indicated by the $y$-axis scaling in Figure 2.

**Further insights** In Appendix B, we present additional experimental results. We compare bagging and refitting for RealMLP-TD and LGBM-TD, finding that refitting multiple models is often better on average. We demonstrate that GBDTs benefit from high early stopping patiences for classification, especially when using accuracy as the stopping metric. When considering AUROC as a stopping metric, we show that stopping on cross-entropy is preferable to accuracy (Appendix B.5).

**Limitations** While our benchmarks cover medium-to-large tabular datasets in standard settings, it is unclear to which extent the obtained defaults can generalize to very small datasets, distribution shifts, datasets with missing numerical values, and other metrics such as log-loss. Additionally, runtimes and the resulting tradeoffs may change with different parallelization, hardware, or (time-aware) HPO algorithms. For computational reasons, we only use a single training-validation split per train-test split. This means that HPO can overfit the validation set more easily than in a cross-validation setup. While we extensively benchmark different NN models from the literature, we do not attempt to equalize non-architectural aspects, and our work should therefore not be seen as a comparison of architectures. We compared to TabR-S-D as a recent promising method with good default parameters [17, 69]. However, due to a surge of recently published deep tabular models [e.g., 7, 8, 29, 33, 41, 57, 67], it is unclear what the current "best" deep tabular model is. In particular, ExcelFormer [7] also promises strong-performing default parameters. For GBDTs, due to the cost of running the benchmarks, our limits on the depth and number of trees are on the lower side of the literature.

## 6   Conclusion

In this paper, we studied the potential of improved default parameters for GBDTs and an improved MLP, evaluated on a large separate meta-test benchmark as well as the benchmark by Grinsztajn et al. [18], and investigated the time-accuracy tradeoffs of various algorithm selection and ensembling scenarios. Our improved MLP mostly outperforms other NNs from the literature with moderate runtime and is competitive with GBDTs in terms of benchmark scores. Since many of the proposed improvements to NNs are orthogonal to the improvements in other papers, they offer exciting opportunities for combinations, as we demonstrated with our RealTabR variant. While the "NNs vs GBDTs" debate remains interesting, our results demonstrate that with good default parameters, it is worth trying both algorithm families even with a moderate training time budget.

## Acknowledgments and Disclosure of Funding

We thank Gaël Varoquaux, Frank Sehnke, Katharina Strecker, Ravid Shwartz-Ziv, Lennart Purucker, and Francis Bach for helpful discussions. We thank Katharina Strecker for help with code refactoring.

Funded by Deutsche Forschungsgemeinschaft (DFG, German Research Foundation) under Germany's Excellence Strategy - EXC 2075 – 390740016. The authors thank the International Max Planck Research School for Intelligent Systems (IMPRS-IS) for supporting David Holzmüller. LG acknowledges support in part by the French Agence Nationale de la Recherche under Grant ANR-20-CHIA-0026 (LearnI). Part of this work was performed on the computational resource bwUniCluster funded by the Ministry of Science, Research and the Arts Baden-Württemberg and the Universities of the State of Baden-Württemberg, Germany, within the framework program bwHPC. Part of this work was performed using HPC resources from GENCI–IDRIS (Grant 2023-AD011012804R1 and 2024-AD011012804R2).

**Contribution statement**    DH and IS conceived the project. DH implemented and experimentally validated the newly proposed methods and wrote the initial paper draft. DH and LG contributed to benchmarking, plotting, and implementing baseline methods. LG and IS helped revise the draft. IS supervised the project and contributed dataset downloading code.

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

# Appendices

**Appendix Contents.**

# A   Further Details on Neural Networks

The detailed hyperparameter settings for RealMLP-TD and RealMLP-TD-S are listed in Table A.1.

## A.1   RealMLP-TD Details

**Architecture**   To make the binary and multi-class cases more similar, we use two output neurons in the binary case, using the same loss function as in the multi-class case.

**Initialization**   We initialize categorical embedding parameters from $\mathcal{N}(0, 1)$. We initialize the components of $\boldsymbol{w}_{\mathrm{emb}}^{(1,i)}$ from $\mathcal{N}(0, 0.1^2)$ and of $\boldsymbol{b}_{\mathrm{emb}}^{(1,i)}$ from $\mathcal{U}[-\pi, \pi]$. The other numerical embedding parameters are initialized according to PyTorch's default initialization, that is, from the uniform distribution $\mathcal{U}[-1/\sqrt{16}, 1/\sqrt{16}]$. For weights and biases of the linear layers, we use a data-dependent initialization. The initialization is performed on the fly during a first forward pass of the network on the training set (which can be subsampled adaptively not to use more than 1 GB of RAM). We realize this by providing `fit_transform()` methods similar to a pipeline in scikit-learn. For the weight matrices, we use a custom two-step procedure: First, we initialize all entries from $\mathcal{N}(0, 1)$. Then, we rescale each row of the weight matrix such that the outputs $\frac{1}{\sqrt{d_l}} \boldsymbol{W}^{(l)} \boldsymbol{x}_j^{(l)}$ have variance 1 over the dataset (i.e. when considering the sample index $j \in \{1, \ldots, n\}$ as a uniformly distributed random variable). This is somewhat similar to the LSUV initialization method [44]. For the biases, we use the data-dependent `he+5` initialization method [called hull+5 in 61].

**Training**   We implement weight decay as in PyTorch using $\theta \leftarrow \theta - \mathrm{lr} \cdot \mathrm{wd} \cdot \theta$, which includes the learning rate unlike the original version [40].

## A.2   RealMLP-TD-S Details

For RealMLP-TD-S, we make the following changes compared to RealMLP-TD:

- We apply one-hot encoding to all categorical variables and do not apply categorical embeddings.
- We do not apply numerical embeddings.
- We use the standard non-parametric versions of the SELU and Mish activation functions.
- We do not use dropout and weight decay.
- We use simpler weight and bias initializations: We initialize weights and biases from $\mathcal{N}(0, 1)$, except in the last layer, where we initialize them to zero.
- We do not clip the outputs, even in the regression case.
- We apply a different base learning rate in the regression case.

## A.3   RealTabR-D Details

To obtain RealTabR-D, we modify TabR-S-D in the following ways:

- We replace the standard numerical preprocessing (a modified quantile transform) with our robust scaling and smooth clipping.
- We set Adam's $\beta_2$ to 0.95 instead of 0.999.
- We use our scaling layer, but modify it to obtain a higher effective learning rate. We do this by modifying the forward pass to

$$x_{i,\mathrm{out}} = \gamma \cdot s_i \cdot x_{i,\mathrm{in}} \ ,$$

  while initializing $s_i$ to $1/\gamma$. This will multiply the gradients of $s_i$ by $\gamma$, which will be ignored by Adam's normalization (when neglecting Adam's $\varepsilon$ parameter). It will also multiply the optimizer updates by $\gamma$, leading to approximately the same effect as multiplying the learning rate by $\gamma$. However, a difference is that multiplying the learning rate by $\gamma$ will also lead to stronger weight decay updates in PyTorch's AdamW implementation, while the introduction of $\gamma$ does not increase the relative magnitude of weight decay updates. We chose the version with $\gamma$ for simplicity of implementation. While RealMLP-TD uses a learning rate factor of 6 for the scaling layer, it uses a higher base learning rate due to the use of the neural tangent parametrization. For all layers except the first one, which have width 256, the neural tangent

Table A.1: **Overview of hyperparameters for RealMLP-TD and RealMLP-TD-S.**

| Hyperparameter | RealMLP-TD classification | RealMLP-TD regression | RealMLP-TD-S classification | RealMLP-TD-S regression |
|---|---|---|---|---|
| Num. embedding type | PBLD | PBLD | None | None |
| Num. embedding periodic init std. | 0.1 | 0.1 | — | — |
| Num. embedding hidden dimension | 16 | 16 | — | — |
| Num. embedding dimension | 4 | 4 | — | — |
| Max one-hot size (without missing) | 8 | 8 | $\infty$ | $\infty$ |
| Num. preprocessing | robust scale + smooth clip | | | |
| Categorical embedding dimension | 8 | 8 | — | — |
| Categorical embedding initialization | $\mathcal{N}(0,1)$ | $\mathcal{N}(0,1)$ | — | — |
| Use scaling layer | yes | | | |
| Scaling layer initialization | 1.0 (constant) | | | |
| Number of linear layers | 4 | | | |
| Hidden layer sizes | [256, 256, 256] | | | |
| Activation function | SELU | Mish | SELU | Mish |
| Use parametric activation function | yes | yes | no | no |
| Parametric activation function initialization | 1.0 | 1.0 | — | — |
| Linear layer parametrization | NTP | | | |
| Last linear layer weight initialization | data-driven | data-driven | zero | zero |
| Other linear layer weight initialization | data-driven | data-driven | std normal | std normal |
| Last linear layer bias initialization | he+5 | he+5 | zero | zero |
| Other linear layer bias initialization | he+5 | he+5 | std normal | std normal |
| Optimizer | AdamW | | | |
| Batch size | 256 | | | |
| Number of epochs | 256 | | | |
| Adam $\beta_1$ | 0.9 | | | |
| Adam $\beta_2$ | 0.95 | | | |
| Adam $\varepsilon$ | 1e-8 | | | |
| Learning rate (base value) | 0.04 | 0.2 | 0.04 | 0.07 |
| Learning rate schedule | $\mathrm{coslog}_4$ | | | |
| Learning rate (num. emb. factor) | 0.1 | 0.1 | — | — |
| Learning rate (scaling layer factor) | 6 | | | |
| Learning rate (bias factor) | 0.1 | | | |
| Learning rate (param. act. factor) | 0.1 | 0.1 | — | — |
| Dropout probability (base value) | 0.15 | 0.15 | 0.0 | 0.0 |
| Dropout schedule | flat_cos | flat_cos | — | — |
| Weight decay (base value) | 0.02 | 0.02 | 0.0 | 0.0 |
| Weight decay schedule | flat_cos | flat_cos | — | — |
| Weight decay (bias factor) | 0.0 | 0.0 | — | — |
| Loss function | cross-entropy | MSE | cross-entropy | MSE |
| Label smoothing $\varepsilon$ | 0.1 | — | 0.1 | — |
| Standardize targets during training | — | yes | — | yes |
| Output min-max clipping | — | yes | — | no |
| Best epoch selection metric | class. error | MSE | class. error | MSE |
| Best epoch selection method | last best validation error | | | |

parametrization in RealMLP-TD uses a factor similar to $\gamma$, which is set to $1/16 = 1/\sqrt{256}$. Hence, RealMLP-TD without NTP should use a base learning rate for these layers that is smaller by a factor of $1/16$, and therefore use a learning rate factor of $6 \cdot 16 = 96$ for the scaling layer. Consequently, we set $\gamma := 96$ for RealTabR-D without further tuning, noting that it performed significantly better than $\gamma = 6$ on the meta-train benchmarks.

- We use our PBLD embeddings for numerical features before the scaling layer, instead of no numerical embeddings in TabR-S-D. In order to make every experiment run on a GPU with 24GB RAM, we decrease the dimension of the hidden embedding layer from 16 to 8, although using 16 would have performed slightly better in our experiments on the meta-train benchmarks.
- For classification, we use label smoothing with parameter $\varepsilon = 0.1$.

Since we adapted hyperparameters like learning rate and weight decay from TabR-S-D without meta-learning them, we refer to the resulting method as RealTabR-D and not RealTabR-TD. We did not include other tricks from RealMLP-TD for various reasons:

- Brief experiments with NTP and the Mish activation deteriorated the performance.
- Parametric activations and increased stopping patience showed small improvements but were excluded due to a larger runtime.
- Other tricks were not tried due to limited time of experimentation, expected increases in the already somewhat large runtime, and/or implementation complexity.

## A.4 Details on Cumulative Ablation

Here, we provide more details on the vanilla MLP and the ablation steps from Figure 1 (c). For each step, we choose the best default learning rate out of a learning rate grid, using $\{0.0004, 0.0007, 0.001, 0.0015, 0.0025, 0.004, 0.007, 0.01, 0.015\}$ for NNs using standard parametrization and $\{0.01, 0.02, 0.03, 0.04, 0.07, 0.1, 0.2, 0.3, 0.4\}$ for NNs using neural tangent parametrization.

- Vanilla MLP: We use three hidden layers with 256 hidden neurons in each layer, just like RealMLP-TD, and the ReLU activation function. Each linear layer uses standard parametrization and the PyTorch default initialization, which is uniform from $[-1/\sqrt{\text{fan\_in}}, 1/\sqrt{\text{fan\_in}}]$ for both weights and biases, where fan_in is the input dimension. Categorical features are embedded using embedding layers, using eight-dimensional embeddings for each feature. Numerical features are transformed using a scikit-learn `QuantileTransformer` to approximately normal-distributed features. Optimization is performed using Adam with constant learning rate and default parameters $\beta_1 = 0.9, \beta_2 = 0.999, \varepsilon = 10^{-8}$ for at most 256 epochs with batch size 256, with constant learning rate. If the best validation error (classification error or RMSE) does not improve for 40 epochs, training is stopped. In each case, the model is reverted to the parameters of the epoch with the best validation score, using the first best epoch in case of a tie.
- Robust scale + smooth clip: We replace the `QuantileTransformer` with robust scaling and smooth clipping.
- One-hot for small cat.: As in RealMLP-TD, we use one-hot encoding for categories with at most eight values, not counting missing values.
- No early stopping: We always train the full 256 epochs.
- Last best epoch: In case of a tie, we use the last of the best epochs.
- $\text{coslog}_4$ lr sched: We use the $\text{coslog}_4$ learning rate schedule instead of a constant one.
- Adam $\beta_2 = 0.95$: We set $\beta_2 = 0.95$.
- Label smoothing (class.): We enable label smoothing with $\varepsilon = 0.1$ in the classification case.
- Output clipping (reg.): For regression, outputs are clipped to the min-max range observed during training.
- NT parametrization: We use the neural tangent parametrization for linear layers, setting the bias learning rate factor to $0.1$.
- Act. fn. SELU / Mish: We change the activation function from ReLU to SELU (classification) or Mish (regression).
- Parametric act. fn.: We use parametric versions of the activation functions, with a learning rate factor of $0.1$ for the parameters.
- Scaling layer: We use a scaling layer with a learning rate factor of $6$ before the first linear layer.
- Num. embeddings: PL: We apply the PL embeddings [16] to numerical features.
- Num. embeddings: PBLD: We apply our PBLD embeddings instead.
- Dropout $p = 0.15$: We apply dropout with probability $0.15$.
- Dropout sched: flat_cos: We apply the flat_cos schedule to the dropout probability.
- Weight decay wd $= 0.02$: We apply weight decay (as in AdamW, PyTorch version) with value $0.02$.
- wd sched: flat_cos: We apply the flat_cos schedule to weight decay.
- Bias init: he+5: We apply the he+5 bias initialization method from Steinwart [61] (originally called hull+5).
- Weight init: data-driven: We apply our data-driven weight initialization method.

### A.5 Discussion

Here, we discuss some of the design decisions behind RealMLP-TD and possible trade-offs. First, our implementation allows us to train RealMLP-TD in a vectorized fashion on multiple train-validation-test splits at the same time. On the one hand, this can lead to speedups on GPUs when training multiple models in parallel, including on the benchmarks. On the other hand, it can hinder the implementation of certain methods like patience-based early stopping or loss-based learning rate schedules. While our ablations in Appendix B.1 show the advantage of our multi-cycle schedule over decreasing learning rate schedules, the latter ones could potentially enable a faster average training time through low-patience early stopping. An interesting follow-up question could be whether the multi-cycle schedule still works well with larger-patience early stopping.

Regarding categorical embeddings, our meta-train benchmark does not contain many high-cardinality categorical variables, and we were not able to conclude whether categorical embeddings are helpful or harmful compared to one-hot encoding (see Appendix B.1). Our motivation to include categorical embeddings stems from Guo and Berkhahn [19] as well as their potential to be more efficient for high-cardinality categorical variables. However, in practice, we find pure one-hot encoding to be faster on most datasets. Regarding the embedding size, we found that 4 already gave good results for numerical embeddings and decided to use 8 for categorical variables.

Additionally, other speed-accuracy tradeoffs are possible. Especially for regression, we observed that more epochs and larger hidden layers can be helpful. When faster networks are desired, the omission of numerical and categorical embedding layers as well as parametric activations from RealMLP-TD can be helpful, while the other omissions in RealMLP-TD-S do not considerably affect the training time. Of course, using larger batch sizes can also be helpful for larger datasets.

One caveat for classification is that cross-entropy with label smoothing is not a proper scoring rule, that is, in the infinite-sample limit, it is not minimized by the true probabilities $P(y|x)$ [14]. Hence, label smoothing might not be suitable when other classification error metrics are used, as demonstrated in Appendix B.5 for AUROC.

## B   More Experiments

In this section, we present more experimental results. Note that XGBoost results are affected by a bug where, if a categorical value is not present in the training or validation set, it could cause adjacent categorical values to be encoded differently during training, validation, and evaluation. This affects the results mainly on the meta-test benchmarks, where the SGM scores for XGB-TD and XGB-D are around 2% lower after fixing the bug. These differences are not large enough to affect our qualitative conclusions. Due to the large computational cost, we did not rerun XGB-HPO and XGB-PBB-D after fixing the bug, and we provide the old XGB-TD and XGB-D results for a fair comparison to XGB-HPO and XGB-PBB-D.

### B.1   MLP Ablations

To assess the importance of different improvements in RealMLP-TD, we perform an ablation study. We perform the ablation study only on the *meta-train* benchmarks, first because they are considerably faster to run, and second because we tune the default parameters only on the meta-train benchmarks. Since the hyperparameters of RealMLP-TD have been tuned on the meta-train benchmarks, the ablation scores are not unbiased but represent some of the considerations that have been made when tuning the defaults. For each ablation, we multiply the default learning rate by learning rate factors from the grid $\{0.1, 0.15, 0.25, 0.35, 0.5, 0.7, 1.0, 1.4, 2.0, 3.0, 4.0\}$ and pick the best one. Table B.1 shows the results of the ablation study in terms of the relative increase of the benchmark score for each ablation.

In general, we observe that ablations often lead to much larger changes for regression than for classification. Perhaps this is because nRMSE is more sensitive to outliers compared to classification error. Another factor could be that the classification benchmark contains more datasets than the regression benchmark. For the specific ablations, we observe a few things:

- For the **numerical embeddings**, we see that PBLD outperforms PL, PLR, and no numerical embeddings. Contrary to Gorishniy et al. [16], PL embeddings perform better than PLR

embeddings in our setting. While the configurations with PLR and no numerical embeddings appear extremely bad for regression, we observed that they can perform more benignly with lower weight decay values.

- Using the Adam default value of $\beta_2 = 0.999$ instead of our default $\beta_2 = 0.95$ leads to considerably worse performance, especially for regression. As for numerical embeddings, we observed that the difference is less pronounced at lower weight decay values.
- Using a cosine decay **learning rate schedule** instead of our multi-cycle schedule leads to small deteriorations. A constant learning rate schedule performs even worse, especially for regression.
- Not employing **label smoothing** for classification is detrimental by around 1.8%.
- The **learnable scaling** layer yields improvements around 1.2% on both benchmarks.
- The use of **parametric activations** results in a considerable 4.8% improvement for regression but is insignificant for classification. We observed that parametric activations can sometimes alleviate optimization difficulties with weight decay.
- The differences between **activation functions** are rather small. For classification, Mish is competitive with SELU in this ablation but we found it to be worse in some other hyperparameter settings, so we keep SELU as the default. For regression, Mish performs best.
- For **dropout** and **weight decay**, we observe that they yield comparable but not always significant benefits for classification and regression. Scheduling dropout and weight decay parameters with the flat_cos schedule is helpful for regression, but not for classification in this setting.
- When comparing the **standard parametrization** (SP) to the neural tangent parametrization (NTP), we disable weight decay for a fair comparison. Moreover, for SP, we set the learning rate factors for weight and bias layers to $1/16 = 1/\sqrt{256}$. This is because, for the weights in NTP, the effective updates by Adam are damped by this factor in all hidden layers except the first one. Compared to NTP without weight decay, SP without weight decay performs insignificantly worse on both benchmarks. It is unclear to us why the parametrization, which has a considerable influence on how the effective learning speed of the first linear layer scales with the number of features, is apparently of little importance.
- When comparing the data-dependent **initialization** of RealMLP-TD to a vanilla initialization with standard normal weights and zero biases, we see that the data-dependent initialization gains around 1% on both benchmarks.
- For selecting the best epoch, we consider selecting the **first best epoch** instead of the last best epoch in case of a tie. This is only relevant for classification metrics like classification error, where ties are somewhat likely to occur, especially on small and "easy" datasets. We observe a non-significant 0.4% deterioration in the benchmark score.
- We do not observe a significant difference when using **one-hot encoding** for all categorical variables, since our benchmarks contain only very few datasets with large-cardinality categorical variables.

## B.2   MLP Preprocessing

In Table B.2, we compare different preprocessing methods for numerical features. Since we want to compare these methods in a relatively conventional setting, we apply them to RealMLP-TD-S (without numerical embeddings) and before one-hot encoding. We compare the following methods:

- Robust scaling and smooth clipping, our method used in RealMLP-TD and RealMLP-TD-S and described in Section 3.
- Robust scaling without smooth clipping.
- Standardization, i.e. subtracting the mean and dividing by the standard deviation. If the standard deviation of a feature is zero, we set the feature to zero.
- Standardization followed by smooth clipping.
- The quantile transformation from scikit-learn [48] with normal output distribution, which is popular in recent works [16–18, 43].
- A variant of the quantile transform, which we call the RTDL version, used by Gorishniy et al. [15] and Gorishniy et al. [17]. This version uses a dataset-size-dependent number of quantiles and adds some noise before fitting the transformation. It also uses a normal output distribution.

Table B.1: **Ablation experiments for RealMLP-TD.** We re-tune the learning rate (picking the one with the best $\mathrm{SGM}_\varepsilon$ benchmark score) for each ablation separately. For each ablation, we specify the increase in the benchmark score ($\mathrm{SGM}_\varepsilon$) relative to RealMLP-TD, with approximate 95% confidence intervals (Appendix C.6), and the best learning rate factor found. In the cases where values are missing, the corresponding option is already the default.

| Ablation | meta-train-class | | meta-train-reg | |
|---|---|---|---|---|
| | Error increase in % | best lr factor | Error increase in % | best lr factor |
| MLP-TD (without ablation) | 0.0 [0.0, 0.0] | 1.0 | 0.0 [0.0, 0.0] | 1.0 |
| Num. embeddings: PL | 0.7 [-0.0, 1.4] | 1.0 | 0.5 [-0.5, 1.6] | 1.0 |
| Num. embeddings: PLR | 4.2 [2.8, 5.7] | 1.0 | 19.0 [13.7, 24.5] | 0.25 |
| Num. embeddings: None | 2.3 [1.7, 2.9] | 1.0 | 20.6 [19.4, 21.8] | 0.25 |
| Adam $\beta_2 = 0.999$ instead of $\beta_2 = 0.95$ | 2.0 [1.6, 2.4] | 2.0 | 22.8 [21.3, 24.4] | 0.35 |
| Learning rate schedule = cosine decay | 1.1 [0.6, 1.5] | 1.0 | 0.4 [-0.5, 1.2] | 3.0 |
| Learning rate schedule = constant | 1.8 [0.9, 2.8] | 0.25 | 13.5 [11.9, 15.0] | 0.15 |
| No label smoothing | 1.8 [1.2, 2.5] | 4.0 | | |
| No learnable scaling | 1.4 [0.7, 2.1] | 2.0 | 1.0 [-0.0, 2.0] | 2.0 |
| Non-parametric activation | 0.5 [-0.2, 1.2] | 3.0 | 4.8 [3.4, 6.2] | 0.35 |
| Activation=Mish | -0.0 [-0.6, 0.6] | 3.0 | | |
| Activation=ReLU | 0.5 [-0.1, 1.2] | 2.0 | 0.7 [-0.1, 1.6] | 1.0 |
| Activation=SELU | | | 2.3 [1.2, 3.6] | 1.0 |
| No dropout | 0.8 [0.2, 1.3] | 3.0 | 0.8 [-0.5, 2.1] | 1.4 |
| Dropout prob. 0.15 (constant) | -0.1 [-1.0, 0.8] | 1.4 | 3.6 [3.0, 4.2] | 1.0 |
| No weight decay | 0.8 [-0.2, 1.8] | 0.5 | 0.9 [-0.1, 1.9] | 0.5 |
| Weight decay = 0.02 (constant) | -0.3 [-0.7, 0.1] | 3.0 | 3.1 [1.7, 4.4] | 1.4 |
| Standard param + no weight decay | 1.1 [0.2, 2.1] | 0.5 | 1.3 [0.7, 1.8] | 0.7 |
| No data-dependent init | 0.9 [0.1, 1.8] | 3.0 | 1.2 [0.2, 2.2] | 1.4 |
| First best epoch instead of last best | 0.4 [-0.1, 1.0] | 4.0 | 0.0 [-0.0, 0.0] | 1.0 |
| Only one-hot encoding | -0.0 [-0.1, 0.0] | 1.0 | 0.0 [-0.0, 0.0] | 1.0 |

- The recent kernel density integral transform [42] with normal output distribution, which interpolates between the quantile transformation and min-max scaling, with default parameter $\alpha = 1$.

Table B.2 shows that on the meta-train benchmark, robust scaling and smooth clipping performs best for both classification and regression.

## B.3 Bagging, Refitting, and Ensembling

In our benchmark, for each training-test split, we only train one model on one training-validation split for efficiency reasons. However, ensembling and cross-validation techniques usually allow additional improvements to models. Here, we study multiple variants for RealMLP-TD and LGBM-TD. Let $\mathcal{D}$ be the available data for training and validation, split into five equal-size subsets $\mathcal{D}_1, \ldots, \mathcal{D}_5$. (When $|\mathcal{D}|$ is not divisible by five, $\mathcal{D}_1 \cup \ldots \cup \mathcal{D}_5 \subsetneq \mathcal{D}$ since we need equal-size validation sets for vectorized NNs.) Let $f_{\mathcal{D},t}(X)$ be the predictions on inputs $X$ of the model trained on training set $\mathcal{D}$ after $t \in \{1, \ldots, T\}$ epochs (for NNs) or iterations (for LGBM). For classification, we consider the class probabilities as predictions. Let $L_{\mathcal{D}'}(f_{\mathcal{D},t})$ be the loss of $f_{\mathcal{D},t}$ on dataset $\mathcal{D}'$. Then, we compare the test errors of an ensemble of $M = 1$ or $M = 5$ models, trained using bagging or refitting, with individual or joint stopping (best-epoch selection), which is formally given as follows:

$$y_{\text{pred}} := \frac{1}{M} \sum_{i=1}^{M} f_{\tilde{\mathcal{D}}_i, t_i^*}(X_{\text{test}}), \qquad (M \text{ models})$$

Table B.2: **Effects of different preprocessing methods for numerical features for RealMLP-TD-S.** We report the relative increase in the shifted geometric mean benchmark scores compared to the standard method used in RealMLP-TD and RealMLP-TD-S, which is robust scaling and smooth clipping. We also report approximate 95% confidence intervals. To have a more common setting, we do not apply the preprocessing methods to one-hot encoded categorical features. In each column, the best score is highlighted in bold, and errors whose confidence interval contains the best score are underlined.

| Method | Error **increase** relative to robust scale + smooth clip in % | |
| --- | --- | --- |
| | meta-train-class | meta-train-reg |
| Robust scale + smooth clip | **0.0** [0.0, 0.0] | **0.0** [0.0, 0.0] |
| Robust scale | 0.5 [-0.4, 1.4] | 9.5 [4.4, 14.8] |
| Standardize + smooth clip | 1.6 [0.9, 2.2] | 1.2 [0.6, 1.8] |
| Standardize | 2.1 [1.2, 3.0] | 8.8 [3.9, 13.9] |
| Quantile transform (output dist. = normal) | 2.3 [1.5, 3.2] | 6.3 [5.5, 7.0] |
| Quantile transform (RTDL version) | 2.6 [1.5, 3.7] | 2.6 [0.4, 4.8] |
| KDI transform ($\alpha = 1$, output dist. = normal) | 4.9 [3.8, 6.0] | 4.4 [2.6, 6.2] |

Table B.3: **Improvements for LGBM-TD by bagging or (ensembled) refitting.** We perform 5-fold cross-validation, stratified for classification, and 5-fold refitting. We compare compare bagging vs. refitting, one model vs. five models, and individual stopping vs. joint stopping. The table shows the relative reduction in shifted geometric mean benchmark scores, including approximated 95% confidence intervals (Appendix C.6). In each column, the best score is highlighted in bold, and errors whose confidence interval contains the best score are underlined.

| Method | Error **reduction** relative to 1 fold in % | | | |
| --- | --- | --- | --- | --- |
| | meta-train-class | meta-test-class | meta-train-reg | meta-test-reg |
| LGBM-TD (bagging, 1 model, indiv. stopping) | -0.0 [-0.0, -0.0] | -0.0 [-0.0, -0.0] | -0.0 [-0.0, -0.0] | -0.0 [-0.0, -0.0] |
| LGBM-TD (bagging, 1 model, joint stopping) | -0.2 [-0.4, 0.1] | -0.7 [-1.3, -0.2] | 0.0 [-0.0, 0.0] | 0.3 [-0.2, 0.8] |
| LGBM-TD (bagging, 5 models, indiv. stopping) | 3.4 [3.0, 3.7] | 4.1 [3.6, 4.5] | **5.3** [4.5, 6.0] | 4.0 [3.6, 4.5] |
| LGBM-TD (bagging, 5 models, joint stopping) | 3.2 [2.8, 3.5] | 3.3 [2.9, 3.6] | 5.2 [4.5, 5.9] | 4.1 [3.7, 4.5] |
| LGBM-TD (refitting, 1 model, indiv. stopping) | 4.8 [4.1, 5.5] | 1.4 [-0.9, 3.6] | 3.8 [2.0, 5.5] | 4.0 [3.3, 4.8] |
| LGBM-TD (refitting, 1 model, joint stopping) | 5.0 [4.5, 5.5] | 4.3 [4.1, 4.6] | 3.7 [2.1, 5.3] | 4.1 [3.2, 4.9] |
| LGBM-TD (refitting, 5 models, indiv. stopping) | **5.6** [5.2, 6.1] | **6.0** [5.3, 6.7] | 5.2 [3.6, 6.7] | **5.5** [4.7, 6.4] |
| LGBM-TD (refitting, 5 models, joint stopping) | 5.4 [5.0, 5.9] | 5.9 [5.6, 6.1] | 5.2 [3.6, 6.7] | 5.5 [4.6, 6.3] |

$$
\tilde{\mathcal{D}}_i := \begin{cases} \mathcal{D} \setminus \mathcal{D}_i & \text{(bagging)} \\ \mathcal{D} & \text{(refitting),} \end{cases}
$$

$$
t_i^* := \begin{cases} \operatorname{argmin}_{t \in \{1,...,T\}} L_{\mathcal{D}_i}(f_{\mathcal{D} \setminus \mathcal{D}_i, t}) & \text{(indiv. stopping)} \\ \operatorname{argmin}_{t \in \{1,...,T\}} \sum_{j=1}^{5} L_{\mathcal{D}_j}(f_{\mathcal{D} \setminus \mathcal{D}_j, t}) & \text{(joint stopping).} \end{cases}
$$

Here, each model is trained with a different random seed. For LGBM, since we use an early stopping patience of 300 for each of the individual models, the argmin in the definition of $t_i^*$ can only go up to the minimum stopping iteration $T$ across the considered models.

The results of our experiments can be found in Table B.3 for LGBM-TD and in Table B.4 for RealMLP-TD. As expected, five models are considerably better than one. We find that refitting is mostly better than bagging, although a disadvantage of refitted models is that no validation scores are available, and it is unclear how HPO would affect this comparison. Comparing individual stopping to joint stopping, we find that individual stopping has a slight advantage in five-model bagging, while joint stopping performs better for single-model refitting. In the other two scenarios, joint stopping appears slightly better for RealMLP-TD and slightly worse for LGBM-TD. We also observe that the benefit of using five models instead of one appears to be larger for RealMLP-TD than for LGBM-TD.

## B.4 Early stopping for GBDTs

In Figure B.1 and Figure B.2, we study the influence of different early stopping patiences and metrics on the resulting benchmark performance of XGB-TD, LGBM-TD, and CatBoost-TD. While the regression results only deteriorate slightly for low patiences of 10 or 20 iterations, classification results are much more hurt by low patiences. In the classification setting, we evaluate the use of

Table B.4: **Improvements for RealMLP-TD by bagging or (ensembled) refitting.** We perform 5-fold cross-validation, stratified for classification, and 5-fold refitting. We compare bagging vs. refitting, one model vs. five models, and individual stopping vs. joint stopping. The table shows the relative reduction in shifted geometric mean benchmark scores, including approximated 95% confidence intervals (Appendix C.6). In each column, the best score is highlighted in bold, and errors whose confidence interval contains the best score are underlined.

| Method | Error **reduction** relative to 1 fold in % | | | |
| | meta-train-class | meta-test-class | meta-train-reg | meta-test-reg |
|---|---|---|---|---|
| RealMLP-TD (bagging, 1 model, indiv. stopping) | -0.0 [-0.0, -0.0] | -0.0 [-0.0, -0.0] | -0.0 [-0.0, -0.0] | -0.0 [-0.0, -0.0] |
| RealMLP-TD (bagging, 1 model, joint stopping) | 1.6 [0.9, 2.4] | 0.7 [0.0, 1.4] | 0.6 [0.1, 1.0] | -0.1 [-1.0, 0.7] |
| RealMLP-TD (bagging, 5 models, indiv. stopping) | 6.7 [6.1, 7.3] | 7.7 [6.9, 8.6] | 6.7 [6.2, 7.2] | 5.1 [4.0, 6.2] |
| RealMLP-TD (bagging, 5 models, joint stopping) | 6.7 [6.1, 7.4] | 7.3 [6.2, 8.3] | 6.7 [6.2, 7.2] | 4.8 [3.7, 5.8] |
| RealMLP-TD (refitting, 1 model, indiv. stopping) | 2.8 [1.7, 3.9] | 3.2 [1.8, 4.6] | 2.8 [1.7, 3.8] | 1.3 [-0.5, 3.0] |
| RealMLP-TD (refitting, 1 model, joint stopping) | 5.3 [4.5, 6.1] | 4.7 [3.9, 5.4] | 4.5 [3.5, 5.6] | 2.6 [0.9, 4.2] |
| RealMLP-TD (refitting, 5 models, indiv. stopping) | 7.6 [6.6, 8.5] | **8.8** [7.9, 9.6] | 8.5 [7.9, 9.1] | 5.3 [3.9, 6.7] |
| RealMLP-TD (refitting, 5 models, joint stopping) | **8.2** [7.5, 8.9] | 8.6 [7.9, 9.3] | **8.7** [8.0, 9.4] | **5.7** [4.5, 6.9] |

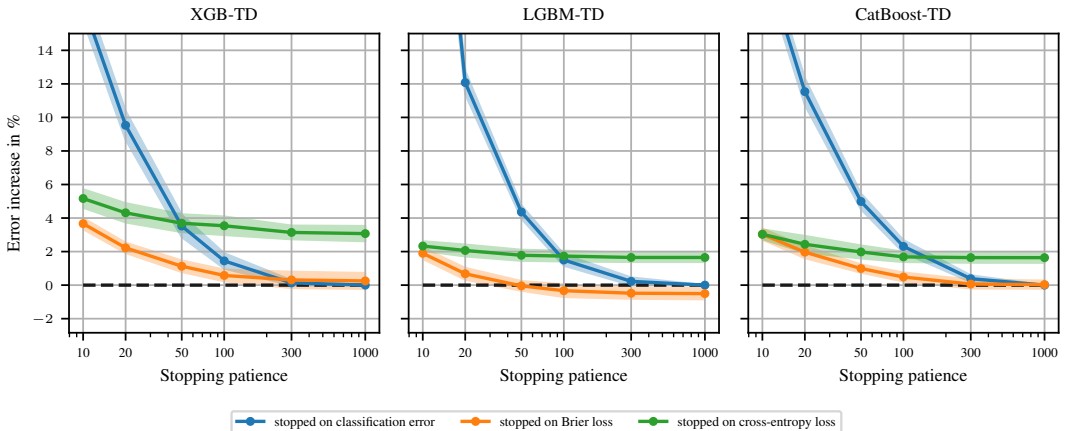

Figure B.1: **Effect of stopping patiences and metrics on the performance of GBDTs on** $\mathcal{B}_{\text{class}}^{\text{train}}$. We run the XGB-TD, LGBM-TD, and CatBoost-TD with different early stopping patiences (`early_stopping_rounds`). We compare three different metrics used for stopping and best-epoch selection: classification error, Brier loss, and cross-entropy loss. The $y$-axis reports the relative increase in the benchmark score relative to stopping on classification error with patience 1000 (i.e., never stopping early). The shaded areas are approximate 95% confidence intervals, cf. Appendix C.6.

different losses for early stopping and for best-epoch selection: classification error, Brier score, and cross-entropy loss. In each case, cross-entropy loss is used as the training loss, and classification error is used for evaluating the models on the test sets in the computation of the benchmark score. We observe that models stopped on classification error strongly deteriorate at low patiences ($\lesssim 100$), while our default patience of 300 achieves close-to-optimal results. Models stopped on cross-entropy loss deteriorate much less at low patiences, but achieve roughly 2% worse benchmark score at high patiences. Stopping on Brier loss achieves very good high-patience performance and is still only slightly more sensitive to the patience than stopping on cross-entropy loss. An interesting follow-up question would be if HPO can attenuate the differences between different settings.

## B.5 Results for AUROC

For classification, there are many different metrics to capture model performance. In the main paper, we use classification error to evaluate models. All TD configurations were tuned for classification error, early stopping and best-epoch selection were performed for classification error, and HPO was performed for classification error. Here, we evaluate models on the area under the ROC curve, also known as AUROC, AUC ROC, or AUC. For the multi-class case, we use the one-vs-rest formulation of AUC, which is faster to evaluate than one-vs-one. Higher AUC values are better and the optimal value is 1. Since we are interested in the shifted geometric mean error, we use $1 - \text{AUC}$ instead.

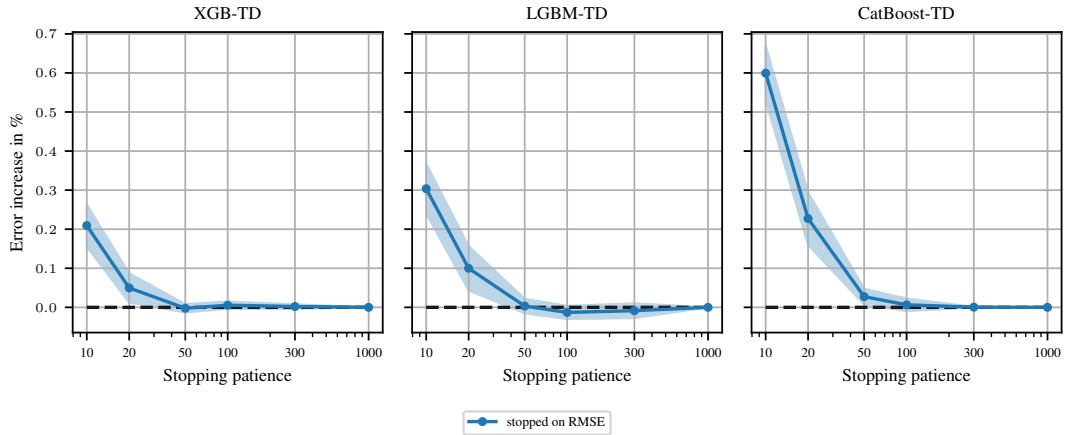

Figure B.2: **Effect of stopping patiences on the performance of GBDTs on $\mathcal{B}_{\text{reg}}^{\text{train}}$.** We run the TD configurations of XGB, LGBM, and CatBoost with different early stopping patiences (`early_stopping_rounds`). As in the remainder of the paper, we use RMSE for early stopping and best-epoch selection. The $y$-axis reports the relative increase in the benchmark score relative to stopping on classification error with patience 1000 (i.e., never stopping early). The shaded areas are approximate 95% confidence intervals, cf. Appendix C.6.

We compare two settings:

(1) A variant of the original setting where early stopping and the selection of the best epoch/iteration is based on accuracy but HPO is performed on $1 - \text{AUC}$. (Thanks to using random search, we do not have to re-run the HPO for this.)

(2) A setting where we use the cross-entropy loss for stopping and selecting the best epoch/iteration. While it would be possible to stop on AUC directly, this can be significantly slower since AUC is slower to evaluate. We do not perform HPO in this setting since it is expensive to run.

In both settings, we also evaluate RealMLP without label smoothing (no ls). Figure 3 shows the results optimized for accuracy and Figure B.3 shows the results optimized for cross-entropy. We make a few observations:

- Stopping for cross-entropy generally performs better than stopping for classification error.
- Label smoothing harms RealMLP for AUC, perhaps because the stopping metric does not use label smoothing, or because it encourages near-constant logits in areas where the model is relatively certain.
- Tuned defaults are mostly still better than the library defaults, except for XGBoost on $\mathcal{B}_{\text{class}}^{\text{test}}$.
- RealMLP without label smoothing is still competitive with GBDTs on the meta-test benchmark but does not perform better than GBDTs unlike what we observed for classification error.

### B.6 Results Without Missing-Value Datasets

To assess whether the results are influenced by our choices in missing value handling and exclusion, Figure B.4 presents results on all meta-test datasets that originally did not contain missing values. Only six meta-test datasets originally contain missing values: Three from $\mathcal{B}_{\text{class}}^{\text{test}}$ (kick, okcupid-stem, and porto-seguro) and three from $\mathcal{B}_{\text{reg}}^{\text{test}}$ (fps_benchmark, house_prices_nominal, SAT11-HAND-runtime-regression). While RealMLP deteriorates slightly, especially due to the exclusion of fps_benchmark, qualitative takeaways remain similar.

### B.7 Comparing Preprocessing Methods for NNs

In the other sections of this paper, we run each NN using the preprocessing from the respective paper that introduced it. Specifically, we use robust scaling and smooth clipping for RealMLP and the RTDL version of the quantile transform for the other papers (see also Appendix B.2). Here,

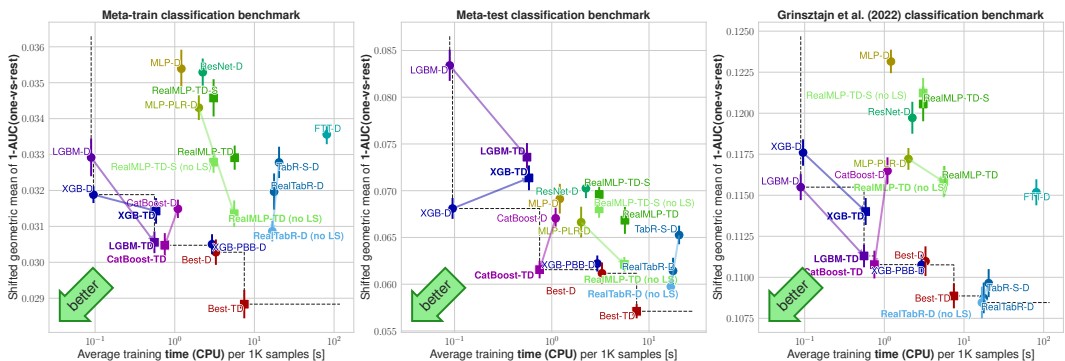

Figure B.3: **Benchmark scores on classification benchmarks vs. average training time for AUC, optimized for cross-entropy.** BestModel-TD uses RealMLP-TD without label smoothing. The $y$-axis shows the shifted geometric mean ($\text{SGM}_\varepsilon$) $1 - \text{AUC}$ as explained in Section 2.2. The $x$-axis shows average training times per 1000 samples (measured on $\mathcal{B}^{\text{train}}$ for efficiency reasons), see Appendix C.7. The error bars are approximate 95% confidence intervals for the limit #splits $\to \infty$, see Appendix C.6.

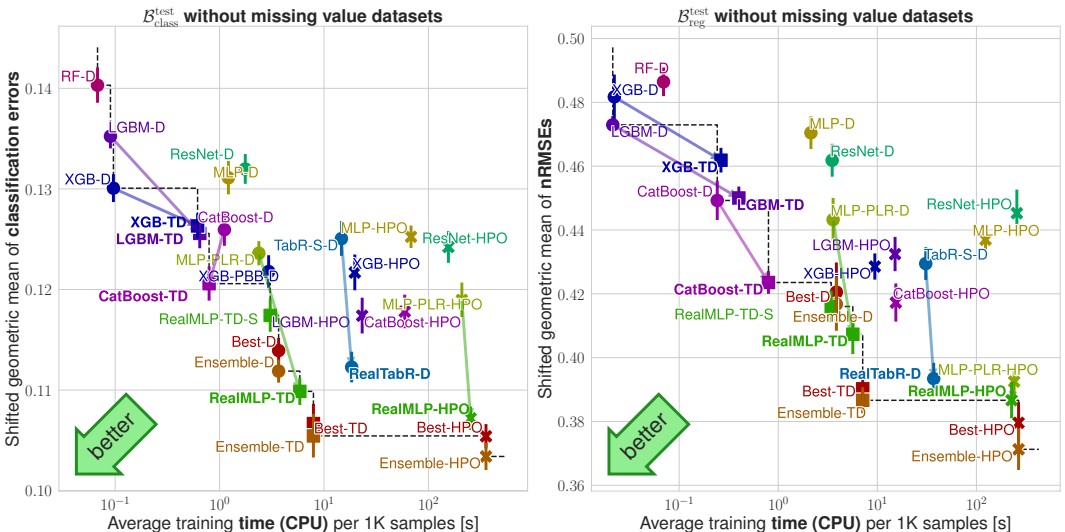

Figure B.4: **Benchmark scores on $\mathcal{B}_{\text{class}}^{\text{test}}$ and $\mathcal{B}_{\text{reg}}^{\text{test}}$ without missing value datasets vs. average training time.** The $y$-axis shows the shifted geometric mean ($\text{SGM}_\varepsilon$) classification error (left) or nRMSE (right) as explained in Section 2.2. The $x$-axis shows average training times per 1000 samples (measured on $\mathcal{B}^{\text{train}}$ for efficiency reasons), see Appendix C.7. The error bars are approximate 95% confidence intervals for the limit #splits $\to \infty$, see Appendix C.6.

we evaluate if robust scaling and smooth clipping can improve MLP, ResNet, MLP-PLR, FTT, and TabR-S as well. This also yields a more direct comparison of the architectures, although the nets still differ in other aspects such as initialization and regularization.

Figure B.5 includes results with robust scaling and smooth clipping (RS+SC) for MLP, ResNet, MLP-PLR, FTT, and TabR-S. While the results look promising for some methods (MLP, TabR) and not so promising for others (MLP-PLR), at least without re-tuning their default parameters, our results also show that trying both preprocessing methods can already give considerable improvements on most benchmarks.

## B.8 Results for Varying Architecture

Table B.5 shows the effects of including the preprocessing and architecture of RealMLP within other models. In particular, we study the benefits of our architectural changes, cf. Figure 1 (c),

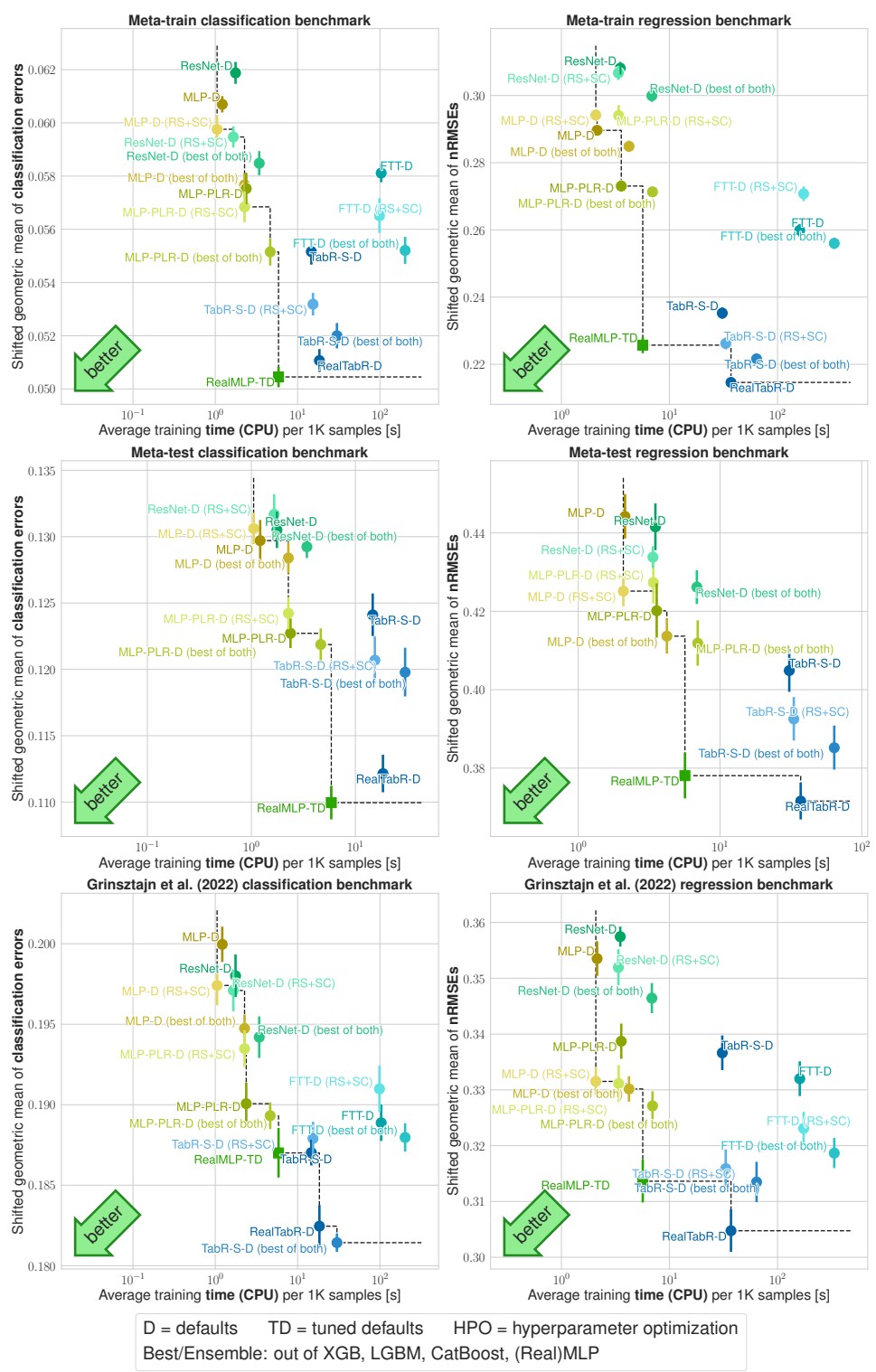

Figure B.5: **Benchmark scores on all benchmarks vs. average training time.** Compared to Figure 2, additional results for robust scale + smooth clip (RS+SC) preprocessing are included. The $y$-axis shows the shifted geometric mean ($\mathrm{SGM}_\varepsilon$) classification error (left) or nRMSE (right) as explained in Section 2.2. The $x$-axis shows average training times per 1000 samples (measured on $\mathcal{B}^{\mathrm{train}}$ for efficiency reasons), see Appendix C.7. The error bars are approximate 95% confidence intervals for the limit #splits $\to \infty$, see Appendix C.6.

Table B.5: **Comparison of preprocessing and architecture for different models.** We include variants with robust scaling and smooth clipping (RS+SC), as well as other modified aspects, cf. Appendix B.8. We report the relative decrease in the shifted geometric mean benchmark scores compared to MLP-D. We also report approximate 95% confidence intervals, cf. Appendix C.6.

| Method | Error **reduction** relative to MLP-D in % | | | |
| --- | --- | --- | --- | --- |
| | meta-train-class | meta-train-reg | meta-test-class | meta-test-reg |
| MLP-D | -0.0 [-0.0, -0.0] | -0.0 [-0.0, -0.0] | -0.0 [-0.0, -0.0] | -0.0 [-0.0, -0.0] |
| MLP-D (RS+SC) | 1.5 [0.7, 2.4] | -1.6 [-1.9, -1.2] | -0.7 [-1.6, 0.2] | 4.3 [3.4, 5.2] |
| MLP-D (RS+SC, no wd, meta-tuned lr) | 2.5 [1.8, 3.3] | -1.0 [-1.5, -0.5] | -1.6 [-2.7, -0.6] | 4.3 [3.3, 5.3] |
| MLP-D (RS+SC, no wd, meta-tuned lr, PL embeddings) | 4.6 [4.0, 5.2] | -1.5 [-1.9, -1.0] | -10.9 [-12.3, -9.4] | 5.4 [4.0, 6.9] |
| MLP-D (RS+SC, no wd, meta-tuned lr, RealMLP architecture) | 7.7 [6.9, 8.5] | 10.4 [9.4, 11.3] | 3.2 [2.0, 4.4] | 9.6 [8.6, 10.6] |
| RealMLP-TD-S | 12.6 [11.9, 13.2] | 13.8 [13.2, 14.4] | 9.8 [8.4, 11.2] | 13.2 [12.1, 14.3] |
| RealMLP-TD | **16.9** [16.1, 17.6] | **22.1** [21.2, 22.9] | **15.2** [14.0, 16.5] | **14.9** [14.0, 15.8] |
| TabR-S-D | 9.1 [8.2, 10.1] | 18.8 [18.3, 19.3] | 4.3 [3.0, 5.6] | 8.9 [7.9, 9.8] |
| TabR-S-D (RS+SC) | 12.4 [11.6, 13.1] | 21.9 [21.1, 22.7] | 7.0 [5.6, 8.3] | 11.6 [10.4, 12.8] |
| ResNet-D | -1.9 [-3.0, -0.9] | -6.4 [-7.0, -5.8] | -0.6 [-1.3, 0.1] | 0.6 [-0.4, 1.6] |
| ResNet-D (RS+SC) | 2.0 [1.3, 2.8] | -5.9 [-6.6, -5.2] | -1.5 [-2.7, -0.4] | 2.3 [1.4, 3.3] |

when applied directly to the setting of MLP-D. To this end, we approximately reproduce MLP-D in our codebase without weight decay (since the optimal value changes when including the NTP) and with marginally different early stopping thresholding logic. We also determine the best default learning rate on the meta-train benchmark, similar to Appendix A.4. Our reproduction achieves benchmark scores within 1% of the benchmark scores of the MLP-D (RS+SC) version. Adding the PL embeddings from Gorishniy et al. [16] with our default settings sometimes gives good results but is significantly worse on $\mathcal{B}^{\text{test}}_{\text{class}}$, indicating that they need more tuning. In contrast, incorporating the RealMLP architectural changes (including their associated learning rate factors) improves scores on all benchmarks by around 5% or more, although they alone do not match the results of TabR-S-D. However, the non-architectural changes in RealMLP-TD make an even larger difference.

## B.9  Comparing HPO Methods

In Figure B.6, we compare two different HPO methods for GBDTs:

- Random search (HPO), as used in the main paper, with 50 steps.
- Tree parzen estimator (HPO-TPE) as implemented in hyperopt [2], with 50 steps. The first 20 of these steps use random search.

While TPE often performs slightly better, the differences in benchmark scores are relatively small.

## B.10  More Time-Error Plots

Here, we provide more time-vs-error plots. Figure B.7 shows results for the arithmetic mean error, Figure B.8 shows results for the arithmetic mean rank, and Figure B.9 shows results for the arithmetic mean normalized error. For the normalized error, the scores are affinely rescaled on each dataset split such that the worst score is 1 and the best score is 0.

## B.11  Critical Difference Diagrams

Figure B.10 analyzes the external validity of differences in average ranks between methods, i.e., whether they will generalize to new datasets from a distribution. While establishing external validity requires a large number of datasets, our meta-test benchmarks show at least the improvements of RealMLP-TD over MLP-D to be externally valid.

## B.12  Win-rate Plots

For pairs of methods, we analyze the percentage of (dataset, split) combinations on which the first method has a lower error than the second method. We plot these win-rates in marix plots: Figure B.11 shows the results on $\mathcal{B}^{\text{train}}_{\text{class}}$, Figure B.12 shows the results on $\mathcal{B}^{\text{test}}_{\text{class}}$, Figure B.13 shows the results on $\mathcal{B}^{\text{Grinsztajn}}_{\text{class}}$, Figure B.14 shows the results on $\mathcal{B}^{\text{train}}_{\text{reg}}$, Figure B.15 shows the results on $\mathcal{B}^{\text{test}}_{\text{reg}}$, and Figure B.16 shows the results on $\mathcal{B}^{\text{Grinsztajn}}_{\text{reg}}$.

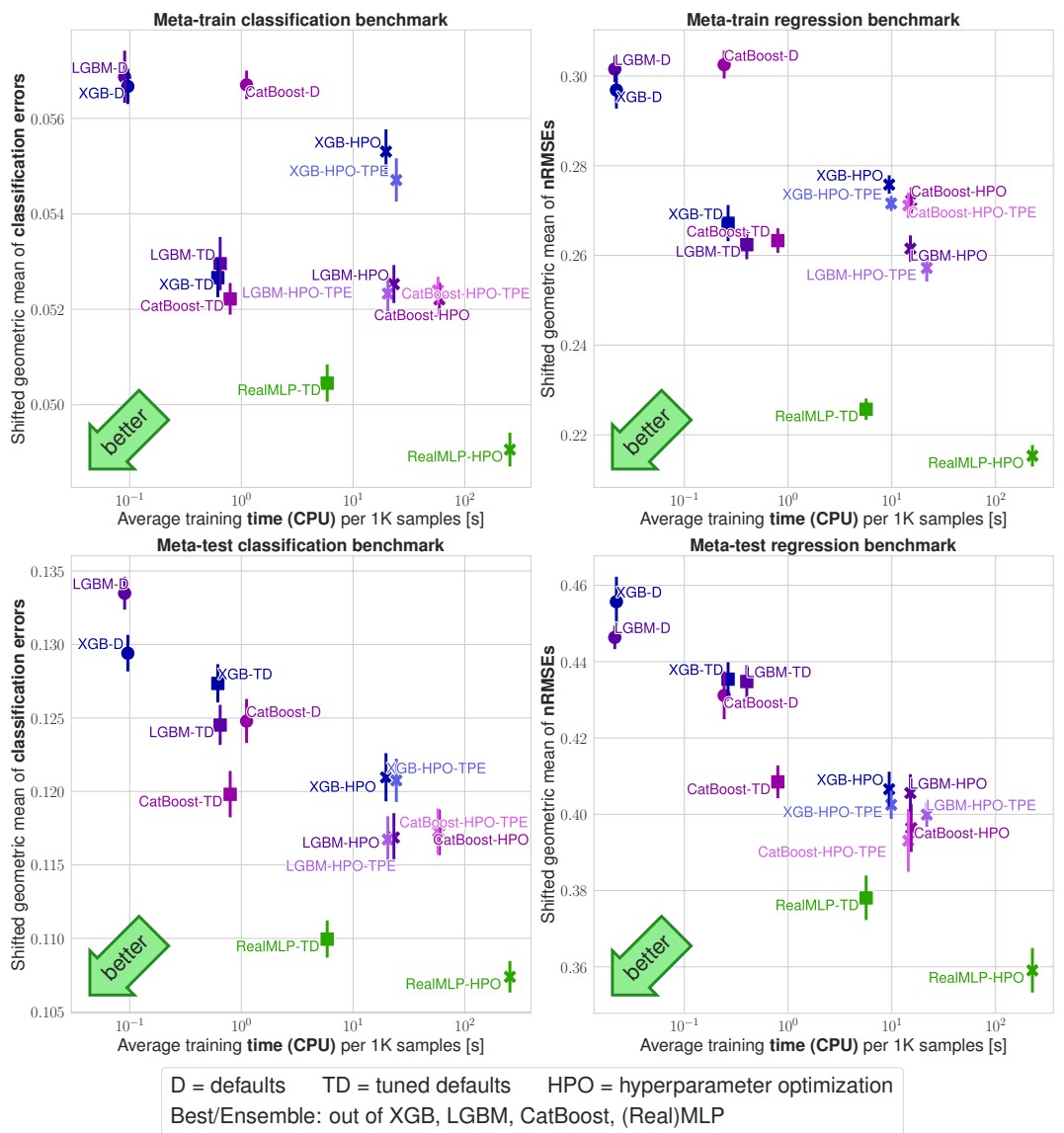

Figure B.6: **Benchmark scores of selected methods on $\mathcal{B}^{\text{train}}_{\text{class}}$, $\mathcal{B}^{\text{train}}_{\text{reg}}$, $\mathcal{B}^{\text{test}}_{\text{class}}$, and $\mathcal{B}^{\text{test}}_{\text{reg}}$ vs. average training time.** The $y$-axis shows the shifted geometric mean ($\text{SGM}_\varepsilon$) classification error (left) or nRMSE (right) as explained in Section 2.2. The $x$-axis shows average training times per 1000 samples (measured on $\mathcal{B}^{\text{train}}$ for efficiency reasons), see Appendix C.7. The error bars are approximate 95% confidence intervals for the limit #splits $\to \infty$, see Appendix C.6.

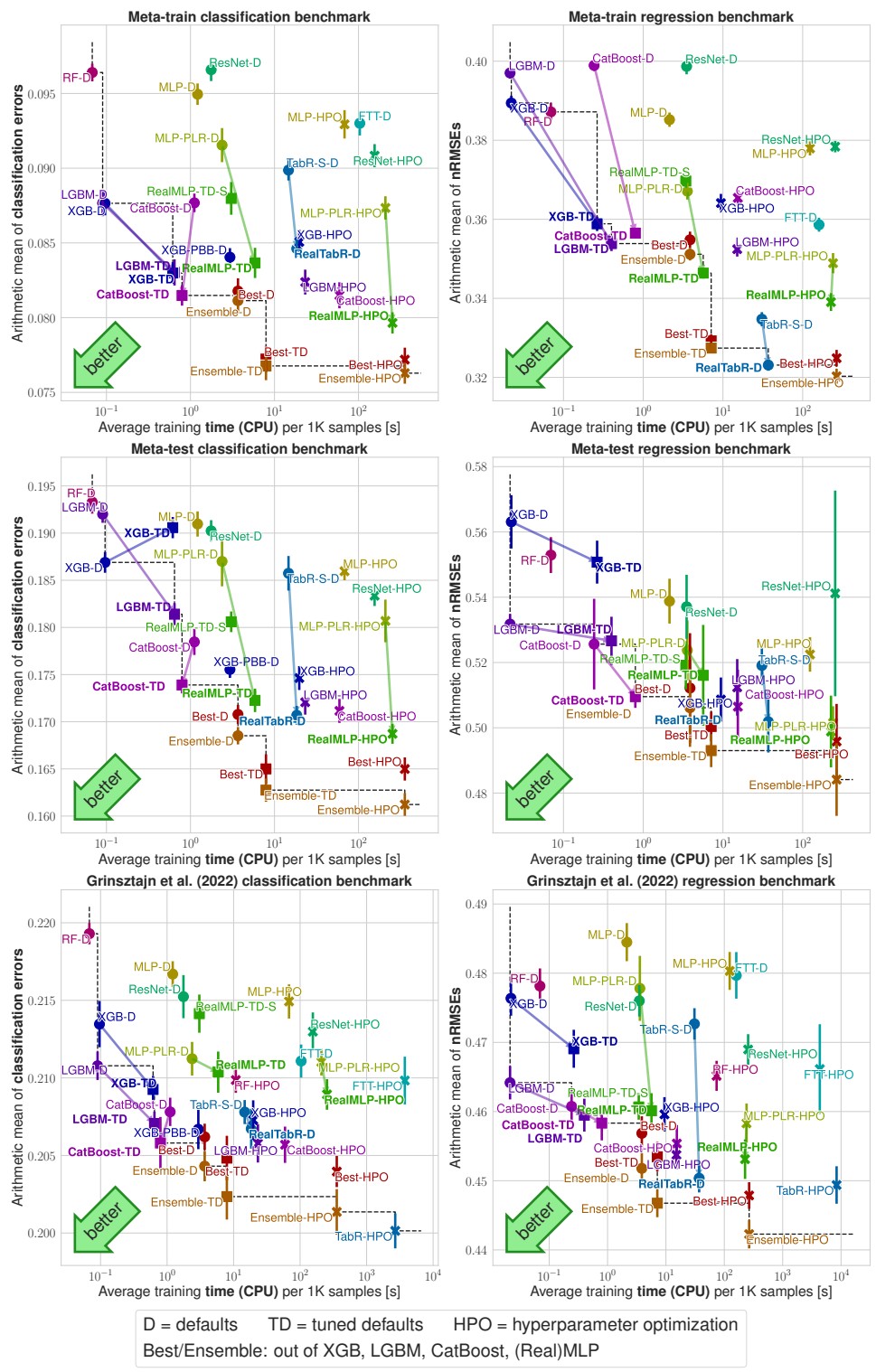

Figure B.7: **Benchmark scores (arithmetic mean) vs. average training time.** The $y$-axis shows the *arithmetic mean* classification error (left) or nRMSE (right). The $x$-axis shows average training times per 1000 samples (measured on $\mathcal{B}^{\text{train}}$ for efficiency reasons), see Appendix C.7. The error bars are approximate 95% confidence intervals for the limit #splits $\to \infty$, see Appendix C.6.

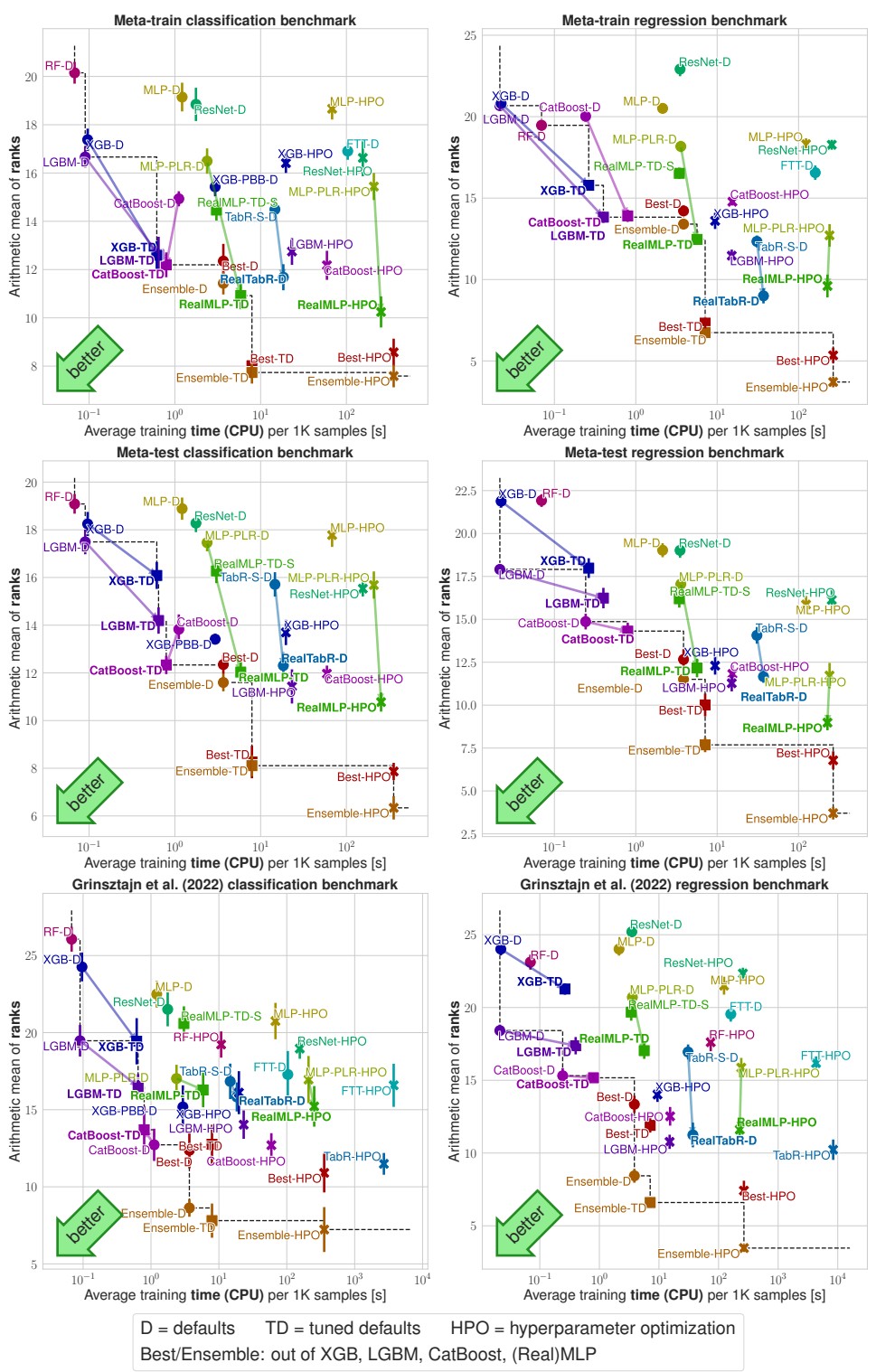

Figure B.8: **Benchmark scores (ranks) vs. average training time.** The $y$-axis shows the *arithmetic mean* rank, averaged over all splits and datasets. The $x$-axis shows average training times per 1000 samples (measured on $\mathcal{B}^{\text{train}}$ for efficiency reasons), see Appendix C.7. The error bars are approximate 95% confidence intervals for the limit #splits $\to \infty$, see Appendix C.6.

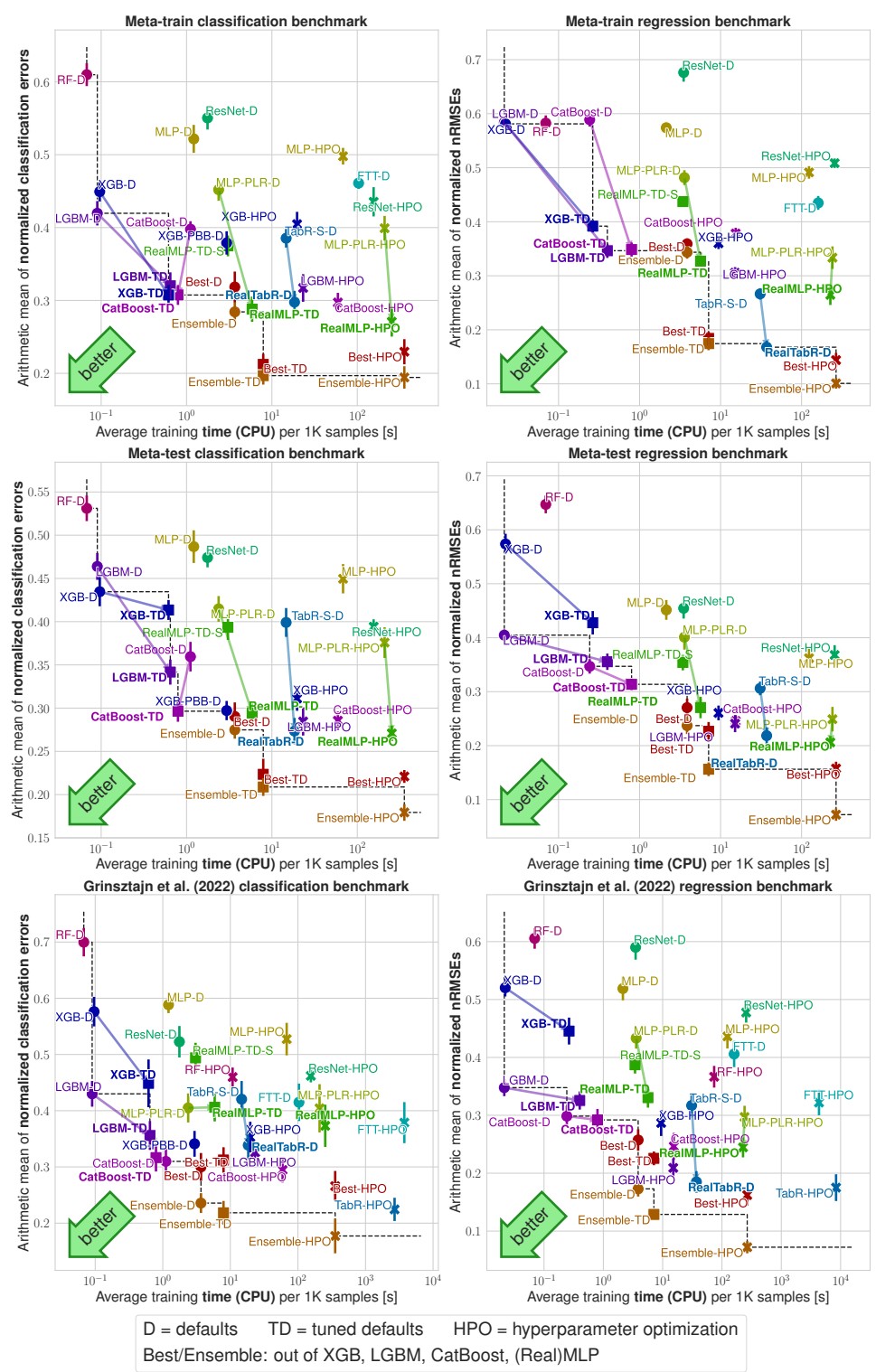

Figure B.9: **Benchmark scores (normalized errors) vs. average training time.** The $y$-axis shows the *arithmetic mean normalized* error, averaged over all splits and datasets. Errors are normalized by rescaling the lowest error to zero and the largest error to one. The $x$-axis shows average training times per 1000 samples (measured on $\mathcal{B}^{\text{train}}$ for efficiency reasons), see Appendix C.7. The error bars are approximate 95% confidence intervals for the limit #splits $\rightarrow \infty$, see Appendix C.6.

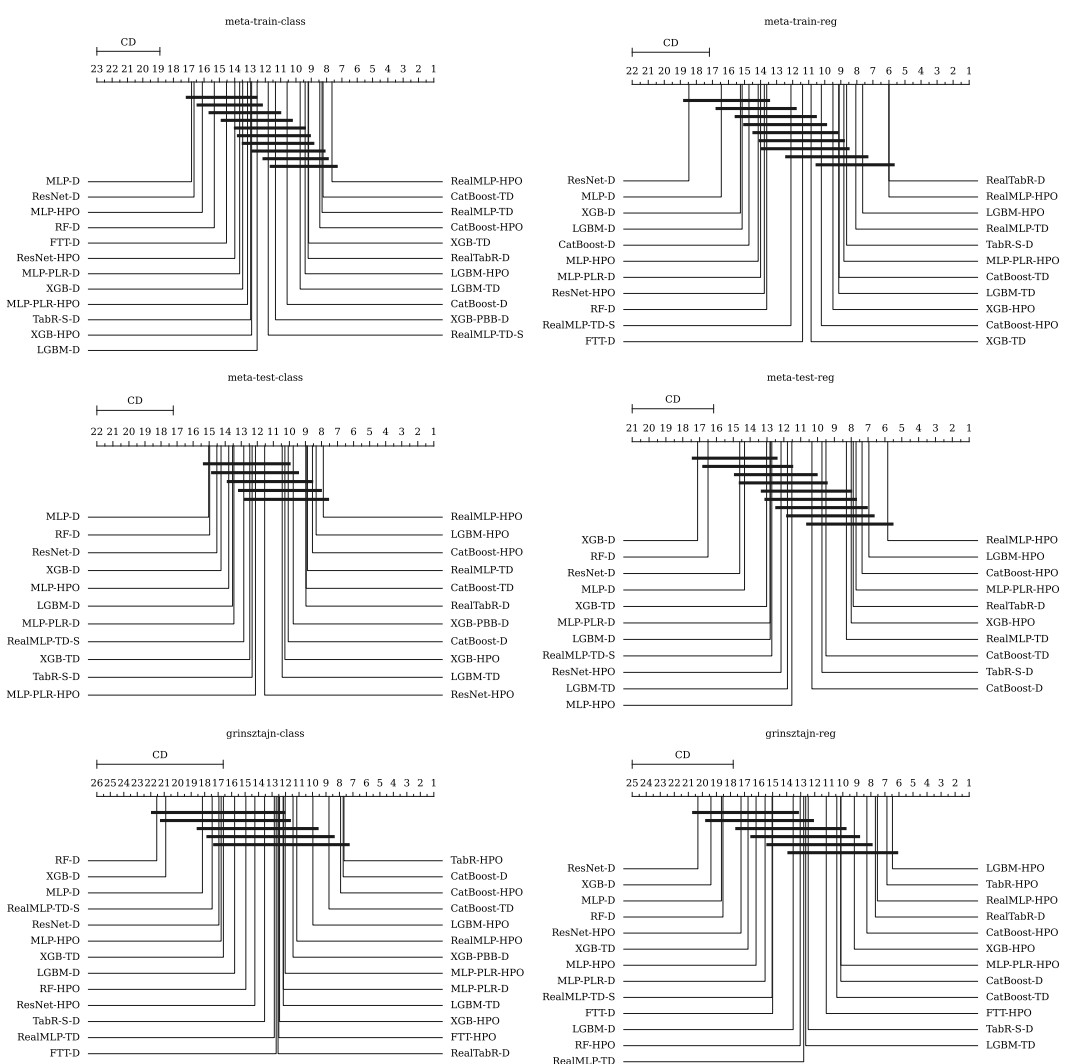

Figure B.10: **Critical difference diagrams on all benchmarks.** The plots show the average rank of methods on each benchmark. Horizontal bars indicate groups of algorithms that are not statistically significantly different at a 95% confidence level according to a Friedman test and post-hoc Nemenyi test implemented in `autorank` [22].

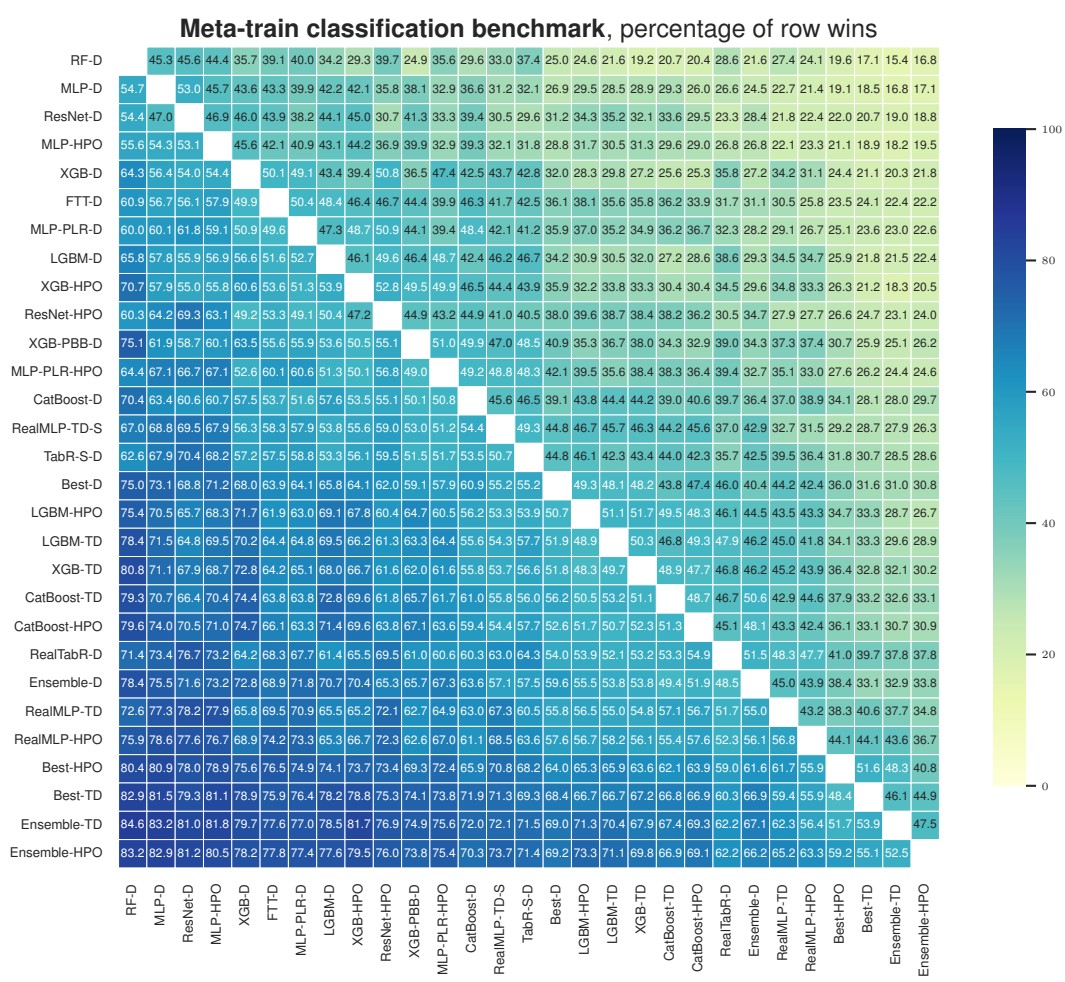

Figure B.11: **Percentages of wins of row algorithms vs column algorithms on** $\mathcal{B}_{\text{class}}^{\text{train}}$. Wins are averaged over all datasets and splits. Ties count as half-wins. Methods are sorted by average win-rate (i.e., the average of the values in the row). When averaging, we use dataset-dependent weighting as explained in Section C.3.1.

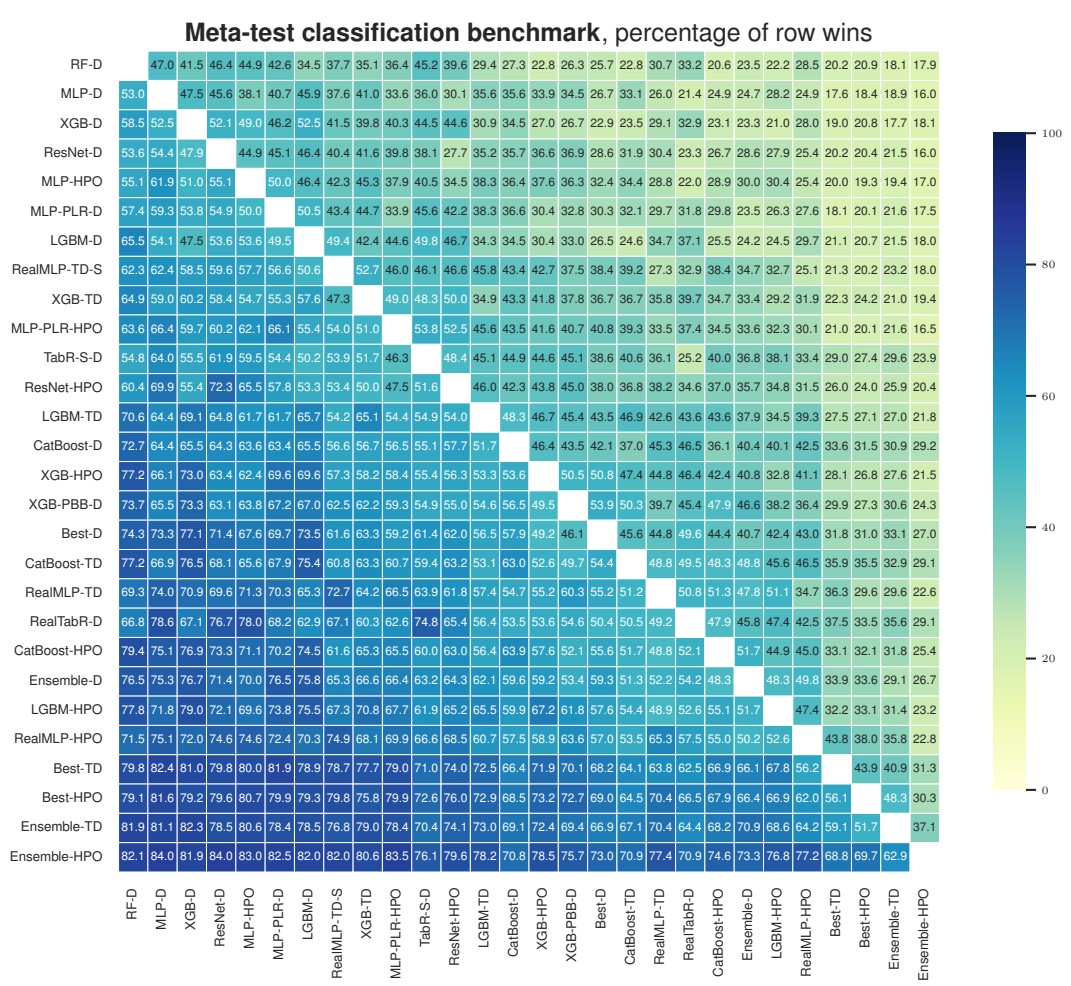

Figure B.12: **Percentages of wins of row algorithms vs column algorithms on $\mathcal{B}_{\text{class}}^{\text{test}}$.** Wins are averaged over all datasets and splits. Ties count as half-wins. Methods are sorted by average win-rate (i.e., the average of the values in the row).

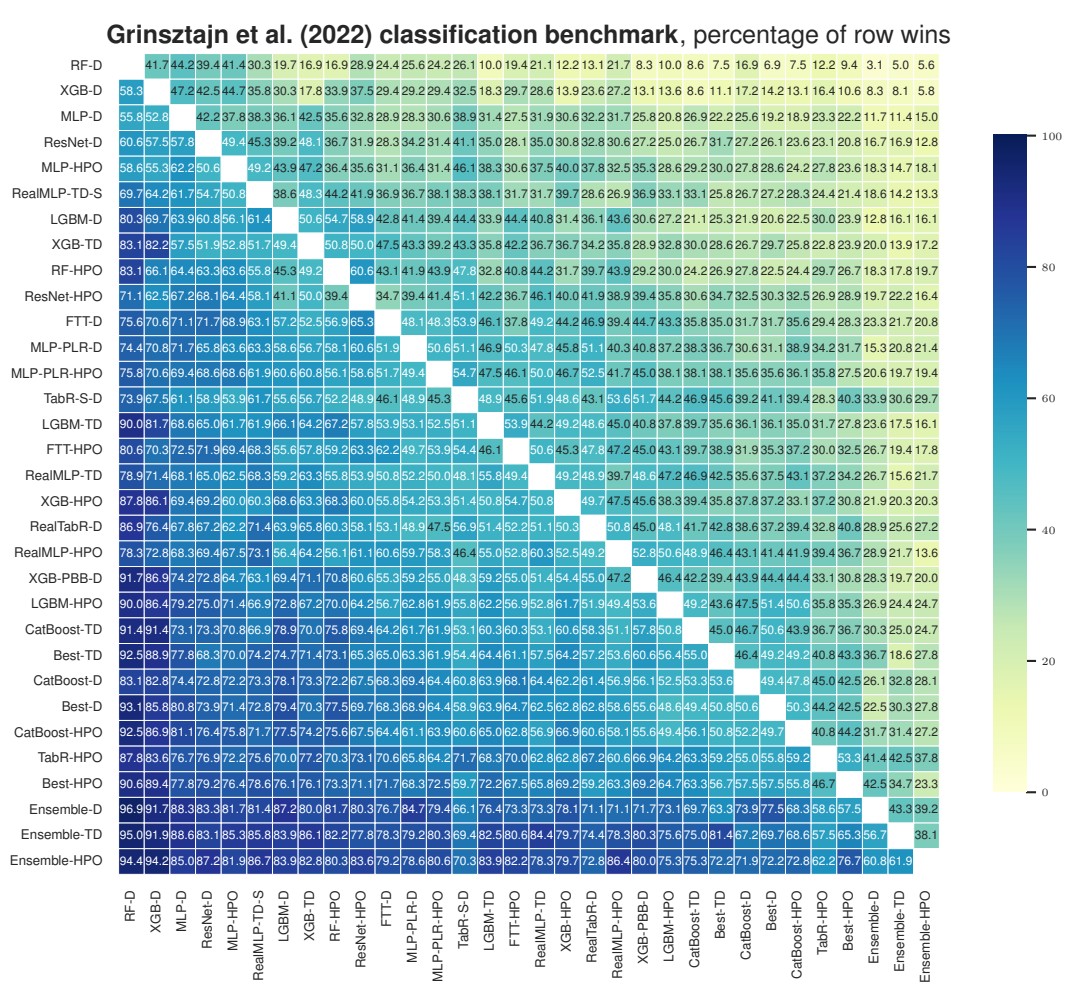

Figure B.13: **Percentages of wins of row algorithms vs column algorithms on** $\mathcal{B}_{\text{class}}^{\text{Grinsztajn}}$**.** Wins are averaged over all datasets and splits. Ties count as half-wins. Methods are sorted by average win-rate (i.e., the average of the values in the row). When averaging, we use dataset-dependent weighting as explained in Section C.3.1.

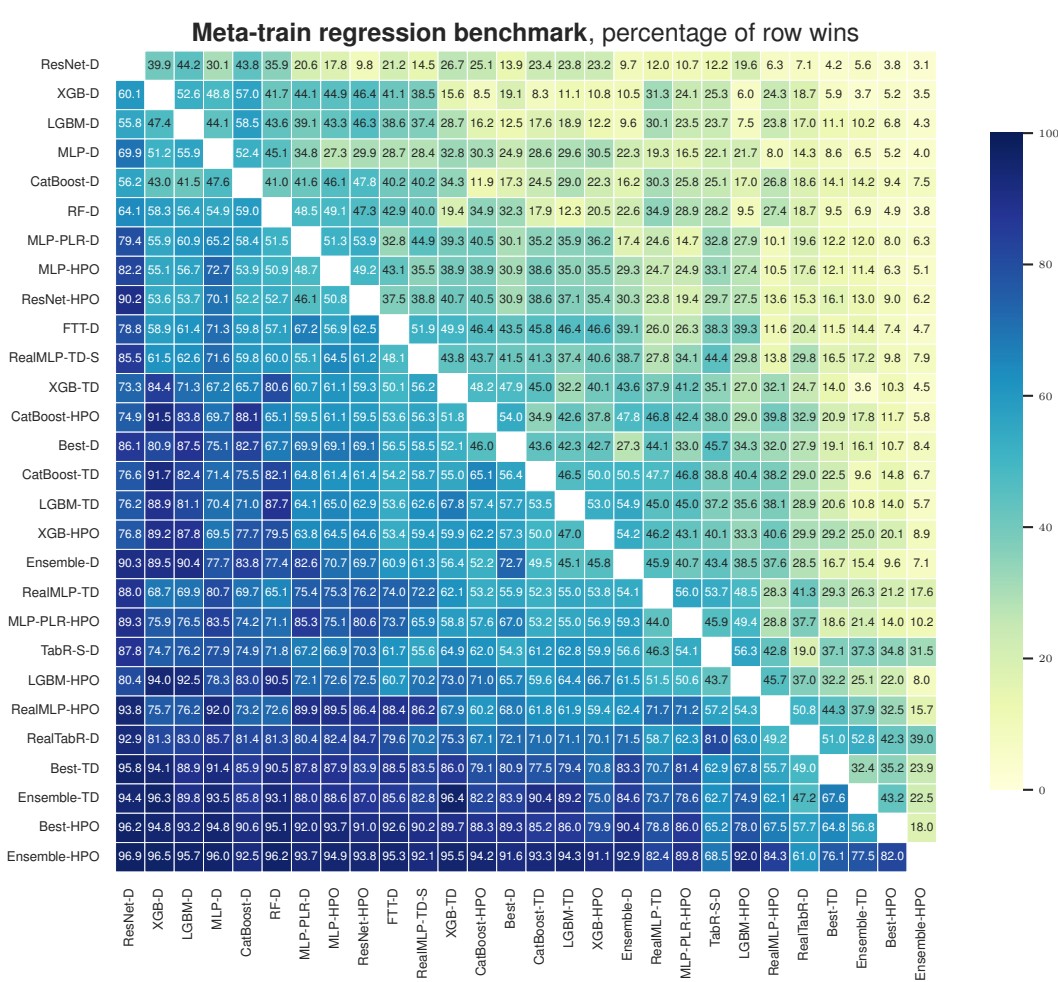

Figure B.14: **Percentages of wins of row algorithms vs column algorithms on $\mathcal{B}_{\mathrm{reg}}^{\mathrm{train}}$.** Wins are averaged over all datasets and splits. Ties count as half-wins. Methods are sorted by average win-rate (i.e., the average of the values in the row). When averaging, we use dataset-dependent weighting as explained in Section C.3.1.

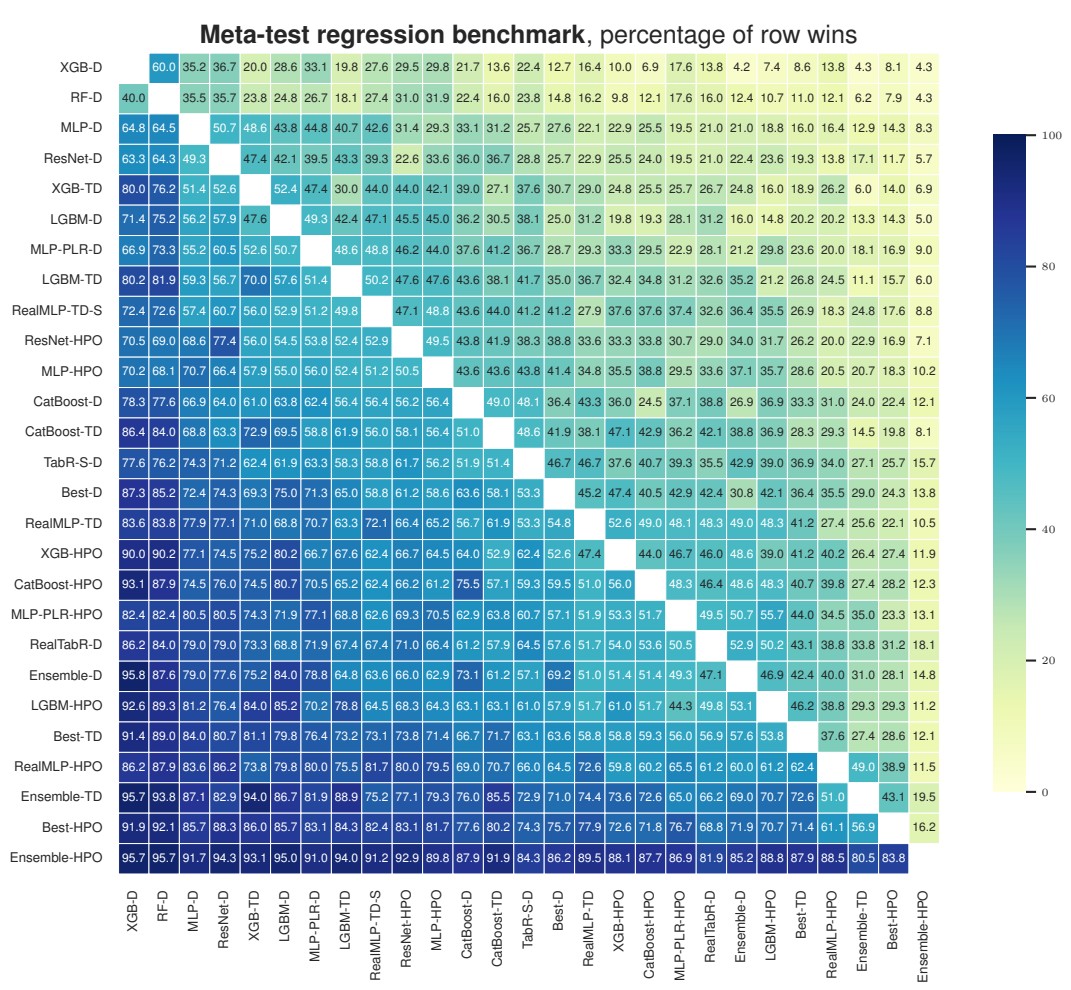

Figure B.15: **Percentages of wins of row algorithms vs column algorithms on $\mathcal{B}_{\text{reg}}^{\text{test}}$.** Wins are averaged over all datasets and splits. Ties count as half-wins. Methods are sorted by average win-rate (i.e., the average of the values in the row).

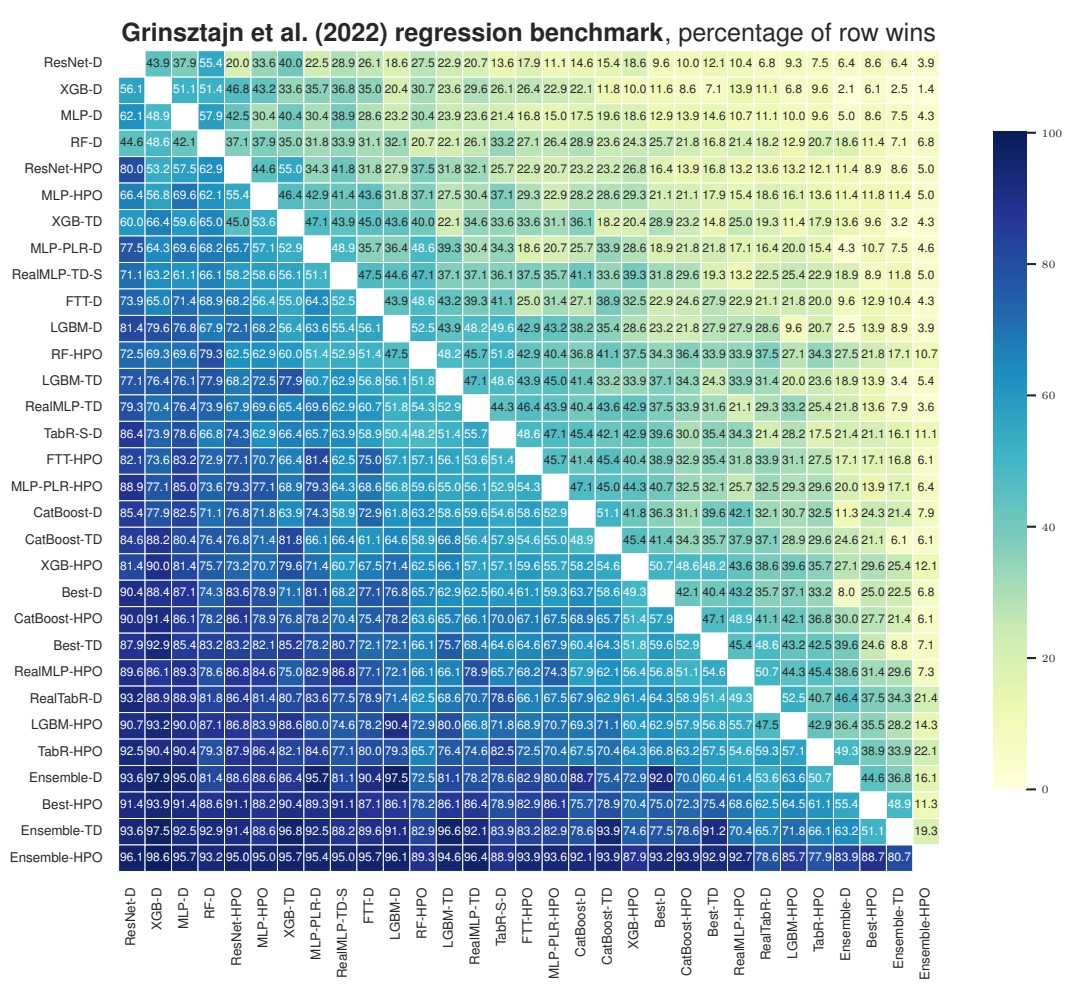

Figure B.16: **Percentages of wins of row algorithms vs column algorithms on $\mathcal{B}_{\text{reg}}^{\text{Grinsztajn}}$.** Wins are averaged over all datasets and splits. Ties count as half-wins. Methods are sorted by average win-rate (i.e., the average of the values in the row).

## C  Benchmark Details

### C.1  Default Configurations

The parameters for RealMLP-TD and RealMLP-TD-S have already been given in Table A.1. Table C.1 shows the hyperparameters of LGBM-TD and LGBM-D. Table C.2 shows the hyperparameters of XGB-TD and XGB-D. Table C.3 shows the hyperparameters of CatBoost-TD and CatBoost-D. The parameters for LGBM-D, XGB-D, and CatBoost-D have been taken from the respective libraries at the time of writing and are given here for completeness. We also provide tables for MLP-D (Table C.4), ResNet-D (Table C.6), MLP-PLR-D (Table C.5), FTT-D (Table C.7), TabR-S-D (Table C.8), and RealTabR-D (Table C.9). By "RTDL quantile transform", we refer to the version adding noise before fitting the quantile transform.

For XGB-PBB-D, we use the default parameters from Probst et al. [50], with the following modifications: We use `hist` gradient boosting since it is the new default in XGBoost 2.0. Moreover, since we have high-cardinality categories, we limit one-hot encoding to categories with less than 20 distinct values (not counting missing values) and use XGBoost's native categorical feature handling for the remaining categorical features. For RF-D, we use the default parameters from scikit-learn, do not give RF-D access to the validation set (to make it more similar to other methods that do not use nested cross-validation), and encode categorical columns using ordinal encoding with a random shuffling of categories.

### C.2  Hyperparameter Optimization

For all methods, we run 50 steps of random search with the search spaces presented in the following. The search spaces for LGBM-HPO (Table C.10), XGB-HPO (Table C.11), and CatBoost-HPO (Table C.12) are adapted from the "tree-friendly" literature, using `n_estimators=1000` in each case. The search space for RF-HPO (Table C.13) is taken from Grinsztajn et al. [18].

For RealMLP-HPO, we provide a custom search space specified in Table C.14. The search spaces for MLP-HPO (Table C.15), MLP-PLR-HPO (Table C.16), ResNet-HPO (Table C.17), FTT-HPO (Table C.18), and TabR-HPO (Table C.19) are adapted from the literature, with minor modifications to decrease RAM usage.

Table C.1: **Hyperparameters for LGBM-TD and LGBM-D.** Italic hyperparameters have not been tuned.

| Hyperparameter | LGBM-TD | | LGBM-D |
| --- | --- | --- | --- |
| | classif. | reg. | |
| num_leaves | 50 | 100 | 31 |
| learning_rate | 0.04 | 0.05 | 0.1 |
| subsample | 0.75 | 0.7 | 1.0 |
| colsample_bytree | 1.0 | 1.0 | 1.0 |
| min_data_in_leaf | 40 | 3 | 20 |
| min_sum_hessian_in_leaf | 1e-7 | 1e-7 | 1e-3 |
| *n_estimators* | *1000* | *1000* | *100* |
| *bagging_freq* | *1* | *1* | *1* |
| *max_bin* | *255* | *255* | *255* |
| *early_stopping_rounds* | *300* | *300* | *1000* |

Table C.2: **Hyperparameters for XGB-TD and XGB-D.** Italic hyperparameters have not been tuned for XGB-TD.

| Hyperparameter | XGB-TD | | XGB-D |
| --- | --- | --- | --- |
| | classif. | reg. | |
| max_depth | 6 | 9 | 6 |
| learning_rate | 0.08 | 0.05 | 0.3 |
| subsample | 0.65 | 0.7 | 1.0 |
| colsample_bytree | 1.0 | 1.0 | 1.0 |
| colsample_bylevel | 0.9 | 1.0 | 1.0 |
| min_child_weight | 5e-6 | 2.0 | 1.0 |
| lambda | 0.0 | 0.0 | 1.0 |
| *tree_method* | *hist* | *hist* | *hist* |
| *n_estimators* | *1000* | *1000* | *100* |
| *max_bin* | *256* | *256* | *256* |
| *early_stopping_rounds* | *300* | *300* | *1000* |

Table C.3: **Hyperparameters for CatBoost-TD and CatBoost-D.** Italic hyperparameters have not been tuned for CatBoost-TD.

| Hyperparameter | CatBoost-TD | | CatBoost-D |
| --- | --- | --- | --- |
| | classif. | reg. | |
| boosting_type | Plain | Plain | Plain |
| bootstrap_type | Bernoulli | Bernoulli | Bayesian |
| max_depth | 7 | 9 | 6 |
| learning_rate | 0.08 | 0.09 | automatic |
| subsample | 0.9 | 0.9 | — |
| bagging_temperature | — | — | 1.0 |
| l2_leaf_reg | 1e-5 | 1e-5 | 3.0 |
| random_strength | 0.8 | 0.0 | 1.0 |
| one_hot_max_size | 15 | 20 | 2 |
| leaf_estimation_iterations | 1 | 20 | None |
| *n_estimators* | *1000* | *1000* | *1000* |
| *max_bin* | *254* | *254* | *256* |
| *od_wait* | *300* | *300* | *None* |
| *od_type* | *Iter* | *Iter* | *Iter* |

Table C.4: Hyperparameters for MLP-D, adapted from McElfresh et al. [43].

| Hyperparameter | Value |
| --- | --- |
| lr scheduler | None |
| n_layers | 3 |
| d_layers | [128, 256, 128] |
| Dropout prob. | 0.1 |
| lr | 1e-3 |
| Optimizer | AdamW |
| d_embedding | 8 |
| batch_size | 128 |
| max_epochs | 1000 |
| early stopping patience | 20 |
| Preprocessing | RTDL quantile transform |
| Activation function | ReLU |
| Initialization | PyTorch default |
| Weight decay | 0.01 |

Table C.5: Hyperparameters for MLP-PLR-D. The MLP hyperparameters are taken from Table C.4 and the PLR embedding hyperparameters are taken as the defaults of the library associated with Gorishniy et al. [16].

| Hyperparameter | Value |
| --- | --- |
| MLP hyperparameters | same as in Table C.4 |
| Num. emb. type | PLR |
| Num. emb. initialization $\sigma$ | 1e-2 |
| Num. emb. #frequencies | 48 |
| Num. emb. dimension | 24 |

Table C.6: Hyperparameters for ResNet-D, adapted from McElfresh et al. [43].

| Hyperparameter | Value |
| --- | --- |
| lr scheduler | None |
| Activation | ReLU |
| Normalization | BatchNorm |
| n_layers | 2 |
| d_layers | [128, 128] |
| d_hidden_factor | 2 |
| hidden_dropout | 0.25 |
| residual_dropout | 0.1 |
| lr | 1e-3 |
| weight_decay | 0.01 |
| Optimizer | AdamW |
| d_embedding | 8 |
| batch_size | 128 |
| max_epochs | 1000 |
| early stopping patience | 20 |
| Preprocessing | RTDL quantile transform |

Table C.7: Hyperparameter search space for FTT-D, adapted from Gorishniy et al. [15]. Differences to Gorishniy et al. [15] are: We limit the number of epochs to 300 as in Grinsztajn et al. [18], we fix the batch size to 256 (Gorishniy et al. [15] use dataset-dependent batch sizes and Grinsztajn et al. [18] uses 512). We do not adopt the larger patience from Grinsztajn et al. [18].

| Hyperparameter | Value |
| --- | --- |
| n_layers | 3 |
| d_token | 192 |
| d_ffn_factor | 4/3 |
| ffn_dropout | 0.1 |
| attention_dropout | 0.2 |
| residual_dropout | 0.0 |
| lr | 1e-4 |
| weight_decay | 1e-5 |
| batch_size | 256 |
| max_epochs | 300 |
| early stopping patience | 16 |
| Preprocessing | RTDL quantile transform |
| n_heads | 8 |

Table C.8: Hyperparameters for TabR-S-D, taken from Gorishniy et al. [17]. The criterion on batch sizes is inferred to match the batch sizes used in the original paper.

| Hyperparameter | Value |
|---|---|
| num_embeddings | None |
| d_main | 265 |
| context_dropout | 0.38920071545944357 |
| d_multiplier | 2.0 |
| encoder_n_blocks | 0 |
| predictor_n_blocks | 1 |
| mixer_normalization | auto |
| dropout0 | 0.38852797479169876 |
| dropout1 | 0.0 |
| normalization | LayerNorm |
| activation | ReLU |
| batch_size | 128 if $N_{\text{train}} < 10$K else 256 if $N_{\text{train}} < 30$K else 512 if $N_{\text{train}} < 200$K else 1024 |
| patience | 16 |
| n_epochs | 100,000 |
| context_size | 96 |
| optimizer | AdamW |
| lr | 0.0003121273641315169 |
| weight_decay | 1.2260352006404615e-06 |
| Preprocessing | RTDL quantile transform |

Table C.9: Hyperparameters for RealTabR-D.

| Hyperparameter | Value |
|---|---|
| num_embeddings | PBLD |
| num. emb. #frequencies | 8 |
| num. emb. d_embedding | 4 |
| num. emb. frequency_scale | 0.1 |
| Preprocessing | robust scale + smooth clip |
| Add scaling layer | yes |
| Scaling layer lr factor | 96 |
| Label smoothing epsilon | 0.1 (for classification) |
| Other hyperparameters | as in Table C.8 |

Table C.10: Hyperparameter seach space for LGBM-HPO, adapted from Prokhorenkova et al. [51] with 1000 estimators instead of 5000.

| Hyperparameter | Space |
|---|---|
| n_estimators | 1000 |
| bagging_freq | 1 |
| early_stopping_rounds | 300 |
| num_leaves | LogUniformInt$[1, e^7]$ |
| learning_rate | LogUniform$[e^{-7}, 1]$ |
| subsample | Uniform$[0.5, 1]$ |
| feature_fraction | Uniform$[0.5, 1]$ |
| min_data_in_leaf | LogUniformInt$[1, e^6]$ |
| min_sum_hessian_in_leaf | LogUniform$[e^{-16}, e^5]$ |
| lambda_l1 | Random$\{0, \text{LogUniform}[e^{-16}, e^2]\}$ |
| lambda_l2 | Random$\{0, \text{LogUniform}[e^{-16}, e^2]\}$ |

Table C.11: Hyperparameter search space for XGB-HPO, adapted from Grinsztajn et al. [18]. We use the `hist` method, which is the new default in XGBoost 2.0 and supports native handling of categorical values, while the old `auto` method selection is not available in XGBoost 2.0. We also increase `early_stopping_rounds` to 300.

| Hyperparameter | Space |
|---|---|
| tree_method | hist |
| n_estimators | 1000 |
| early_stopping_rounds | 300 |
| max_depth | UniformInt[1, 11] |
| learning_rate | LogUniform[1e-5, 0.7] |
| subsample | Uniform[0.5, 1] |
| colsample_bytree | Uniform[0.5, 1] |
| colsample_bylevel | Uniform[0.5, 1] |
| min_child_weight | LogUniformInt[1, 100] |
| alpha | LogUniform[1e-8, 1e-2] |
| lambda | LogUniform[1, 4] |
| gamma | LogUniform[1e-8, 7.0] |

Table C.12: Hyperparameter search space for CatBoost-HPO, adapted from Shwartz-Ziv and Armon [58], who did not specify the number of estimators.

| Hyperparameter | Space |
|---|---|
| boosting_type | Plain |
| bootstrap_type | Bayesian |
| n_estimators | 1000 |
| max_depth | 6 |
| od_wait | 300 |
| od_type | Iter |
| learning_rate | LogUniform[$e^{-5}$, 1] |
| bagging_temperature | Uniform[0, 1] |
| l2_leaf_reg | LogUniform[1, 10] |
| random_strength | UniformInt[1, 20] |
| one_hot_max_size | UniformInt[0, 25] |
| leaf_estimation_iterations | UniformInt[1, 20] |

Table C.13: Hyperparameter search space for RF-HPO, taken from Grinsztajn et al. [18].

| Hyperparameter | Space |
|---|---|
| n_estimators | 250 |
| max_depth | Choice([None, 2, 3, 4], p=[0.7, 0.1, 0.1, 0.1]) |
| criterion | Choice([gini, entropy]) if classification else Choice([squared_error, absolute_error]) |
| max_features | Choice([sqrt, sqrt, log2, None, 0.1, 0.2, 0.3, 0.4, 0.5, 0.6, 0.7, 0.8, 0.9]) |
| min_samples_split | Choice([2, 3], p=[0.95, 0.05]) |
| min_samples_leaf | LogUniformInt[1.5, 50.5] |
| bootstrap | Choice(True, False) |
| min_impurity_decrease | Choice([0, 0.01, 0.02, 0.05], p=[0.85, 0.05, 0.05, 0.05]) |

Table C.14: Hyperparameter search space for RealMLP-HPO. The remaining hyperparameters are set as in RealMLP-TD. For best performance, it might be beneficial to use a larger search space for the init standard deviation of the first embedding layer, and to tune the embedding dimensions, as in Table C.16.

| Hyperparameter | classif. | reg. |
|---|---|---|
| Num. embedding type | Choice([None, PBLD, PL, PLR]) | same |
| Use scaling layer | Choice([True, False], p=[0.6, 0.4]) | same |
| Learning rate | LogUniform([2e-2, 3e-1]) | same |
| Dropout prob. | Choice([0.0, 0.15, 0.3], p=[0.3, 0.5, 0.2]) | same |
| Activation fct. | Choice([ReLU, SELU, Mish]) | same |
| Hidden layer sizes | Choice([[256, 256, 256], [64, 64, 64, 64, 64], [512]], p=[0.6, 0.2, 0.2]) | same |
| Weight decay | Choice([0.0, 2e-2]) | same |
| $\boldsymbol{w}_{\mathrm{emb}}^{(1,i)}$ init std. | LogUniform([0.05, 0.5]) | |
| Label smoothing $\varepsilon$ | Choice([0.0, 0.1], p=[0.3, 0.7]) | no label smoothing |

Table C.15: Hyperparameter search space for MLP-HPO, adapted from Gorishniy et al. [15]. We reduced the embedding dimension upper bound, and the maximum number of epochs to have a more acceptable runtime on the meta-test benchmarks. As in the original paper, the size of the first and the last layers are tuned and set separately, while the size for "in-between" layers is the same for all of them.

| Hyperparameter | Space | |
|---|---|---|
| | $N \leq 100,000$ | $N > 100,000$ |
| n_layers | UniformInt[1, 8] | UniformInt[1, 16] |
| d_hidden_layers | UniformInt[1, 512] | UniformInt[1, 1024] |
| d_first_layer | UniformInt[1, 512] | UniformInt[1, 1024] |
| d_last_layer | UniformInt[1, 512] | UniformInt[1, 1024] |
| dropout | Choice(0, Uniform[0, 0.5]) | |
| lr | LogUniform[1e-5, 1e-2] | |
| weight decay | Choice(0, LogUniform[1e-6, 1e-3]) | |
| d_embedding | UniformInt[1, 64] | |
| batch_size | 128 if $N_{\mathrm{train}} < 10K$ else 256 if $N_{\mathrm{train}} < 30K$ else 512 if $N_{\mathrm{train}} < 100K$ else 1024 | |
| lr_scheduler | None | |
| Optimizer | AdamW | |
| max #epochs | 400 | |
| early stopping patience | 16 | |
| Preprocessing | RTDL quantile transform | |

Table C.16: Hyperparameter search space for MLP-PLR-HPO, adapted from Gorishniy et al. [16]. Differences to Gorishniy et al. [16] are: (1) For the MLP part of the search space, we use the same space as for MLP, which includes categorical embeddings and slightly different ranges for some hyperparameters. (2) We shrank the search space for $\sigma$, as recommended by one of the authors in private communication. (3) We reduced the maximum embedding dimension from 128 to 64 to avoid RAM issues on datasets with many numerical features.

| Hyperparameter | Space |
|---|---|
| MLP hyperparameters | as in Table C.15 |
| Num. emb. type | PLR |
| Num. emb. initialization $\sigma$ | LogUniform[1e-2, 1e1] |
| Num. emb. #frequencies | Uniform[1, 64] |
| Num. emb. dimension | Uniform[1, 64] |

Table C.17: Hyperparameter search space for ResNet-HPO, adapted from Gorishniy et al. [15]. We reduced the embedding dimension upper bound, the maximum number of epochs, and the number of layers to have a more acceptable runtime on the meta-test benchmarks. As in the original paper, the size of the first and the last layers are tuned and set separately, while the size for "in-between" layers is the same for all of them.

| Hyperparameter | Space | |
| --- | --- | --- |
| | $N \leq 100,000$ | $N > 100,000$ |
| n_layers | UniformInt[1, 8] | UniformInt[1, 16] |
| d_hidden_layers | UniformInt[1, 512] | UniformInt[1, 1024] |
| d_hidden_factor | UniformInt[1, 4] | |
| hidden_dropout | Uniform[0, 0.5] | |
| residual_dropout | Choice(0, Uniform[0, 0.5]) | |
| lr | LogUniform[1e-5, 1e-2] | |
| weight decay | Choice(0, LogUniform[1e-6, 1e-3]) | |
| d_embedding | UniformInt[1, 64] | |
| batch_size | 128 if $N_{\text{train}} < 10K$ else 256 if $N_{\text{train}} < 30K$ else 512 if $N_{\text{train}} < 100K$ else 1024 | |
| activation | ReLU | |
| normalization | BatchNorm | |
| lr_scheduler | None | |
| Optimizer | AdamW | |
| max #epochs | 400 | |
| early stopping patience | 16 | |
| Preprocessing | RTDL quantile transform | |

Table C.18: Hyperparameter search space for FTT-HPO, adapted from Gorishniy et al. [17]. Differences to Gorishniy et al. [17] are: We limit the number of epochs to 400, and the batch size choices might differ slightly since the criterion in Gorishniy et al. [17] is unclear to us.

| Hyperparameter | Space |
| --- | --- |
| n_layers | UniformInt[1, 4] |
| d_token | $8 \cdot$ UniformInt[2, 48] |
| d_ffn_factor | Uniform[2/3, 8/3] |
| ffn_dropout | Uniform[0, 0.5] |
| attention_dropout | Uniform[0, 0.5] |
| residual_dropout | Choice(0, Uniform[0, 0.2]) |
| lr | LogUniform[1e-5, 1e-3] |
| weight_decay | Choice(0, LogUniform[1e-6, 1e-4]) |
| batch_size | 128 if $N_{\text{train}} < 10K$ else 256 if $N_{\text{train}} < 30K$ else 512 if $N_{\text{train}} < 100K$ else 1024 |
| max_epochs | 400 |
| early stopping patience | 16 |
| Preprocessing | RTDL quantile transform |
| n_heads | 8 |

Table C.19: Hyperparameter search space for TabR-HPO, taken from Gorishniy et al. [17]. Non-specified hyperparameters are chosen as in TabR-S-D (Table C.8). For the weight decay, we used an upper bound of 1e-4 as used in the original code, and not 1e-3 as specified in the paper.

| Hyperparameter | Space |
|---|---|
| d_main | UniformInt[96, 384] |
| context_dropout | Uniform[0.0, 0.6] |
| dropout0 | Uniform[0.0, 0.6] |
| dropout1 | 0.0 |
| lr | LogUniform[3e-5, 1e-3] |
| weight_decay | Choice(0, LogUniform[1e-6, 1e-4]) |
| encoder_n_blocks | UniformInt[0, 1] |
| predictor_n_blocks | UniformInt[1, 2] |
| num. emb. type | PLR |
| num. emb. n_frequencies | UniformInt[16, 96] |
| num. emb. d_embedding | UniformInt[16, 65] |
| num. emb. frequency_scale | LogUniform[1e-2, 1e2] |
| num. emb. lite | True |

### C.3 Dataset Selection and Preprocessing

#### C.3.1 Meta-train Benchmarks

For the meta-train benchmarks, we adapt code from Steinwart [61] to collect all datasets from the UCI repository that follow certain criteria:

- Between 2,500 and 50,000 samples.
- Number of features at most 1,000.
- Labeled as classification or regression task.
- Description made it straightforward to convert the original dataset into a numeric .csv format.
- Uploaded before 2019-05-08.

We remove rows with missing values and keep only those datasets that still have at least 2,500 samples.[3] Some datasets are labeled both as regression and classification datasets, in which case we use them for both. Some datasets contain different versions (e.g., different target columns), in which case we use all of them. To avoid biasing the results towards one dataset, we compute benchmark scores using weights proportional to $1/\#$versions. In total, we obtain 71 classification datasets (including versions) out of 46 original datasets, and 47 regression datasets (including versions) out of 26 original datasets. Tables C.20 and C.21 summarize key characteristics of these datasets. We count datasets with the same prefix (before the first underscore) as being versions of the same dataset for weighting, except for the two "facebook" datasets in $\mathcal{B}_{\mathrm{reg}}^{\mathrm{train}}$, which we count as distinct because they are taken from different sources. For regression, we standardize the targets to have mean zero and variance 1 on the whole dataset. This does not introduce leakage since all neural networks standardize regression targets based on the training set, and tree-based methods are invariant to affine rescaling.

During earlier development of the MLP, the meta-train benchmark used to include an epileptic seizure recognition dataset, which has since been removed from the UCI repository, hence we do not report results on it.

#### C.3.2 Meta-test Benchmarks

The meta-test benchmarks consist of datasets from the AutoML Benchmark [13] and additional regression datasets from the OpenML-CTR23 benchmark [12], obtained from OpenML [65].

We make the following modifications:

- We use brazilian_houses from OpenML-CTR23 and exclude Brazilian_houses from the AutoML regression benchmark, since the latter contains three additional features that should not be used for predicting the target.
- We use another version of the sarcos dataset where the original test set is not included, since the original test set consists of duplicates of training samples.
- We excluded the following datasets because versions of them were already contained in the meta-training set:
  - For classification: kr-vs-kp, wilt, ozone-level-8hr, first-order-theorem-proving, GesturePhaseSegmentationProcessed, PhishingWebsites, wine-quality-white, nomao, bank-marketing, adult
  - For regression: wine_quality, abalone, OnlineNewsPopularity, Brazilian_houses, physicochemical_protein, naval_propulsion_plant, superconductivity, white_wine, red_wine, grid_stability

We preprocess the datasets as follows:

- We remove rows with missing continuous values
- We subsample large datasets to contain at most 500,000 samples. Since the dionis dataset was particularly slow to train with GBDT models due to its 355 classes, we subsampled it to 100,000 samples.
- We encode missing categorical values as a separate category.
- For regression, we standardize the targets to have mean zero and variance 1. This does not introduce leakage since all neural networks standardize regression targets based on the training set, and tree-based methods are invariant to affine rescaling.

---

[3]We noticed later that the ozone_level_1hr and ozone_level_8hr datasets contain less than 2,500 samples, but we decided to keep them since we already used them for tuning the hyperparameters.

Table C.20: Datasets in the meta-train classification benchmark.

| Name | #samples | #num. features | #cat. features | largest #categories | #classes |
|---|---|---|---|---|---|
| abalone | 4177 | 8 | 0 | | 3 |
| adult | 45222 | 7 | 7 | 41 | 2 |
| anuran_calls_families | 7127 | 22 | 0 | | 3 |
| anuran_calls_genus | 6073 | 22 | 0 | | 5 |
| anuran_calls_species | 5696 | 22 | 0 | | 7 |
| avila | 20867 | 10 | 0 | | 12 |
| bank_marketing | 41579 | 12 | 5 | 11 | 2 |
| bank_marketing_additional | 39457 | 19 | 3 | 11 | 2 |
| chess | 3196 | 1 | 31 | 3 | 2 |
| chess_krvk | 28056 | 3 | 3 | 8 | 18 |
| crowd_sourced_mapping | 10494 | 28 | 0 | | 4 |
| default_credit_card | 30000 | 23 | 1 | 2 | 2 |
| eeg_eye_state | 14980 | 14 | 0 | | 2 |
| electrical_grid_stability_simulated | 10000 | 12 | 0 | | 2 |
| facebook_live_sellers_thailand_status | 6622 | 9 | 0 | | 2 |
| firm_teacher_clave | 10800 | 0 | 16 | 2 | 4 |
| first_order_theorem_proving | 6118 | 51 | 0 | | 2 |
| gas_sensor_drift_class | 13910 | 128 | 0 | | 6 |
| gesture_phase_segmentation_raw | 9900 | 19 | 0 | | 5 |
| gesture_phase_segmentation_va3 | 9873 | 32 | 0 | | 5 |
| htru2 | 17898 | 8 | 0 | | 2 |
| human_activity_smartphone | 10299 | 561 | 0 | | 6 |
| indoor_loc_building | 21048 | 470 | 50 | 2 | 3 |
| indoor_loc_relative | 21048 | 470 | 50 | 2 | 3 |
| insurance_benchmark | 9822 | 80 | 4 | 5 | 2 |
| landsat_satimage | 6435 | 36 | 0 | | 6 |
| letter_recognition | 20000 | 16 | 0 | | 26 |
| madelon | 2600 | 500 | 0 | | 2 |
| magic_gamma_telescope | 19020 | 10 | 0 | | 2 |
| mushroom | 8124 | 0 | 21 | 12 | 2 |
| musk | 6598 | 166 | 0 | | 2 |
| nomao | 34465 | 118 | 2 | 2 | 2 |
| nursery | 12960 | 7 | 1 | 2 | 4 |
| occupancy_detection | 20560 | 7 | 0 | | 2 |
| online_shoppers_attention | 12330 | 16 | 2 | 3 | 2 |
| optical_recognition_handwritten_digits | 5620 | 59 | 3 | 2 | 10 |
| ozone_level_1hr | 1848 | 72 | 0 | | 2 |
| ozone_level_8hr | 1847 | 72 | 0 | | 2 |
| page_blocks | 5473 | 10 | 0 | | 5 |
| pen_recognition_handwritten_characters | 10992 | 16 | 0 | | 10 |
| phishing | 11055 | 8 | 22 | 2 | 2 |
| polish_companies_bankruptcy_1year | 7027 | 64 | 0 | | 2 |
| polish_companies_bankruptcy_2year | 10173 | 64 | 0 | | 2 |
| polish_companies_bankruptcy_3year | 10503 | 64 | 0 | | 2 |
| polish_companies_bankruptcy_4year | 9792 | 64 | 0 | | 2 |
| polish_companies_bankruptcy_5year | 5910 | 64 | 0 | | 2 |
| seismic_bumps | 2584 | 12 | 3 | 2 | 2 |
| skill_craft | 3338 | 18 | 0 | | 7 |
| smartphone_human_activity | 5744 | 561 | 0 | | 6 |
| smartphone_human_activity_postural | 10411 | 561 | 0 | | 6 |
| spambase | 4601 | 57 | 0 | | 2 |
| superconductivity_class | 21263 | 81 | 0 | | 2 |
| thyroid_all_bp | 3621 | 6 | 17 | 5 | 2 |
| thyroid_all_hyper | 3621 | 6 | 17 | 5 | 2 |
| thyroid_all_hypo | 3621 | 6 | 17 | 5 | 3 |
| thyroid_all_rep | 3621 | 6 | 17 | 5 | 2 |
| thyroid_ann | 7200 | 6 | 11 | 3 | 3 |
| thyroid_dis | 3621 | 6 | 17 | 5 | 2 |
| thyroid_hypo | 2700 | 7 | 14 | 3 | 2 |
| thyroid_sick | 3621 | 6 | 17 | 5 | 2 |
| thyroid_sick_eu | 3163 | 8 | 18 | 2 | 2 |
| turkiye_student_evaluation | 5820 | 32 | 0 | | 3 |
| wall_follow_robot_2 | 5456 | 2 | 0 | | 4 |
| wall_follow_robot_24 | 5456 | 24 | 0 | | 4 |
| wall_follow_robot_4 | 5456 | 4 | 0 | | 4 |
| waveform | 5000 | 21 | 0 | | 3 |
| waveform_noise | 5000 | 40 | 0 | | 3 |
| wilt | 4839 | 5 | 0 | | 2 |
| wine_quality_all | 6497 | 11 | 1 | 2 | 7 |
| wine_quality_type | 6497 | 11 | 0 | | 2 |
| wine_quality_white | 4898 | 11 | 0 | | 7 |

Table C.21: Datasets in the meta-train regression benchmark.

| Name | #samples | #num. features | #cat. features | largest #categories |
|---|---|---|---|---|
| air_quality_bc | 8991 | 10 | 0 | |
| air_quality_co2 | 7674 | 10 | 0 | |
| air_quality_no2 | 7715 | 10 | 0 | |
| air_quality_nox | 7718 | 10 | 0 | |
| appliances_energy | 19735 | 29 | 0 | |
| bejing_pm25 | 41757 | 12 | 0 | |
| bike_sharing_casual | 17379 | 9 | 3 | 2 |
| bike_sharing_total | 17379 | 9 | 3 | 2 |
| carbon_nanotubes_u | 10721 | 5 | 0 | |
| carbon_nanotubes_v | 10721 | 5 | 0 | |
| carbon_nanotubes_w | 10721 | 5 | 0 | |
| chess_krvk | 28056 | 3 | 3 | 8 |
| cycle_power_plant | 9568 | 4 | 0 | |
| electrical_grid_stability_simulated | 10000 | 12 | 0 | |
| facebook_comment_volume | 40949 | 38 | 2 | 7 |
| facebook_live_sellers_thailand_shares | 7050 | 9 | 0 | |
| five_cities_beijing_pm25 | 19062 | 14 | 0 | |
| five_cities_chengdu_pm25 | 21074 | 14 | 0 | |
| five_cities_guangzhou_pm25 | 20074 | 14 | 0 | |
| five_cities_shanghai_pm25 | 21436 | 14 | 0 | |
| five_cities_shenyang_pm25 | 19038 | 14 | 0 | |
| gas_sensor_drift_class | 13910 | 128 | 0 | |
| gas_sensor_drift_conc | 13910 | 128 | 0 | |
| indoor_loc_alt | 21048 | 470 | 50 | 2 |
| indoor_loc_lat | 21048 | 470 | 50 | 2 |
| indoor_loc_long | 21048 | 470 | 50 | 2 |
| insurance_benchmark | 9822 | 80 | 4 | 5 |
| metro_interstate_traffic_volume_long | 48204 | 6 | 2 | 38 |
| metro_interstate_traffic_volume_short | 48204 | 6 | 2 | 11 |
| naval_propulsion_comp | 11934 | 14 | 0 | |
| naval_propulsion_turb | 11934 | 14 | 0 | |
| nursery | 12960 | 7 | 1 | 2 |
| online_news_popularity | 39644 | 44 | 3 | 7 |
| parking_birmingham | 35717 | 5 | 0 | |
| parkinson_motor | 5875 | 18 | 1 | 2 |
| parkinson_total | 5875 | 18 | 1 | 2 |
| protein_tertiary_structure | 45730 | 9 | 0 | |
| skill_craft | 3338 | 18 | 0 | |
| sml2010_dining | 4137 | 17 | 0 | |
| sml2010_room | 4137 | 17 | 0 | |
| superconductivity | 21263 | 81 | 0 | |
| travel_review_ratings | 5456 | 23 | 0 | |
| wall_follow_robot_2 | 5456 | 2 | 0 | |
| wall_follow_robot_24 | 5456 | 24 | 0 | |
| wall_follow_robot_4 | 5456 | 4 | 0 | |
| wine_quality_all | 6497 | 11 | 1 | 2 |
| wine_quality_white | 4898 | 11 | 0 | |

After preprocessing, we

- exclude datasets with less than 1,000 samples, these were
  - for classification: albert, APSFailure, arcene, Australian, blood-transfusion-service-center, eucalyptus, KDDCup09_appetency, KDDCup09-Upselling, micro-mass, vehicle
  - for regression: boston, cars, colleges, energy_efficiency, forest_fires, Moneyball, QSAR_fish_toxicity, sensory, student_performance_por, tecator, us_crime
- exclude datasets that have more than 10,000 features after one-hot encoding. These were Amazon_employee_access, Click_prediction_small, and sf-police-incidents (all classification).

### C.3.3 Grinsztajn et al. [18] Benchmarks

We select the datasets as follows:

- We use the newer version of the benchmark on OpenML.
- When a dataset is used both in benchmarks with and without categorical features, we use the version with categorical features.
- We exclude the eye_movements dataset since a leak in the dataset was reported by Gorishniy et al. [17].

Table C.22: Datasets in the meta-test classification benchmark.

| Name | #samples | #num. features | #cat. features | largest #categories | #classes | OpenML task ID |
|---|---|---|---|---|---|---|
| Bioresponse | 3751 | 1776 | 0 | | 2 | 359967 |
| Diabetes130US | 101766 | 13 | 36 | 789 | 3 | 211986 |
| Fashion-MNIST | 70000 | 784 | 0 | | 10 | 359976 |
| Higgs | 500000 | 28 | 0 | | 2 | 360114 |
| Internet-Advertisements | 3279 | 3 | 1555 | 2 | 2 | 359966 |
| KDDCup99 | 500000 | 32 | 9 | 65 | 21 | 360112 |
| MiniBooNE | 130064 | 50 | 0 | | 2 | 359990 |
| Satellite | 5100 | 36 | 0 | | 2 | 359975 |
| ada | 4147 | 48 | 0 | | 2 | 190411 |
| airlines | 500000 | 3 | 4 | 293 | 2 | 189354 |
| amazon-commerce-reviews | 1500 | 10000 | 0 | | 50 | 10090 |
| car | 1728 | 0 | 6 | 4 | 4 | 359960 |
| christine | 5418 | 1599 | 37 | 2 | 2 | 359973 |
| churn | 5000 | 16 | 4 | 10 | 2 | 359968 |
| cmc | 1473 | 2 | 7 | 4 | 3 | 359959 |
| cnae-9 | 1080 | 856 | 0 | | 9 | 359957 |
| connect-4 | 67557 | 0 | 42 | 3 | 3 | 359977 |
| covertype | 500000 | 10 | 44 | 2 | 7 | 7593 |
| credit-g | 1000 | 7 | 13 | 10 | 2 | 168757 |
| dilbert | 10000 | 2000 | 0 | | 5 | 168909 |
| dionis | 100000 | 60 | 0 | | 355 | 189355 |
| dna | 3186 | 0 | 180 | 2 | 3 | 359964 |
| fabert | 8237 | 800 | 0 | | 7 | 168910 |
| gina | 3153 | 970 | 0 | | 2 | 189922 |
| guillermo | 20000 | 4296 | 0 | | 2 | 359988 |
| helena | 65196 | 27 | 0 | | 100 | 359984 |
| jannis | 83733 | 54 | 0 | | 4 | 211979 |
| jasmine | 2984 | 8 | 136 | 2 | 2 | 168911 |
| jungle_chess_2pcs_raw_endgame_complete | 44819 | 6 | 0 | | 3 | 359981 |
| kc1 | 2109 | 21 | 0 | | 2 | 359962 |
| kick | 72600 | 14 | 18 | 1054 | 2 | 359991 |
| madeline | 3140 | 259 | 0 | | 2 | 190392 |
| mfeat-factors | 2000 | 216 | 0 | | 10 | 359961 |
| numerai28.6 | 96320 | 21 | 0 | | 2 | 167120 |
| okcupid-stem | 50788 | 2 | 17 | 7019 | 3 | 359993 |
| pc4 | 1458 | 37 | 0 | | 2 | 359958 |
| philippine | 5832 | 308 | 0 | | 2 | 190410 |
| phoneme | 5404 | 5 | 0 | | 2 | 168350 |
| porto-seguro | 453046 | 26 | 31 | 102 | 2 | 360113 |
| qsar-biodeg | 1055 | 41 | 0 | | 2 | 359956 |
| riccardo | 20000 | 4296 | 0 | | 2 | 359989 |
| robert | 10000 | 7200 | 0 | | 10 | 359986 |
| segment | 2310 | 16 | 0 | | 7 | 359963 |
| shuttle | 58000 | 9 | 0 | | 7 | 359987 |
| steel-plates-fault | 1941 | 27 | 0 | | 7 | 168784 |
| sylvine | 5124 | 20 | 0 | | 2 | 359972 |
| volkert | 58310 | 180 | 0 | | 10 | 359985 |
| yeast | 1484 | 8 | 0 | | 10 | 2073 |

Table C.23: Datasets in the meta-test regression benchmark.

| Name | #samples | #num. features | #cat. features | largest #categories | OpenML task ID |
|---|---|---|---|---|---|
| Airlines_DepDelay_10M | 500000 | 6 | 3 | 359 | 359929 |
| Allstate_Claims_Severity | 188318 | 14 | 116 | 326 | 233212 |
| Buzzinsocialmedia_Twitter | 500000 | 77 | 0 | | 233213 |
| MIP-2016-regression | 1090 | 143 | 1 | 5 | 360945 |
| Mercedes_Benz_Greener_Manufacturing | 4209 | 368 | 8 | 47 | 233215 |
| QSAR-TID-10980 | 5766 | 1024 | 0 | | 360933 |
| QSAR-TID-11 | 5742 | 1024 | 0 | | 360932 |
| SAT11-HAND-runtime-regression | 1725 | 115 | 1 | 15 | 359948 |
| Santander_transaction_value | 4459 | 4991 | 0 | | 233214 |
| Yolanda | 400000 | 100 | 0 | | 317614 |
| airfoil_self_noise | 1503 | 5 | 0 | | 361235 |
| auction_verification | 2043 | 5 | 2 | 6 | 361236 |
| black_friday | 166821 | 5 | 4 | 7 | 359937 |
| brazilian_houses | 10692 | 5 | 4 | 35 | 361267 |
| california_housing | 20640 | 8 | 0 | | 361255 |
| concrete_compressive_strength | 1030 | 8 | 0 | | 361237 |
| cps88wages | 28155 | 2 | 4 | 4 | 361261 |
| cpu_activity | 8192 | 21 | 0 | | 361256 |
| diamonds | 53940 | 6 | 3 | 8 | 361257 |
| elevators | 16599 | 18 | 0 | | 359936 |
| fifa | 19178 | 27 | 1 | 163 | 361272 |
| fps_benchmark | 2592 | 29 | 14 | 24 | 361268 |
| geographical_origin_of_music | 1059 | 116 | 0 | | 361243 |
| health_insurance | 22272 | 4 | 7 | 6 | 361269 |
| house_16H | 22784 | 16 | 0 | | 359952 |
| house_prices_nominal | 1121 | 36 | 43 | 25 | 359951 |
| house_sales | 21613 | 20 | 1 | 70 | 359949 |
| kin8nm | 8192 | 8 | 0 | | 361258 |
| kings_county | 21613 | 17 | 4 | 70 | 361266 |
| miami_housing | 13932 | 15 | 0 | | 361260 |
| nyc-taxi-green-dec-2016 | 500000 | 9 | 9 | 259 | 359943 |
| pol | 15000 | 48 | 0 | | 359946 |
| pumadyn32nh | 8192 | 32 | 0 | | 361259 |
| quake | 2178 | 3 | 0 | | 359930 |
| sarcos | 44484 | 21 | 0 | | 361011 |
| socmob | 1156 | 1 | 4 | 17 | 361264 |
| solar_flare | 1066 | 2 | 8 | 6 | 361244 |
| space_ga | 3107 | 6 | 0 | | 361623 |
| topo_2_1 | 8885 | 266 | 0 | | 359939 |
| video_transcoding | 68784 | 16 | 2 | 4 | 361252 |
| wave_energy | 72000 | 48 | 0 | | 361253 |
| yprop_4_1 | 8885 | 251 | 0 | | 359940 |

Table C.24: Datasets in the Grinsztajn et al. [18] classification benchmark.

| Name | #samples | #num. features | #cat. features | largest #categories | #classes | OpenML task ID |
|---|---|---|---|---|---|---|
| Bioresponse | 3434 | 419 | 0 | | 2 | 361276 |
| Diabetes130US | 71090 | 7 | 0 | | 2 | 361273 |
| Higgs | 500000 | 24 | 0 | | 2 | 361069 |
| MagicTelescope | 13376 | 10 | 0 | | 2 | 361065 |
| MiniBooNE | 72998 | 50 | 0 | | 2 | 361068 |
| albert | 58252 | 21 | 10 | 14 | 2 | 361282 |
| bank-marketing | 10578 | 7 | 0 | | 2 | 361066 |
| california | 20634 | 8 | 0 | | 2 | 361277 |
| compas-two-years | 4966 | 3 | 8 | 2 | 2 | 361286 |
| covertype | 423680 | 10 | 44 | 2 | 2 | 361113 |
| credit | 16714 | 10 | 0 | | 2 | 361055 |
| default-of-credit-card-clients | 13272 | 20 | 1 | 2 | 2 | 361283 |
| electricity | 38474 | 7 | 1 | 7 | 2 | 361110 |
| heloc | 10000 | 22 | 0 | | 2 | 361278 |
| house_16H | 13488 | 16 | 0 | | 2 | 361063 |
| jannis | 57580 | 54 | 0 | | 2 | 361274 |
| pol | 10082 | 26 | 0 | | 2 | 361062 |
| road-safety | 111762 | 29 | 3 | 2 | 2 | 361285 |

Table C.25: Datasets in the Grinsztajn et al. [18] regression benchmark.

| Name | #samples | #num. features | #cat. features | largest #categories | OpenML task ID |
|---|---|---|---|---|---|
| Ailerons | 13750 | 33 | 0 | | 361077 |
| Airlines_DepDelay_1M | 500000 | 5 | 0 | | 361293 |
| Allstate_Claims_Severity | 188318 | 14 | 110 | 20 | 361292 |
| Bike_Sharing_Demand | 17379 | 6 | 5 | 4 | 361099 |
| Brazilian_houses | 10692 | 8 | 3 | 5 | 361098 |
| Mercedes_Benz_Greener_Manufacturing | 4209 | 0 | 359 | 12 | 361097 |
| MiamiHousing2016 | 13932 | 13 | 0 | | 361087 |
| SGEMM_GPU_kernel_performance | 241600 | 3 | 6 | 2 | 361104 |
| abalone | 4177 | 7 | 1 | 3 | 361288 |
| analcatdata_supreme | 4052 | 2 | 5 | 2 | 361093 |
| cpu_act | 8192 | 21 | 0 | | 361072 |
| delays_zurich_transport | 500000 | 8 | 3 | 7 | 361291 |
| diamonds | 53940 | 6 | 3 | 8 | 361096 |
| elevators | 16599 | 16 | 0 | | 361074 |
| house_16H | 22784 | 16 | 0 | | 361079 |
| house_sales | 21613 | 15 | 2 | 2 | 361102 |
| houses | 20640 | 8 | 0 | | 361078 |
| medical_charges | 163065 | 3 | 0 | | 361294 |
| nyc-taxi-green-dec-2016 | 500000 | 9 | 7 | 5 | 361101 |
| particulate-matter-ukair-2017 | 394299 | 3 | 3 | 12 | 361103 |
| pol | 15000 | 26 | 0 | | 361073 |
| seattlecrime6 | 52031 | 2 | 2 | 17 | 361289 |
| sulfur | 10081 | 6 | 0 | | 361085 |
| superconduct | 21263 | 79 | 0 | | 361088 |
| topo_2_1 | 8885 | 252 | 3 | 2 | 361287 |
| visualizing_soil | 8641 | 3 | 1 | 2 | 361094 |
| wine_quality | 6497 | 11 | 0 | | 361076 |
| yprop_4_1 | 8885 | 42 | 0 | | 361279 |

## C.4 Comparison with Standard Grinsztajn et al. [18] Benchmark

Here, we compare two versions of the Grinsztajn et al. [18] benchmark:

(a) The "new" version, using our benchmarking setup with the datasets of the Grinsztajn et al. [18] benchmark. This version is used in all plots except Figure C.2 and Figure C.3.

(b) The "old" version, which is a slightly modified version of the original code, described in Appendix C.5.

The corresponding results for the most comparable metrics are shown in Figure C.1 for the new paper version, and Figure C.2 for the old version. We decided to use the new version for multiple reasons, including a more realistic validation setting, having the exact same baselines, and having more options for evaluation and plotting. Here is a list of differences in our adapted version:

- In the new version, we removed the `eye_movements` dataset due to a leak reported in Gorishniy et al. [17].
- We subsample datasets after downloading them to 500K samples (all train-test splits are performed on the same 500K samples).
- We always standardize targets, to make our results independent of the scaling of the datasets. (In contrast, HPO methods on the original benchmark have standardization as an option in their tuning space.)
- We limit training+validation set sizes to 13333, such that at most 10K samples are used for training. Of these samples, we always use 25% for validation, unlike the original benchmark, which limits the training and validation set sizes separately to 10K and 50K samples.
- The new version does not use separate validation sets for early stopping and for HPO, which avoids unfairly disadvantaging D and TD methods compared to HPO methods.
- With the new version, we mostly report results using different aggregation strategies and using nRMSE instead of $R^2$ for regression, but try to provide comparable aggregated metrics in Figure C.1.
- The new version uses different random hyperparameter configurations on different train-test splits, which should provide more accurate results and allows computing confidence intervals as in Appendix C.6.
- The new version uses ten train-test splits on all datasets, instead of a smaller dataset-size-dependent number.
- The new version measures all runtimes on the CPU, while the old version measures NN runtimes on the GPU.
- The old version uses slightly different baseline configurations:
  - The old version uses (in the code) a simplified version of the MLP without dropout and without weight decay.
  - The old version sometimes replaces search spaces like Choice([0, LogUniform[1e-6, 1e-3]]) with more simple spaces.
  - The new version doesn't use as large categorical embedding sizes for ResNet and MLP models (up to 64 instead of [64, 512]).
  - The old version uses larger stopping patiences for default models than in the original literature [15].
  - In the old version, ResNet-HPO tunes the normalization, unlike the original paper [15].
  - In the old version, the batch size is tuned for some models.
  - In the old version, XGBoost uses the *exact* tree method with one-hot encoding, while in the new version, we use the *hist method* that supports native categorical feature handling. This makes XGBoost slower but also more accurate in the older version.
  - The old version uses different versions of quantile preprocessing for NN methods, while we use the RTDL quantile transform for all methods except RealMLP.

Both the new and the old version use early stopping and best-epoch selection on accuracy (for classification) / RMSE (for regression).

## C.5 Closer-to-original Version of the Grinsztajn et al. [18] Benchmark

In the following, we document the benchmark settings for obtaining the results in Figure C.2 and Figure C.3. The results were obtained using a modification of the original code.

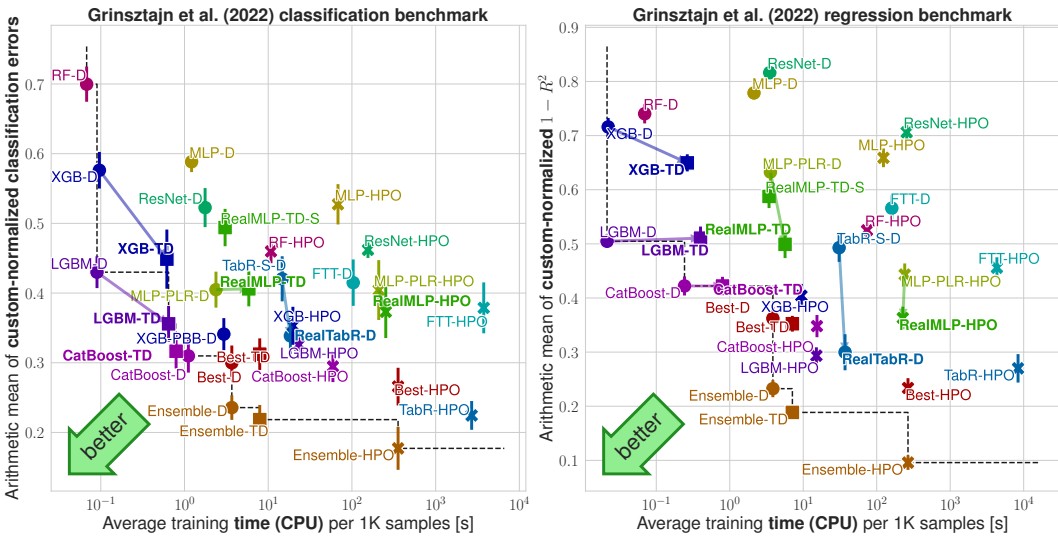

Figure C.1: **Benchmark scores (custom normalized errors) vs. average training time.** The $y$-axis shows the *arithmetic mean* normalized error as described in Appendix C.5, averaged over all splits and datasets. Errors are normalized by rescaling the lowest error to zero and the largest error to one. The $x$-axis shows average training times per 1000 samples (measured on $\mathcal{B}^{\text{train}}$ for efficiency reasons), see Appendix C.7. The error bars are approximate 95% confidence intervals for the limit #splits $\rightarrow \infty$, see Appendix C.6.

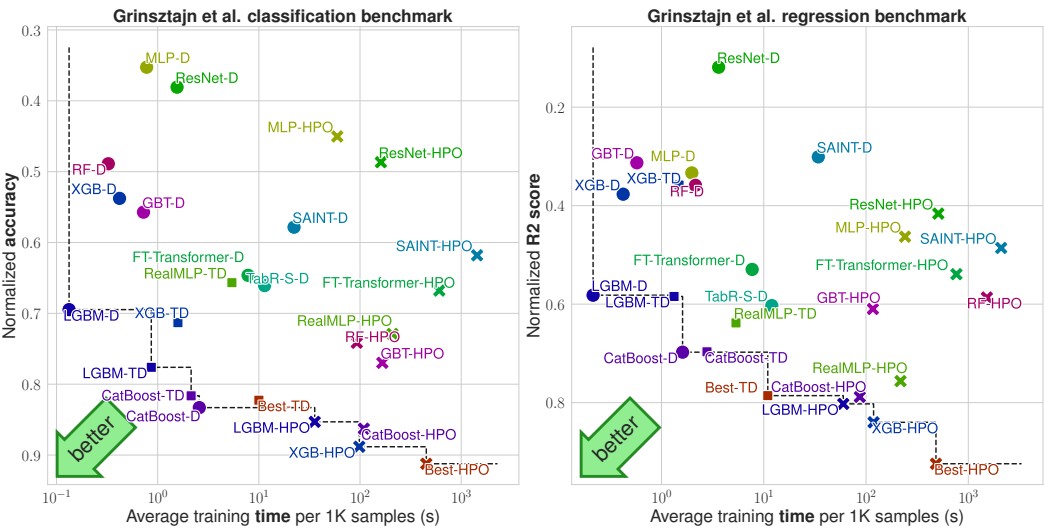

Figure C.2: **Results on the benchmarks of Grinsztajn et al. [18], using closer-to-original settings (Appendix C.5).** The $y$-axis (inverted) shows the normalized accuracy / R2 score used in the original paper (see Appendix C.5). The $x$-axis shows average training times per 1000 samples, using GPUs for NNs as in Grinsztajn et al. [18], see Appendix C.5.

The datasets are taken from the benchmarks described in Grinsztajn et al. [18]. When a dataset is used both in benchmarks with and without categorical features, we use the version with categorical features. We preprocess the datasets following the same steps as in Grinsztajn et al. [18]:

- For neural networks, we quantile-transform the features to have a Gaussian distribution. For TabR [17], we use the modified quantile transform from the TabR paper. For RealMLP, we use the preprocessing described in Section 3, namely robust scaling and smooth clipping.
- For neural networks, we add as a hyperparameter the possibility to normalize the target variable for the model fit and transform it back for evaluation (via scikit-learn's Transformed-

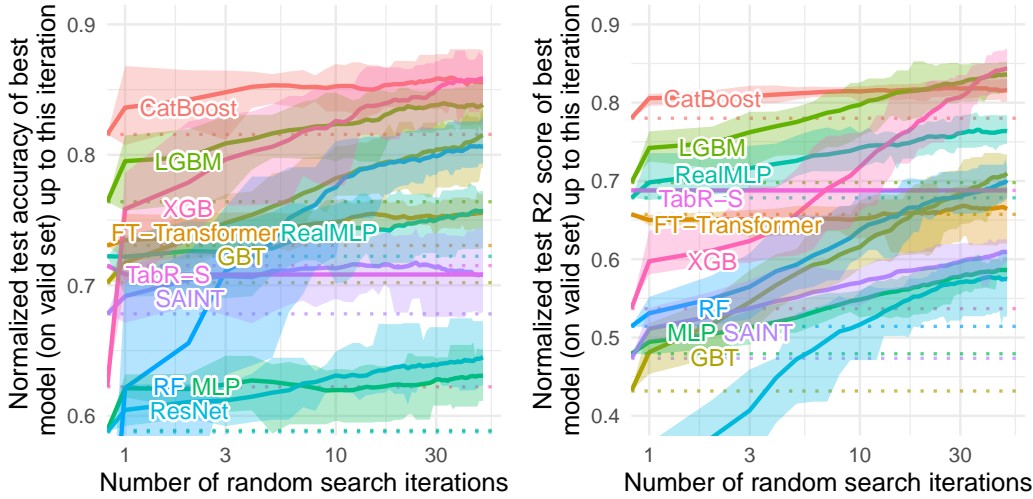

Figure C.3: **Results on the benchmarks of Grinsztajn et al. [18], for classification (left) and regression (right), using the closer-to-original settings (Appendix C.5).** The plot is similar to the one in the main part of Grinsztajn et al. [18], with our algorithms added. The $y$-axis shows the result of the best (on val, but evaluated on test) hyperparameter combination up to n steps of random step ($x$-axis). As in the original paper, we normalize each score between the max and the 10% quantile (classification) or 50% (regression), and truncate scores below 0 for regression.

TargetRegressor and StandardScaler, which differs from the QuantileTransformer from the original paper, as we found it to work better). The same standardization is also applied to all default-parameter versions of neural networks.

- For models that do not handle categorical variables natively, we encode categorical features using `OneHotEncoder` from scikit-learn.
- Train size is restricted to 10,000 samples and test and validation size to 50,000 samples.

Note that the datasets from the original benchmark are already slightly preprocessed, e.g., heavy-tailed targets are standardized and missing values are removed. More details can be found in the original paper.

**Results normalization** For Figure C.2, as in the original paper, we normalize the R2 or accuracy score for each dataset before averaging them. We use an affine normalization between 0 and 1, 1 corresponding to the score of the best model for each dataset, and 0 corresponding to the score of the worst model (for classification) and the 10th percentile of the scores (for regression). We use slightly different percentiles compared to the original paper as we normalize across the scores of the tuned and default models, and not all steps of the random search, which reduces the number of outliers. Other aggregation metrics are shown in Appendix B.10.

**Time measurement** We follow the original paper and run neural networks on a GPU and the other models on 1 core of an AMD EPYC 7742 64-Core processor, and we average the time across all random steps (for each random step, the time is averaged across splits). To compute the runtime of neural networks, we restrict ourselves to steps ran on the same GPU model (NVIDIA A100-40GB), which means that we exclude datasets for which we have less than 15 steps of each model on this GPU (leaving us with 11 datasets for classification and 15 for regression). We then compute the average runtime per 1000 samples on each dataset and average them.

**Other details** We rerun classification results for neural networks compared to the original results to early stop on accuracy rather than on cross-entropy, to make results more comparable with the rest of this paper.

At `https://github.com/LeoGrin/tabular-benchmark/tree/better_by_default`, we provide code for the adapted original Grinsztajn et al. [18] benchmark.

## C.6 Confidence Intervals

Here, we specify how our confidence intervals are computed. Let $X_{ij}$ denote the score (error/rank) of a method on dataset $i$ and split $j$, with $i \in \{1, \ldots, n\}$ and $j \in \{1, \ldots, m\}$. Then, the benchmark score $\mathcal{S}$ can be written as

$$\mathcal{S} = g \left( \sum_{i=1}^{n} \frac{w_i}{m} \sum_{j=1}^{m} f(X_{ij}) \right), \tag{1}$$

where $f = g = \mathrm{id}$ for the arithmetic mean. For the shifted geometric mean, we instead have $g = \exp$ and $f(x) = \log(x + \varepsilon)$, $\varepsilon = 0.01$. We interpret the benchmark datasets as fixed, but the splits as random. For each dataset $i$, $X_{i1}, \ldots, X_{im}$ are i.i.d. random variables. We first take the dataset averages

$$Z_j := \sum_{i=1}^{n} w_i f(X_{ij}) .$$

The random variables $X_{1j}, \ldots, X_{nj}$ are independent but not identically distributed. Still, for lack of a better option, we assume that the $Z_j$ are normally distributed with unknown mean and variance. We know that the $Z_j$ are i.i.d., hence we use the confidence intervals from the Student's $t$-distribution for normally distributed random variables with unknown mean and variance. This gives us a confidence interval $[a, b]$ for $\frac{1}{m} \sum_{j=1}^{m} Z_j$. Since $g$ is increasing, we hence obtain a confidence interval $[g(a), g(b)]$ for $\mathcal{S} = g \left( \frac{1}{m} \sum_{j=1}^{m} Z_j \right)$.

**Comparison of two methods** We often compute the error increase in % in the benchmark score of method A compared to method B with the shifted geometric mean, given by

$$100 \cdot \left( \frac{\mathcal{S}^{(A)}}{\mathcal{S}^{(B)}} - 1 \right) .$$

Here, we leverage that the shifted geometric mean uses $g = \exp$ to write

$$\frac{\mathcal{S}^{(A)}}{\mathcal{S}^{(B)}} = g \left( \sum_{i=1}^{n} \frac{w_i}{m} \sum_{j=1}^{m} (f(X_{ij}^{(A)}) - f(X_{ij}^{(B)})) \right) ,$$

which is of the same form as Eq. (1). Hence, we obtain confidence intervals for this quantity using the same method.

## C.7 Time Measurements

For our meta-train and meta-test benchmarks, we report training times measured as follows: We run all methods on a single compute node with a 32-core AMD Ryzen Threadripper Pro 3975 WX CPU, using 32 threads for GBDTs and the PyTorch default settings for NNs. No method is run on GPUs. We run methods sequentially on one split on each dataset of the meta-train-class and meta-train-reg benchmarks. For random-search-based HPO methods, we only run one (TabR-HPO, FTT-HPO) or two (other methods) random search steps and extrapolate the runtime to 50 steps. Runtimes for combinations of models (Best and Ensemble) are computed as the sum of the individual runtimes. We compute the runtime per 1000 samples on each dataset and then average them. For simplicity, we do not use the dataset-dependent weighting employed otherwise on the meta-train benchmark.

## C.8 Compute Resources

While we did not measure compute resources precisely, our experiments required at least around 3000 hours on RTX 3090 GPUs and other GPUs, as well as roughly 10,000 hours on HPC CPU nodes (32–64 cores).

## C.9 Used Libraries

Our implementation uses various libraries, out of which we would like to particularly acknowledge PyTorch [47], Scikit-learn [48], Ray [46], XGBoost [9], LightGBM [31], and CatBoost [51]. For using XGBoost, LightGBM, and CatBoost, we adapted wrapping code from the CatBoost quality benchmarks [51].

# D  Results for Individual Datasets

Here, we provide and compare the results of central methods per dataset. Figures D.1 – D.7 show scatterplot comparisons for different models.

Table D.1 and Table D.2 show results on $\mathcal{B}_{\text{class}}^{\text{train}}$. Table D.3 and Table D.4 show results on $\mathcal{B}_{\text{reg}}^{\text{train}}$. Table D.5 and Table D.6 show results on $\mathcal{B}_{\text{class}}^{\text{test}}$. Table D.7 and Table D.8 show results on $\mathcal{B}_{\text{reg}}^{\text{test}}$. Table D.9 and Table D.10 show results on $\mathcal{B}_{\text{class}}^{\text{Grinsztajn}}$. Table D.11 and Table D.12 show results on $\mathcal{B}_{\text{reg}}^{\text{Grinsztajn}}$.

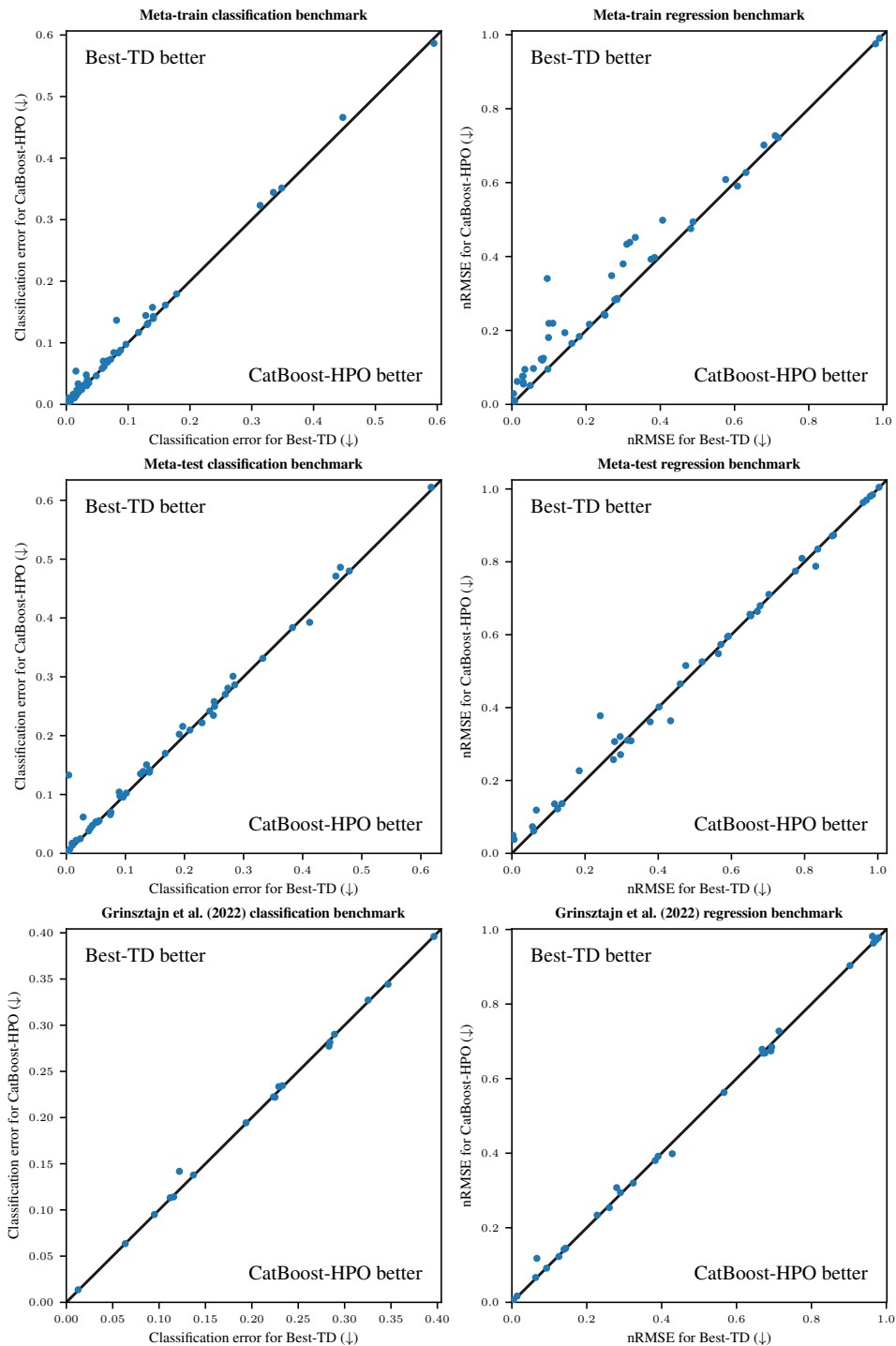

Figure D.1: **Best-TD vs CatBoost-HPO on individual datasets.** Each point represents the errors of both models on a dataset, averaged across 10 train-valid-test splits. The black line represents equal errors ($x = y$).

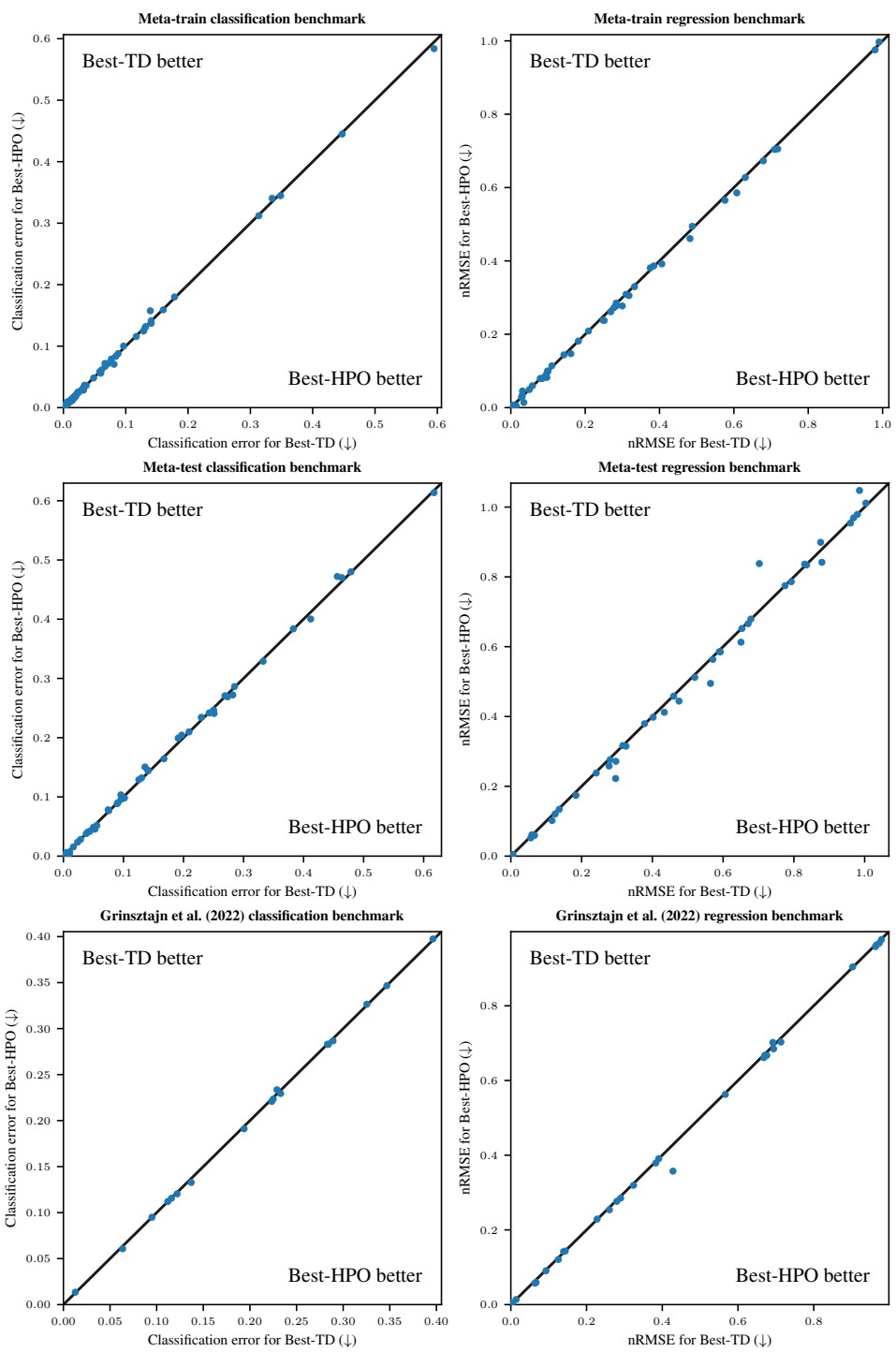

Figure D.2: **Best-TD vs Best-HPO on individual datasets.** Each point represents the errors of both models on a dataset, averaged across 10 train-valid-test splits. The black line represents equal errors $(x = y)$.

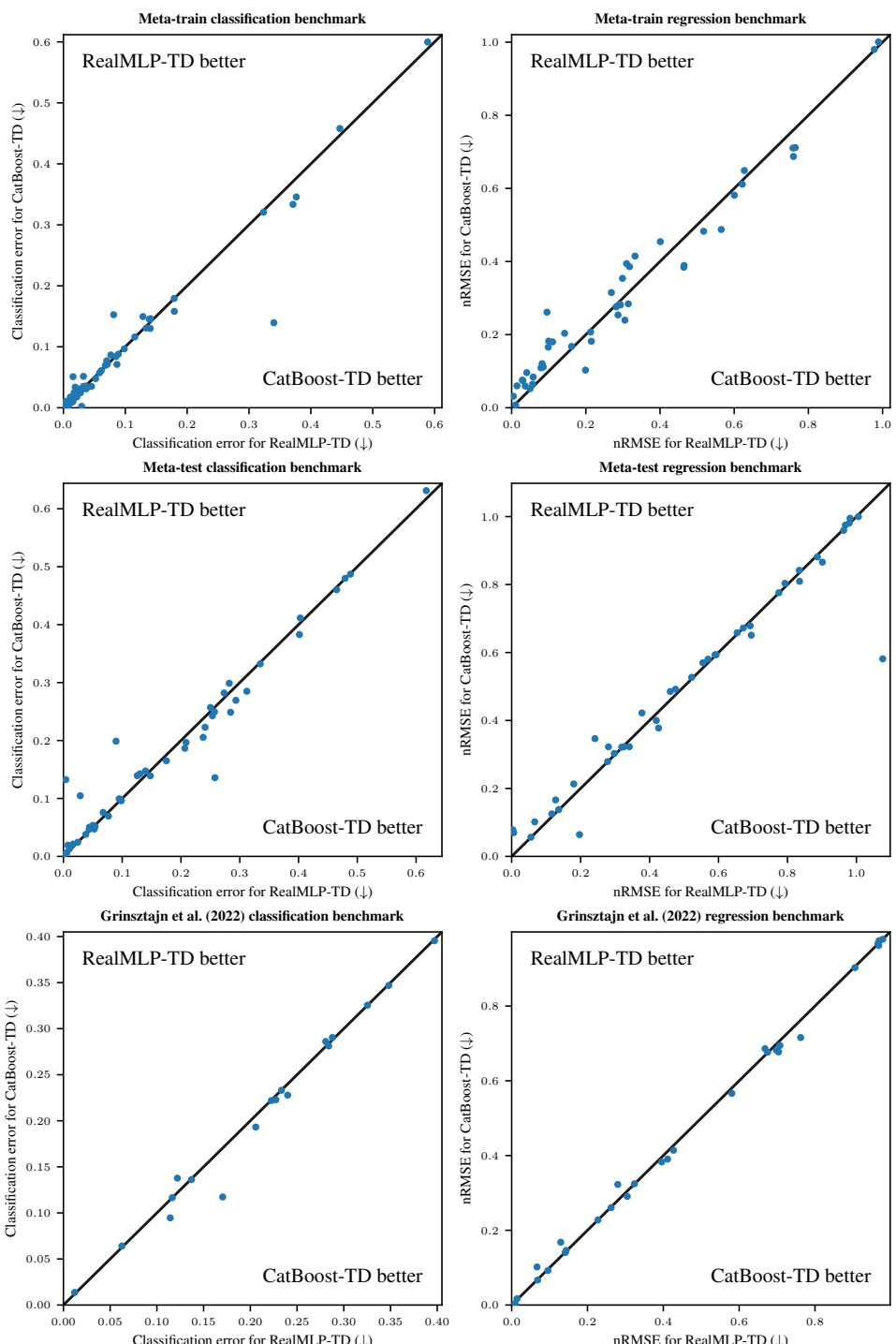

Figure D.3: **RealMLP-TD vs CatBoost-TD on individual datasets.** Each point represents the errors of both models on a dataset, averaged across 10 train-valid-test splits. The black line represents equal errors ($x = y$).

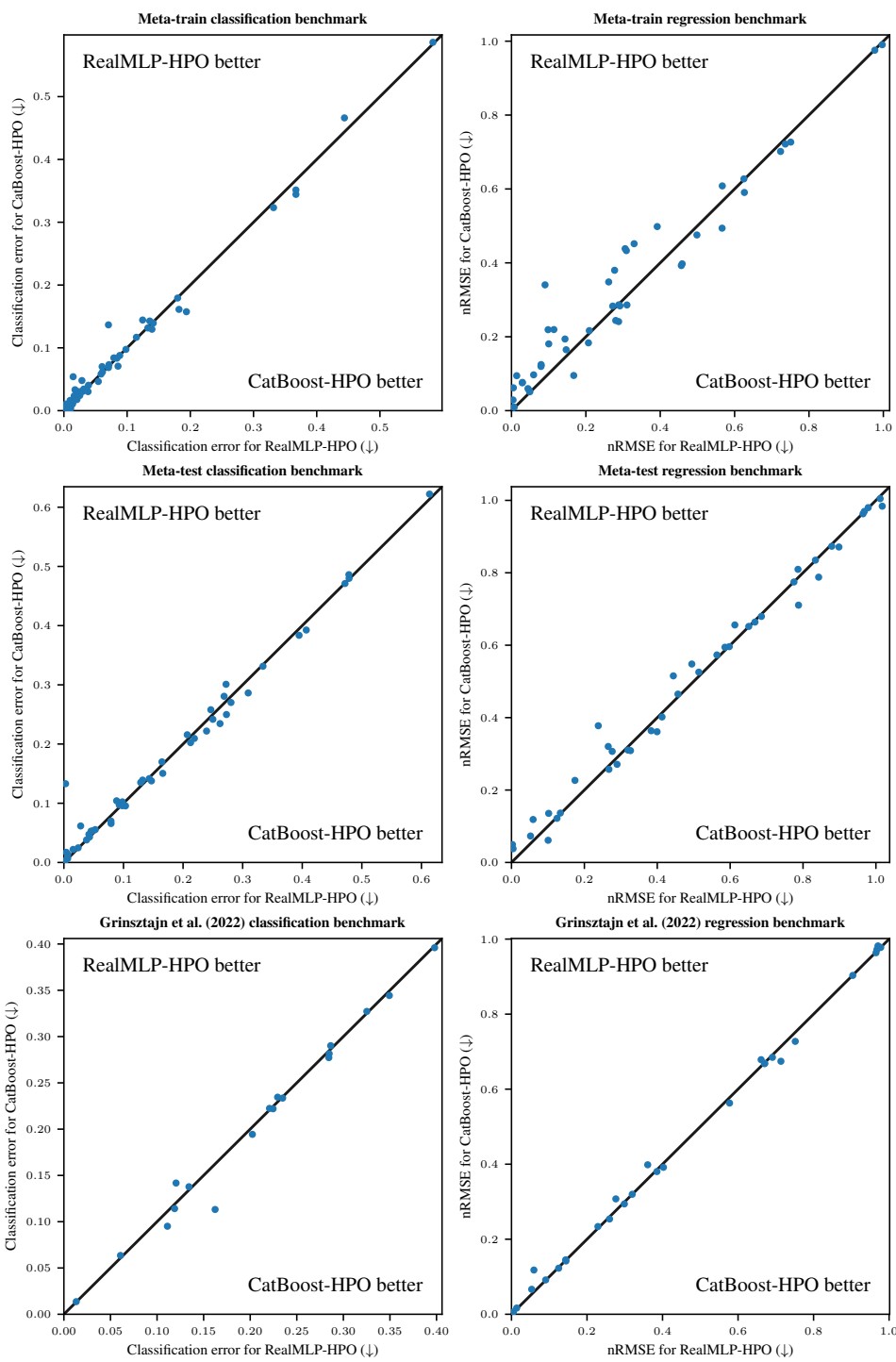

Figure D.4: **RealMLP-HPO vs CatBoost-HPO on individual datasets.** Each point represents the errors of both models on a dataset, averaged across 10 train-valid-test splits. The black line represents equal errors ($x = y$).

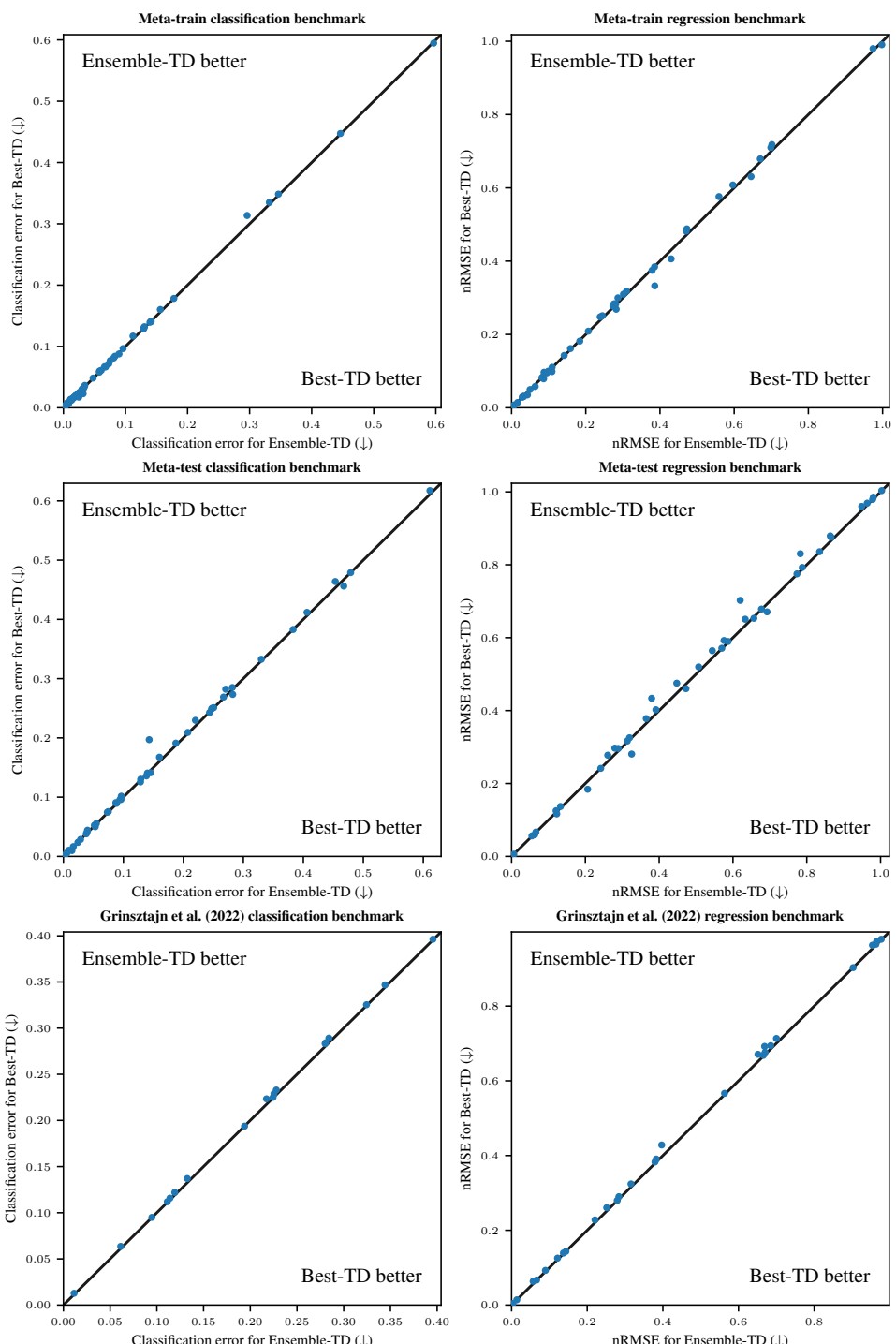

Figure D.5: **Ensemble-TD vs Best-TD on individual datasets.** Each point represents the errors of both models on a dataset, averaged across 10 train-valid-test splits. The black line represents equal errors ($x = y$).

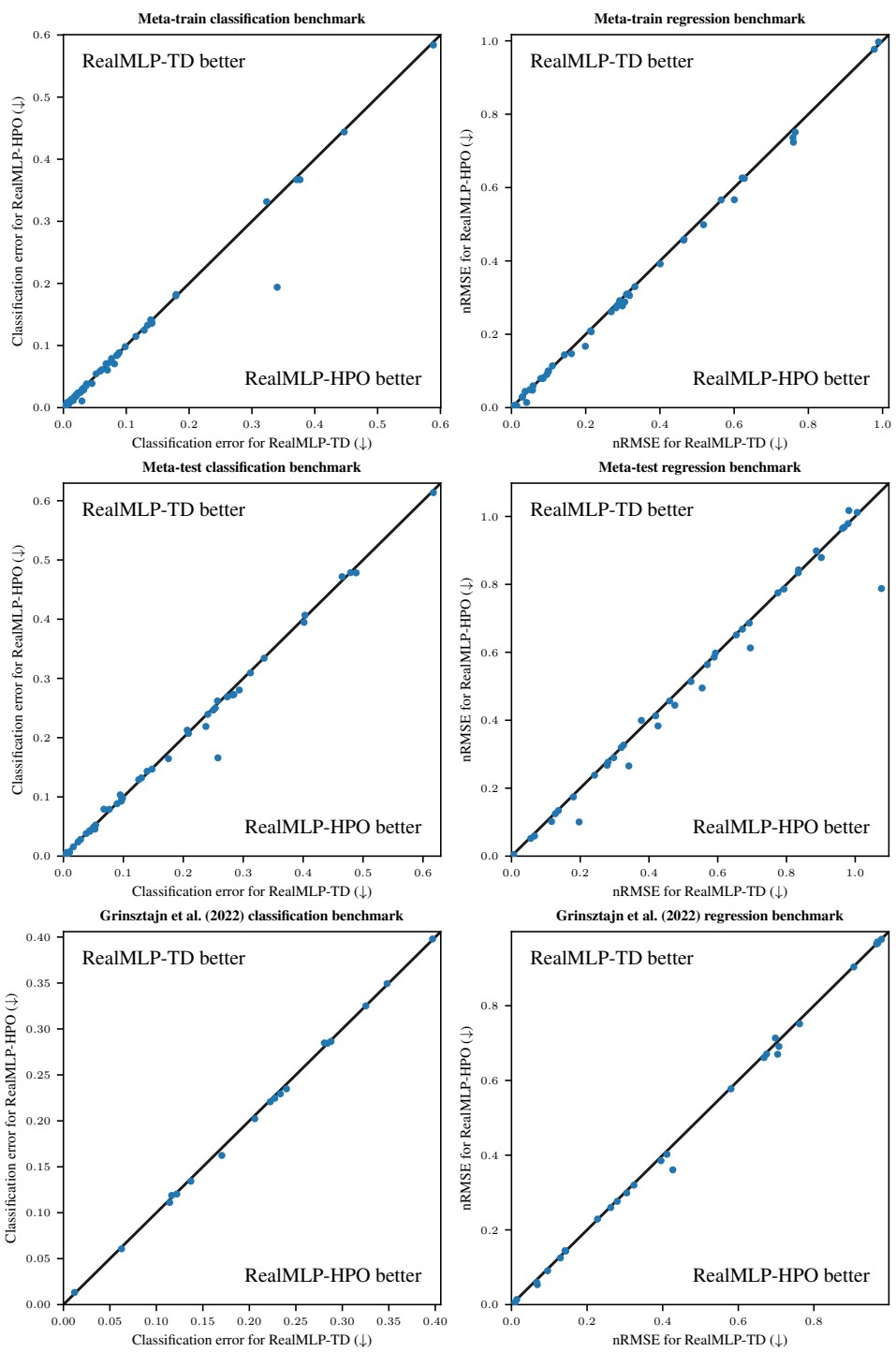

Figure D.6: **RealMLP-TD vs RealMLP-HPO on individual datasets.** Each point represents the errors of both models on a dataset, averaged across 10 train-valid-test splits. The black line represents equal errors ($x = y$).

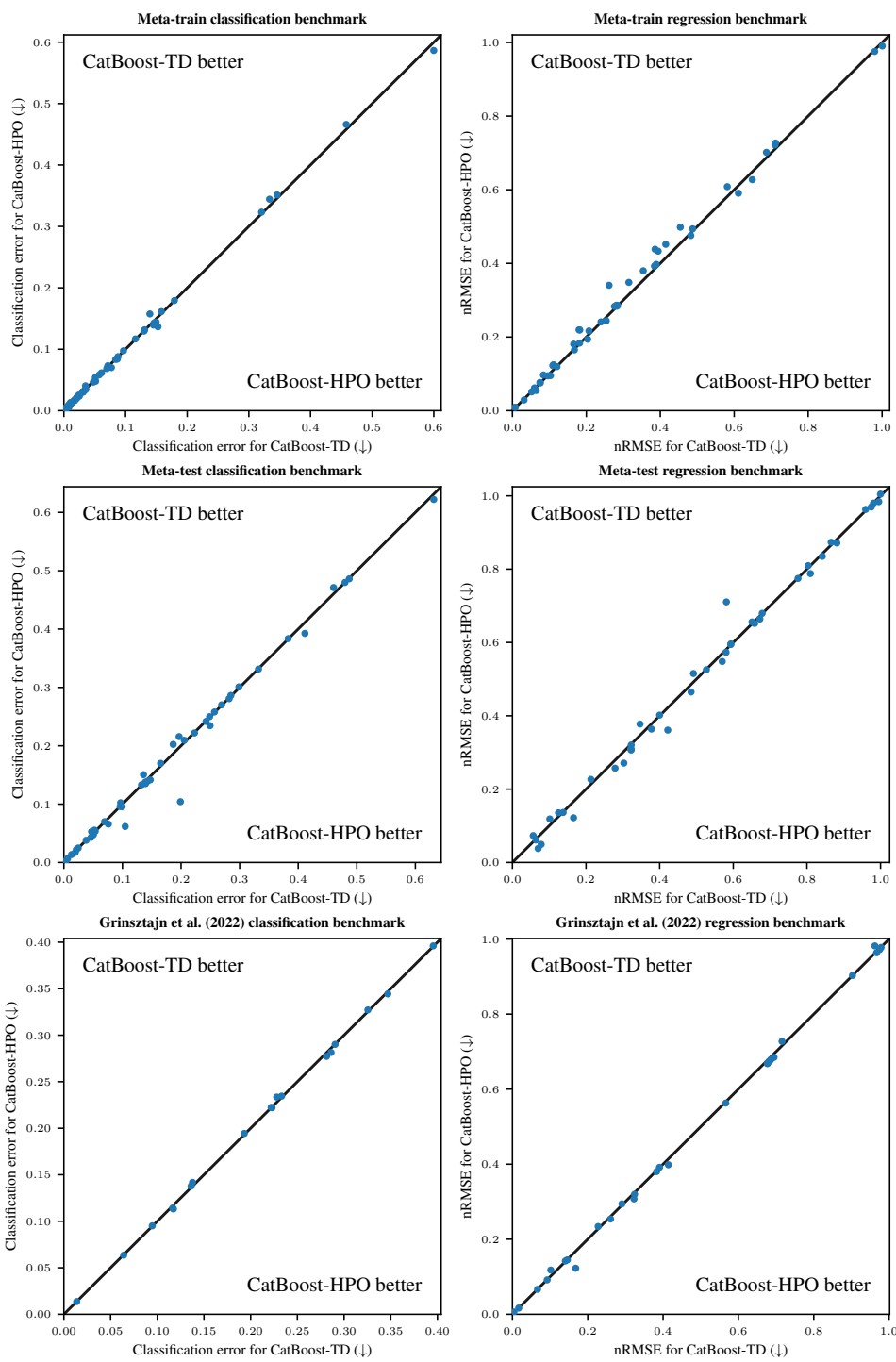

Figure D.7: **CatBoost-TD vs CatBoost-HPO on individual datasets.** Each point represents the errors of both models on a dataset, averaged across 10 train-valid-test splits. The black line represents equal errors ($x = y$).

Table D.1: Classification error of *untuned* methods on datasets in $\mathcal{B}_{\text{class}}^{\text{train}}$, averaged over ten train-validation-test splits. When we write $a \pm b$, $a$ is the mean error on the dataset and $[a - b, a + b]$ is an approximate 95% confidence interval for the mean in the #splits $\to \infty$ limit. The confidence interval is computed from the $t$-distribution using a normality assumption as in Appendix C.6. In each row, the lowest mean error is highlighted in bold, and errors whose confidence interval contains the lowest error are underlined.

| Dataset | RealMLP-TD | RealTabR-D | TabR-S-D | MLP-PLR-D | MLP-D | CatBoost-TD | LGBM-TD | XGB-TD | RF-D |
|---|---|---|---|---|---|---|---|---|---|
| abalone | 0.447±0.014 | 0.445±0.006 | **0.440**±0.010 | 0.453±0.010 | 0.448±0.014 | 0.458±0.009 | 0.455±0.012 | 0.451±0.013 | 0.457±0.010 |
| adult | 0.140±0.004 | 0.134±0.004 | 0.142±0.004 | 0.144±0.003 | 0.144±0.003 | **0.130**±0.003 | 0.131±0.003 | 0.131±0.003 | 0.146±0.003 |
| anuran_calls_families | **0.006**±0.001 | 0.006±0.001 | 0.008±0.002 | 0.009±0.002 | 0.009±0.002 | 0.007±0.002 | 0.009±0.002 | 0.008±0.003 | 0.012±0.003 |
| anuran_calls_genus | **0.007**±0.002 | 0.008±0.002 | 0.008±0.002 | 0.011±0.002 | 0.010±0.002 | 0.008±0.002 | 0.009±0.002 | 0.009±0.002 | 0.012±0.003 |
| anuran_calls_species | **0.006**±0.001 | 0.007±0.001 | 0.010±0.002 | 0.008±0.002 | 0.009±0.002 | 0.007±0.001 | 0.008±0.002 | 0.008±0.002 | 0.010±0.001 |
| avila | **0.000**±0.000 | 0.000±0.000 | 0.001±0.000 | 0.003±0.001 | 0.016±0.002 | 0.001±0.000 | 0.001±0.001 | 0.001±0.000 | 0.011±0.001 |
| bank_marketing | 0.089±0.002 | **0.087**±0.003 | 0.088±0.002 | 0.089±0.001 | 0.091±0.001 | 0.088±0.002 | 0.090±0.002 | 0.090±0.002 | 0.091±0.002 |
| bank_marketing_additional | 0.085±0.003 | 0.084±0.002 | 0.086±0.002 | 0.085±0.002 | 0.086±0.002 | 0.084±0.003 | **0.084**±0.003 | 0.085±0.002 | 0.086±0.002 |
| chess | 0.005±0.002 | 0.013±0.003 | 0.015±0.003 | 0.010±0.004 | 0.008±0.003 | 0.008±0.003 | 0.011±0.002 | **0.005**±0.001 | 0.015±0.003 |
| chess_krvk | **0.081**±0.004 | 0.120±0.005 | 0.128±0.004 | 0.121±0.006 | 0.141±0.009 | 0.153±0.002 | 0.147±0.003 | 0.146±0.003 | 0.293±0.017 |
| crowd_sourced_mapping | 0.032±0.004 | 0.031±0.003 | **0.028**±0.001 | 0.037±0.003 | 0.034±0.004 | 0.035±0.003 | 0.031±0.003 | 0.033±0.003 | 0.058±0.004 |
| default_credit_card | 0.179±0.004 | 0.181±0.004 | 0.182±0.003 | 0.179±0.004 | 0.181±0.004 | 0.179±0.004 | **0.178**±0.004 | 0.180±0.004 | 0.183±0.003 |
| eeg_eye_state | 0.016±0.002 | **0.011**±0.001 | 0.107±0.024 | 0.176±0.014 | 0.120±0.017 | 0.051±0.002 | 0.052±0.001 | 0.051±0.002 | 0.083±0.003 |
| electrical_grid_stability_simulated | **0.032**±0.004 | 0.036±0.003 | 0.048±0.004 | 0.039±0.004 | 0.057±0.003 | 0.051±0.004 | 0.053±0.004 | 0.057±0.003 | 0.086±0.005 |
| facebook_live_sellers_thailand_status | 0.134±0.007 | 0.137±0.005 | 0.139±0.007 | 0.137±0.008 | 0.139±0.007 | **0.131**±0.005 | 0.135±0.006 | 0.133±0.008 | 0.141±0.004 |
| firm_teacher_clave | 0.129±0.006 | **0.128**±0.005 | 0.130±0.005 | 0.133±0.007 | 0.134±0.005 | 0.150±0.006 | 0.149±0.003 | 0.149±0.005 | 0.191±0.006 |
| first_order_theorem_proving | 0.179±0.006 | 0.180±0.007 | 0.182±0.007 | 0.188±0.008 | 0.181±0.006 | **0.158**±0.008 | 0.160±0.008 | 0.160±0.006 | 0.162±0.007 |
| gas_sensor_drift_class | 0.005±0.001 | 0.006±0.001 | **0.004**±0.001 | 0.006±0.001 | 0.005±0.001 | 0.006±0.001 | 0.006±0.001 | 0.006±0.001 | 0.008±0.001 |
| gesture_phase_segmentation_raw | 0.087±0.006 | 0.079±0.009 | 0.079±0.006 | 0.108±0.006 | 0.106±0.007 | 0.071±0.005 | 0.067±0.005 | **0.066**±0.004 | 0.074±0.005 |
| gesture_phase_segmentation_va3 | 0.323±0.007 | **0.289**±0.008 | 0.293±0.006 | 0.347±0.011 | 0.368±0.008 | 0.321±0.007 | 0.312±0.006 | 0.316±0.009 | 0.355±0.006 |
| htru2 | 0.020±0.002 | 0.019±0.002 | 0.020±0.001 | 0.020±0.002 | 0.020±0.002 | 0.021±0.002 | **0.019**±0.002 | 0.020±0.002 | 0.020±0.002 |
| human_activity_smartphone | 0.008±0.001 | 0.009±0.001 | 0.010±0.001 | 0.011±0.002 | 0.015±0.002 | 0.008±0.002 | 0.008±0.002 | **0.008**±0.001 | 0.024±0.002 |
| indoor_loc_building | 0.002±0.000 | 0.002±0.000 | 0.002±0.000 | 0.002±0.000 | 0.002±0.000 | 0.002±0.000 | 0.003±0.001 | 0.002±0.000 | **0.002**±0.000 |
| indoor_loc_relative | 0.070±0.002 | 0.081±0.005 | 0.090±0.002 | 0.084±0.003 | 0.099±0.004 | 0.077±0.003 | **0.059**±0.003 | 0.066±0.007 | 0.077±0.003 |
| insurance_benchmark | 0.061±0.005 | 0.060±0.004 | 0.060±0.004 | 0.060±0.004 | 0.060±0.004 | 0.061±0.005 | **0.059**±0.004 | 0.061±0.004 | 0.072±0.004 |
| landsat_satimage | **0.077**±0.006 | 0.083±0.006 | 0.094±0.006 | 0.085±0.007 | 0.090±0.005 | 0.087±0.005 | 0.080±0.004 | 0.080±0.004 | 0.090±0.004 |
| letter_recognition | 0.019±0.001 | **0.019**±0.001 | 0.020±0.001 | 0.039±0.002 | 0.030±0.002 | 0.034±0.003 | 0.032±0.002 | 0.033±0.002 | 0.045±0.002 |
| madelon | 0.340±0.016 | 0.401±0.019 | 0.443±0.011 | 0.349±0.027 | 0.435±0.019 | **0.139**±0.010 | 0.211±0.010 | 0.210±0.014 | 0.320±0.011 |
| magic_gamma_telescope | 0.115±0.005 | **0.108**±0.004 | 0.109±0.004 | 0.119±0.005 | 0.122±0.003 | 0.116±0.004 | 0.118±0.004 | 0.118±0.003 | 0.122±0.003 |
| mushroom | **0.000**±0.000 | 0.000±0.000 | 0.000±0.000 | 0.000±0.000 | 0.000±0.000 | 0.000±0.000 | 0.000±0.000 | 0.000±0.000 | 0.000±0.000 |
| musk | **0.003**±0.002 | 0.004±0.002 | 0.005±0.002 | 0.007±0.001 | 0.011±0.003 | 0.011±0.002 | 0.012±0.002 | 0.012±0.002 | 0.028±0.002 |
| nomao | 0.022±0.002 | 0.020±0.001 | 0.022±0.001 | 0.024±0.001 | 0.026±0.002 | 0.018±0.001 | **0.017**±0.001 | 0.017±0.002 | 0.020±0.002 |
| nursery | 0.020±0.001 | 0.012±0.003 | **0.011**±0.002 | 0.022±0.002 | 0.024±0.002 | 0.021±0.001 | 0.026±0.002 | 0.024±0.002 | 0.034±0.001 |
| occupancy_detection | **0.006**±0.001 | 0.007±0.001 | 0.008±0.001 | 0.009±0.001 | 0.008±0.001 | 0.007±0.001 | 0.007±0.001 | 0.007±0.000 | 0.006±0.001 |
| online_shoppers_attention | 0.098±0.004 | 0.101±0.004 | 0.098±0.004 | 0.095±0.003 | 0.099±0.004 | 0.097±0.003 | 0.096±0.005 | 0.097±0.004 | 0.096±0.003 |
| optical_recognition_handwritten_digits | **0.011**±0.003 | 0.012±0.002 | 0.015±0.003 | 0.020±0.003 | 0.018±0.003 | 0.017±0.003 | 0.015±0.003 | 0.016±0.003 | 0.020±0.004 |
| ozone_level_1hr | 0.036±0.007 | 0.037±0.008 | 0.035±0.008 | **0.035**±0.008 | 0.035±0.008 | 0.036±0.009 | 0.035±0.007 | 0.035±0.007 | 0.035±0.007 |
| ozone_level_8hr | 0.071±0.010 | 0.070±0.010 | 0.072±0.012 | 0.067±0.011 | 0.072±0.010 | 0.071±0.011 | 0.070±0.012 | 0.067±0.011 | 0.070±0.009 |
| page_blocks | 0.028±0.004 | 0.026±0.005 | 0.027±0.004 | 0.026±0.004 | 0.028±0.004 | **0.025**±0.003 | 0.025±0.003 | 0.025±0.004 | 0.026±0.003 |
| pen_recognition_handwritten_characters | **0.004**±0.001 | 0.004±0.001 | 0.007±0.001 | 0.007±0.002 | 0.008±0.002 | 0.007±0.001 | 0.007±0.001 | 0.006±0.001 | 0.011±0.001 |
| phishing | 0.031±0.002 | 0.029±0.002 | **0.029**±0.003 | 0.032±0.003 | 0.034±0.002 | 0.031±0.002 | 0.032±0.002 | 0.031±0.002 | 0.032±0.003 |
| polish_companies_bankruptcy_1year | **0.018**±0.003 | 0.019±0.001 | 0.026±0.003 | 0.026±0.003 | 0.028±0.002 | 0.021±0.002 | 0.022±0.002 | 0.021±0.002 | 0.027±0.003 |
| polish_companies_bankruptcy_2year | 0.017±0.002 | **0.017**±0.001 | 0.041±0.002 | 0.041±0.002 | 0.041±0.002 | 0.025±0.003 | 0.025±0.003 | 0.026±0.003 | 0.035±0.002 |
| polish_companies_bankruptcy_3year | **0.023**±0.002 | 0.025±0.002 | 0.041±0.004 | 0.038±0.002 | 0.046±0.003 | 0.031±0.003 | 0.033±0.003 | 0.033±0.003 | 0.040±0.003 |
| polish_companies_bankruptcy_4year | **0.029**±0.003 | 0.030±0.002 | 0.051±0.002 | 0.048±0.004 | 0.053±0.002 | 0.031±0.002 | 0.036±0.002 | 0.035±0.002 | 0.048±0.002 |
| polish_companies_bankruptcy_5year | 0.037±0.002 | 0.040±0.002 | 0.064±0.004 | 0.056±0.004 | 0.066±0.002 | **0.031**±0.004 | 0.033±0.003 | 0.036±0.004 | 0.050±0.005 |
| seismic_bumps | 0.068±0.009 | 0.065±0.009 | 0.066±0.009 | 0.065±0.009 | 0.065±0.008 | 0.070±0.007 | 0.066±0.009 | 0.067±0.007 | 0.068±0.008 |
| skill_craft | 0.589±0.010 | 0.582±0.014 | 0.601±0.012 | **0.574**±0.019 | 0.597±0.015 | 0.600±0.014 | 0.607±0.013 | 0.610±0.014 | 0.593±0.014 |
| smartphone_human_activity | 0.045±0.004 | **0.035**±0.007 | 0.042±0.004 | 0.064±0.006 | 0.071±0.004 | 0.033±0.005 | 0.035±0.005 | 0.036±0.004 | 0.079±0.004 |
| smartphone_human_activity_postural | 0.009±0.002 | 0.008±0.002 | 0.010±0.001 | 0.011±0.002 | 0.013±0.002 | **0.006**±0.001 | 0.007±0.001 | 0.007±0.001 | 0.025±0.002 |
| spambase | 0.052±0.008 | 0.059±0.005 | 0.052±0.007 | 0.051±0.006 | 0.054±0.006 | 0.048±0.008 | 0.049±0.008 | **0.048**±0.008 | 0.052±0.008 |
| superconductivity_class | 0.058±0.003 | 0.063±0.004 | 0.059±0.004 | 0.067±0.002 | 0.063±0.003 | **0.057**±0.003 | 0.058±0.003 | 0.058±0.003 | 0.059±0.002 |
| thyroid_all_bp | 0.026±0.004 | 0.027±0.003 | 0.028±0.003 | 0.027±0.006 | 0.029±0.005 | 0.024±0.004 | 0.025±0.003 | **0.022**±0.003 | 0.027±0.003 |
| thyroid_all_hyper | 0.015±0.002 | 0.016±0.003 | 0.017±0.003 | 0.014±0.002 | 0.018±0.003 | 0.014±0.003 | 0.014±0.003 | **0.014**±0.003 | 0.015±0.003 |
| thyroid_all_hypo | 0.008±0.002 | 0.012±0.002 | 0.018±0.004 | 0.013±0.002 | 0.020±0.002 | **0.003**±0.002 | 0.005±0.002 | 0.005±0.002 | 0.007±0.002 |
| thyroid_all_rep | 0.009±0.002 | 0.009±0.003 | 0.009±0.002 | 0.013±0.004 | 0.016±0.005 | 0.005±0.002 | 0.006±0.002 | **0.005**±0.002 | 0.009±0.002 |
| thyroid_ann | 0.008±0.002 | 0.009±0.002 | 0.012±0.004 | 0.006±0.002 | 0.010±0.002 | **0.003**±0.001 | 0.003±0.001 | 0.005±0.002 | 0.003±0.001 |
| thyroid_dis | 0.013±0.002 | 0.013±0.002 | 0.014±0.002 | 0.015±0.003 | 0.016±0.003 | **0.010**±0.002 | 0.011±0.002 | 0.010±0.003 | 0.013±0.002 |
| thyroid_hypo | 0.014±0.003 | 0.014±0.003 | 0.014±0.003 | 0.014±0.005 | 0.016±0.003 | **0.009**±0.004 | 0.010±0.002 | 0.012±0.003 | 0.012±0.003 |
| thyroid_sick | 0.016±0.003 | 0.020±0.003 | 0.022±0.004 | 0.019±0.004 | 0.029±0.003 | **0.011**±0.003 | 0.018±0.004 | 0.011±0.003 | 0.017±0.005 |
| thyroid_sick_eu | **0.000**±0.000 | 0.000±0.000 | 0.001±0.001 | 0.000±0.000 | 0.000±0.000 | 0.000±0.000 | 0.000±0.000 | 0.000±0.000 | 0.000±0.000 |
| turkiye_student_evaluation | 0.016±0.002 | **0.016**±0.002 | 0.017±0.003 | 0.016±0.002 | 0.033±0.005 | 0.016±0.002 | 0.016±0.002 | 0.017±0.003 | 0.113±0.005 |
| wall_follow_robot_2 | 0.002±0.001 | 0.008±0.003 | 0.009±0.002 | 0.004±0.001 | 0.008±0.002 | 0.002±0.001 | 0.004±0.002 | 0.002±0.001 | **0.001**±0.001 |
| wall_follow_robot_24 | 0.029±0.004 | 0.042±0.003 | 0.040±0.007 | 0.016±0.004 | 0.042±0.005 | **0.002**±0.001 | 0.005±0.002 | 0.004±0.001 | 0.007±0.001 |
| wall_follow_robot_4 | 0.002±0.001 | 0.009±0.003 | 0.012±0.003 | 0.007±0.003 | 0.011±0.003 | 0.002±0.001 | 0.004±0.002 | 0.003±0.001 | **0.001**±0.001 |
| waveform | 0.141±0.004 | **0.140**±0.004 | 0.147±0.008 | 0.145±0.004 | 0.145±0.006 | 0.146±0.005 | 0.149±0.006 | 0.147±0.005 | 0.148±0.005 |
| waveform_noise | **0.139**±0.006 | 0.145±0.010 | 0.161±0.014 | 0.140±0.009 | 0.143±0.007 | 0.145±0.007 | 0.146±0.009 | 0.145±0.007 | 0.150±0.008 |
| wilt | 0.012±0.003 | 0.011±0.002 | 0.011±0.002 | 0.014±0.002 | **0.011**±0.002 | 0.014±0.002 | 0.014±0.003 | 0.014±0.003 | 0.016±0.002 |
| wine_quality_all | 0.377±0.008 | 0.353±0.007 | 0.352±0.010 | 0.422±0.016 | 0.412±0.015 | 0.346±0.008 | 0.351±0.007 | 0.349±0.007 | **0.338**±0.009 |
| wine_quality_type | 0.004±0.001 | 0.004±0.002 | 0.004±0.001 | 0.005±0.002 | 0.004±0.001 | 0.004±0.001 | 0.005±0.001 | **0.003**±0.001 | 0.006±0.001 |
| wine_quality_white | 0.371±0.009 | 0.342±0.007 | 0.345±0.008 | 0.403±0.021 | 0.405±0.018 | 0.334±0.008 | 0.343±0.009 | 0.338±0.008 | **0.330**±0.008 |

Table D.2: Classification error of *tuned* methods on datasets in $\mathcal{B}_{\text{class}}^{\text{train}}$, averaged over ten train-validation-test splits. When we write $a \pm b$, $a$ is the mean error on the dataset and $[a - b, a + b]$ is an approximate 95% confidence interval for the mean in the #splits $\to \infty$ limit. The confidence interval is computed from the $t$-distribution using a normality assumption as in Appendix C.6. In each row, the lowest mean error is highlighted in bold, and errors whose confidence interval contains the lowest error are underlined.

| Dataset | RealMLP-HPO | MLP-PLR-HPO | ResNet-HPO | MLP-HPO | CatBoost-HPO | LGBM-HPO | XGB-HPO |
|---|---|---|---|---|---|---|---|
| abalone | 0.444±0.011 | 0.451±0.011 | **0.443**±0.007 | 0.446±0.013 | 0.466±0.017 | 0.463±0.013 | 0.459±0.016 |
| adult | 0.139±0.003 | 0.136±0.003 | 0.145±0.004 | 0.146±0.003 | 0.130±0.003 | **0.129**±0.003 | 0.131±0.003 |
| anuran_calls_families | **0.006**±0.001 | 0.010±0.002 | 0.006±0.001 | 0.009±0.001 | 0.008±0.002 | 0.009±0.003 | 0.011±0.002 |
| anuran_calls_genus | **0.006**±0.002 | 0.013±0.003 | 0.007±0.002 | 0.011±0.002 | 0.009±0.002 | 0.008±0.002 | 0.012±0.002 |
| anuran_calls_species | **0.007**±0.002 | 0.010±0.002 | 0.009±0.002 | 0.009±0.003 | 0.008±0.001 | 0.009±0.002 | 0.010±0.002 |
| avila | **0.001**±0.000 | 0.001±0.000 | 0.015±0.002 | 0.014±0.004 | 0.001±0.000 | 0.001±0.000 | 0.002±0.000 |
| bank_marketing | 0.088±0.002 | 0.090±0.002 | 0.091±0.002 | 0.092±0.002 | 0.088±0.002 | **0.088**±0.002 | 0.090±0.002 |
| bank_marketing_additional | 0.084±0.003 | 0.084±0.002 | 0.086±0.002 | 0.087±0.002 | 0.083±0.002 | **0.083**±0.002 | 0.084±0.002 |
| chess | 0.008±0.004 | 0.010±0.004 | **0.005**±0.002 | 0.009±0.003 | 0.006±0.002 | 0.007±0.003 | 0.011±0.003 |
| chess_krvk | **0.070**±0.004 | 0.112±0.008 | 0.106±0.006 | 0.130±0.010 | 0.137±0.005 | 0.149±0.005 | 0.194±0.008 |
| crowd_sourced_mapping | **0.031**±0.005 | 0.036±0.004 | 0.031±0.003 | 0.034±0.003 | 0.034±0.003 | 0.034±0.004 | 0.039±0.003 |
| default_credit_card | 0.180±0.004 | 0.180±0.004 | 0.180±0.003 | 0.181±0.004 | 0.179±0.003 | 0.179±0.004 | 0.180±0.004 |
| eeg_eye_state | **0.015**±0.001 | 0.102±0.011 | 0.081±0.018 | 0.113±0.011 | 0.054±0.003 | 0.050±0.003 | 0.065±0.006 |
| electrical_grid_stability_simulated | **0.029**±0.003 | 0.035±0.004 | 0.046±0.005 | 0.056±0.004 | 0.048±0.004 | 0.054±0.003 | 0.057±0.005 |
| facebook_live_sellers_thailand_status | 0.133±0.007 | 0.137±0.007 | 0.138±0.018 | 0.138±0.005 | 0.132±0.006 | 0.136±0.006 | 0.138±0.007 |
| firm_teacher_clave | **0.125**±0.006 | 0.132±0.004 | 0.128±0.006 | 0.130±0.006 | 0.144±0.007 | 0.143±0.006 | 0.141±0.004 |
| first_order_theorem_proving | 0.182±0.009 | 0.182±0.004 | 0.181±0.008 | 0.184±0.007 | 0.161±0.008 | **0.160**±0.006 | 0.164±0.009 |
| gas_sensor_drift_class | 0.005±0.001 | 0.006±0.001 | **0.004**±0.001 | 0.005±0.001 | 0.006±0.001 | 0.006±0.001 | 0.007±0.001 |
| gesture_phase_segmentation_raw | 0.086±0.004 | 0.091±0.007 | 0.103±0.006 | 0.098±0.007 | 0.071±0.004 | **0.066**±0.004 | 0.069±0.004 |
| gesture_phase_segmentation_va3 | 0.332±0.009 | 0.333±0.008 | 0.343±0.009 | 0.353±0.008 | 0.323±0.008 | **0.307**±0.005 | 0.331±0.010 |
| htru2 | 0.020±0.001 | 0.020±0.001 | 0.020±0.002 | **0.019**±0.002 | 0.020±0.002 | 0.020±0.002 | 0.020±0.002 |
| human_activity_smartphone | 0.008±0.001 | 0.012±0.003 | 0.011±0.002 | 0.014±0.003 | 0.009±0.002 | **0.007**±0.001 | 0.011±0.002 |
| indoor_loc_building | 0.002±0.000 | 0.002±0.000 | 0.002±0.000 | 0.002±0.000 | 0.002±0.000 | 0.002±0.000 | 0.002±0.000 |
| indoor_loc_relative | 0.060±0.004 | 0.062±0.003 | 0.094±0.003 | 0.095±0.003 | 0.070±0.003 | 0.056±0.002 | 0.063±0.004 |
| insurance_benchmark | 0.061±0.004 | **0.060**±0.004 | 0.061±0.005 | 0.060±0.004 | 0.062±0.005 | 0.061±0.004 | 0.061±0.004 |
| landsat_satimage | 0.079±0.005 | 0.089±0.006 | 0.090±0.005 | 0.088±0.004 | 0.084±0.004 | 0.080±0.003 | 0.088±0.007 |
| letter_recognition | **0.018**±0.001 | 0.039±0.003 | 0.024±0.003 | 0.030±0.002 | 0.033±0.002 | 0.034±0.002 | 0.044±0.002 |
| madelon | 0.194±0.019 | 0.311±0.022 | 0.414±0.015 | 0.421±0.015 | **0.158**±0.015 | 0.176±0.009 | 0.182±0.009 |
| magic_gamma_telescope | **0.115**±0.003 | 0.116±0.005 | 0.115±0.004 | 0.121±0.002 | 0.117±0.004 | 0.118±0.004 | 0.119±0.004 |
| mushroom | **0.000**±0.000 | 0.000±0.000 | 0.000±0.000 | 0.000±0.000 | 0.000±0.000 | 0.000±0.000 | **0.000**±0.000 |
| musk | **0.003**±0.002 | 0.007±0.002 | 0.009±0.002 | 0.008±0.002 | 0.010±0.003 | 0.010±0.003 | 0.017±0.003 |
| nomao | 0.021±0.001 | 0.021±0.002 | 0.024±0.002 | 0.025±0.001 | 0.018±0.001 | **0.017**±0.001 | 0.018±0.001 |
| nursery | **0.019**±0.002 | 0.020±0.002 | 0.021±0.001 | 0.022±0.002 | 0.021±0.002 | 0.021±0.002 | 0.023±0.003 |
| occupancy_detection | 0.007±0.001 | 0.007±0.001 | 0.009±0.001 | 0.009±0.001 | 0.007±0.001 | **0.006**±0.001 | 0.007±0.001 |
| online_shoppers_attention | 0.098±0.006 | **0.094**±0.003 | 0.097±0.005 | 0.098±0.006 | 0.098±0.004 | 0.098±0.005 | 0.098±0.004 |
| optical_recognition_handwritten_digits | **0.010**±0.002 | 0.021±0.005 | 0.013±0.004 | 0.018±0.003 | 0.016±0.003 | 0.015±0.004 | 0.020±0.004 |
| ozone_level_1hr | 0.035±0.008 | 0.038±0.007 | 0.036±0.007 | 0.036±0.007 | 0.035±0.008 | 0.038±0.008 | **0.035**±0.008 |
| ozone_level_8hr | 0.071±0.011 | 0.071±0.011 | 0.068±0.012 | 0.071±0.013 | 0.073±0.011 | 0.077±0.014 | 0.073±0.007 |
| page_blocks | 0.025±0.003 | 0.027±0.006 | 0.028±0.005 | 0.027±0.003 | **0.024**±0.004 | 0.026±0.003 | 0.025±0.004 |
| pen_recognition_handwritten_characters | **0.004**±0.001 | 0.007±0.002 | 0.007±0.001 | 0.008±0.001 | 0.006±0.001 | 0.006±0.001 | 0.009±0.001 |
| phishing | 0.030±0.002 | 0.033±0.003 | 0.031±0.001 | 0.033±0.002 | 0.031±0.002 | **0.030**±0.002 | 0.033±0.002 |
| polish_companies_bankruptcy_1year | **0.018**±0.003 | 0.025±0.002 | 0.025±0.003 | 0.028±0.003 | 0.021±0.002 | 0.021±0.001 | 0.022±0.002 |
| polish_companies_bankruptcy_2year | **0.016**±0.002 | 0.035±0.003 | 0.041±0.002 | 0.040±0.003 | 0.024±0.003 | 0.024±0.003 | 0.024±0.003 |
| polish_companies_bankruptcy_3year | **0.024**±0.002 | 0.037±0.004 | 0.043±0.004 | 0.041±0.005 | 0.030±0.004 | 0.031±0.004 | 0.032±0.004 |
| polish_companies_bankruptcy_4year | **0.029**±0.002 | 0.044±0.002 | 0.050±0.003 | 0.053±0.002 | 0.031±0.002 | 0.032±0.002 | 0.034±0.002 |
| polish_companies_bankruptcy_5year | 0.038±0.003 | 0.055±0.005 | 0.064±0.005 | 0.063±0.003 | **0.030**±0.004 | 0.033±0.004 | 0.036±0.004 |
| seismic_bumps | 0.071±0.010 | **0.065**±0.008 | 0.070±0.008 | 0.066±0.009 | 0.069±0.009 | 0.071±0.009 | 0.070±0.008 |
| skill_craft | 0.584±0.012 | **0.577**±0.010 | 0.599±0.011 | 0.595±0.013 | 0.587±0.017 | 0.609±0.015 | 0.602±0.015 |
| smartphone_human_activity | **0.039**±0.006 | 0.065±0.007 | 0.059±0.006 | 0.072±0.005 | 0.040±0.003 | 0.040±0.005 | 0.048±0.004 |
| smartphone_human_activity_postural | 0.007±0.002 | 0.012±0.002 | 0.011±0.002 | 0.014±0.002 | 0.007±0.001 | **0.006**±0.001 | 0.011±0.002 |
| spambase | 0.054±0.007 | 0.055±0.007 | 0.053±0.005 | 0.053±0.007 | **0.046**±0.006 | 0.051±0.008 | 0.055±0.006 |
| superconductivity_class | 0.059±0.003 | 0.060±0.002 | 0.061±0.002 | 0.061±0.002 | 0.058±0.001 | 0.058±0.003 | **0.057**±0.002 |
| thyroid_all_bp | **0.024**±0.004 | 0.026±0.005 | 0.028±0.003 | 0.029±0.005 | 0.025±0.004 | 0.025±0.004 | 0.027±0.003 |
| thyroid_all_hyper | **0.014**±0.002 | 0.015±0.002 | 0.018±0.003 | 0.018±0.004 | 0.014±0.003 | 0.015±0.002 | 0.015±0.002 |
| thyroid_all_hypo | 0.007±0.002 | 0.010±0.002 | 0.021±0.003 | 0.021±0.003 | **0.004**±0.002 | 0.006±0.002 | 0.005±0.002 |
| thyroid_all_rep | 0.010±0.003 | 0.009±0.003 | 0.014±0.003 | 0.015±0.004 | **0.005**±0.003 | 0.007±0.003 | 0.007±0.002 |
| thyroid_ann | 0.005±0.001 | 0.005±0.001 | 0.013±0.002 | 0.012±0.002 | 0.004±0.001 | 0.003±0.001 | **0.002**±0.001 |
| thyroid_dis | 0.013±0.004 | 0.014±0.003 | 0.017±0.003 | 0.016±0.003 | **0.010**±0.002 | 0.013±0.002 | 0.013±0.003 |
| thyroid_hypo | 0.014±0.002 | 0.011±0.003 | 0.016±0.003 | 0.018±0.005 | 0.011±0.003 | 0.011±0.004 | **0.009**±0.003 |
| thyroid_sick | **0.011**±0.003 | 0.018±0.004 | 0.026±0.006 | 0.025±0.004 | 0.013±0.003 | 0.015±0.004 | 0.017±0.004 |
| thyroid_sick_eu | **0.000**±0.000 | 0.000±0.000 | 0.001±0.002 | 0.000±0.000 | 0.000±0.000 | 0.000±0.000 | 0.000±0.000 |
| turkiye_student_evaluation | **0.016**±0.002 | 0.017±0.002 | 0.032±0.004 | 0.021±0.004 | 0.016±0.003 | 0.019±0.002 | 0.017±0.003 |
| wall_follow_robot_2 | 0.002±0.002 | 0.003±0.001 | 0.011±0.003 | 0.005±0.002 | 0.002±0.001 | 0.003±0.001 | **0.001**±0.001 |
| wall_follow_robot_24 | 0.011±0.004 | 0.012±0.003 | 0.041±0.006 | 0.041±0.004 | **0.003**±0.001 | 0.005±0.001 | 0.004±0.002 |
| wall_follow_robot_4 | 0.002±0.001 | 0.004±0.001 | 0.018±0.004 | 0.010±0.002 | 0.003±0.002 | 0.003±0.001 | **0.002**±0.001 |
| waveform | 0.136±0.005 | **0.136**±0.005 | 0.136±0.005 | 0.140±0.007 | 0.143±0.003 | 0.152±0.007 | 0.148±0.005 |
| waveform_noise | 0.141±0.008 | 0.141±0.006 | **0.137**±0.007 | 0.143±0.007 | 0.139±0.008 | 0.145±0.008 | 0.146±0.007 |
| wilt | 0.013±0.003 | 0.012±0.002 | **0.011**±0.002 | 0.012±0.002 | 0.014±0.003 | 0.014±0.003 | 0.015±0.003 |
| wine_quality_all | 0.367±0.011 | 0.383±0.012 | 0.388±0.014 | 0.386±0.010 | 0.351±0.009 | **0.344**±0.009 | 0.350±0.011 |
| wine_quality_type | 0.005±0.002 | 0.005±0.001 | **0.004**±0.001 | 0.004±0.001 | 0.004±0.002 | 0.004±0.001 | 0.006±0.002 |
| wine_quality_white | 0.367±0.008 | 0.382±0.011 | 0.378±0.005 | 0.375±0.010 | 0.344±0.008 | **0.338**±0.011 | 0.352±0.016 |

Table D.3: nRMSE of *untuned* methods on datasets in $\mathcal{B}_{\text{reg}}^{\text{train}}$, averaged over ten train-validation-test splits. When we write $a \pm b$, $a$ is the mean error on the dataset and $[a - b, a + b]$ is an approximate 95% confidence interval for the mean in the #splits $\to \infty$ limit. The confidence interval is computed from the $t$-distribution using a normality assumption as in Appendix C.6. In each row, the lowest mean error is highlighted in bold, and errors whose confidence interval contains the lowest error are underlined.

| Dataset | RealMLP-TD | RealTabR-D | TabR-S-D | MLP-PLR-D | MLP-D | CatBoost-TD | LGBM-TD | XGB-TD | RF-D |
|---|---|---|---|---|---|---|---|---|---|
| air_quality_bc | **0.005**±0.000 | 0.008±0.001 | 0.025±0.004 | 0.040±0.005 | 0.043±0.003 | 0.031±0.007 | 0.030±0.005 | 0.033±0.007 | 0.013±0.004 |
| air_quality_co2 | 0.305±0.018 | **0.233**±0.010 | 0.246±0.011 | 0.295±0.017 | 0.297±0.017 | 0.240±0.015 | 0.264±0.017 | 0.278±0.018 | 0.294±0.016 |
| air_quality_no2 | 0.315±0.004 | **0.263**±0.009 | 0.281±0.005 | 0.335±0.006 | 0.335±0.008 | 0.284±0.005 | 0.294±0.010 | 0.294±0.006 | 0.323±0.006 |
| air_quality_nox | 0.287±0.016 | 0.256±0.019 | 0.261±0.019 | 0.287±0.015 | 0.288±0.014 | 0.254±0.020 | **0.252**±0.019 | 0.256±0.017 | 0.257±0.012 |
| appliances_energy | 0.760±0.015 | **0.604**±0.011 | 0.651±0.014 | 0.770±0.014 | 0.796±0.011 | 0.687±0.007 | 0.679±0.007 | 0.677±0.006 | 0.706±0.008 |
| bejing_pm25 | 0.310±0.006 | **0.256**±0.008 | 0.279±0.005 | 0.389±0.010 | 0.440±0.012 | 0.394±0.008 | 0.377±0.007 | 0.378±0.006 | 0.420±0.005 |
| bike_sharing_casual | 0.282±0.006 | **0.257**±0.005 | 0.271±0.007 | 0.292±0.006 | 0.289±0.005 | 0.276±0.006 | 0.278±0.007 | 0.284±0.008 | 0.306±0.008 |
| bike_sharing_total | 0.213±0.006 | 0.215±0.007 | 0.221±0.005 | 0.225±0.006 | 0.224±0.006 | **0.207**±0.006 | 0.211±0.006 | 0.215±0.006 | 0.242±0.007 |
| carbon_nanotubes_u | 0.010±0.000 | 0.010±0.000 | 0.013±0.001 | 0.022±0.002 | 0.026±0.003 | **0.007**±0.000 | 0.009±0.000 | 0.010±0.000 | 0.011±0.000 |
| carbon_nanotubes_v | 0.010±0.000 | 0.010±0.000 | 0.014±0.001 | 0.022±0.002 | 0.028±0.005 | **0.007**±0.000 | 0.009±0.000 | 0.010±0.000 | 0.011±0.000 |
| carbon_nanotubes_w | 0.050±0.013 | 0.050±0.013 | 0.053±0.013 | 0.054±0.011 | 0.061±0.010 | 0.052±0.012 | 0.056±0.009 | 0.058±0.009 | 0.060±0.007 |
| chess_krvk | **0.095**±0.005 | 0.125±0.005 | 0.137±0.009 | 0.126±0.005 | 0.122±0.006 | 0.261±0.003 | 0.226±0.004 | 0.237±0.004 | 0.439±0.034 |
| cycle_power_plant | 0.215±0.005 | **0.167**±0.005 | 0.169±0.005 | 0.222±0.005 | 0.223±0.003 | 0.182±0.004 | 0.184±0.004 | 0.188±0.004 | 0.201±0.003 |
| electrical_grid_stability_simulated | **0.143**±0.003 | 0.149±0.003 | 0.178±0.004 | 0.166±0.003 | 0.187±0.004 | 0.204±0.004 | 0.217±0.003 | 0.251±0.003 | 0.331±0.005 |
| facebook_comment_volume | 0.622±0.045 | 0.646±0.034 | 0.637±0.041 | 0.599±0.035 | 0.641±0.025 | 0.611±0.043 | **0.596**±0.045 | 0.602±0.046 | 0.599±0.045 |
| facebook_live_sellers_thailand_shares | 0.565±0.039 | **0.483**±0.034 | 0.498±0.034 | 0.495±0.036 | 0.500±0.039 | 0.488±0.038 | 0.483±0.034 | 0.484±0.038 | 0.494±0.050 |
| five_cities_beijing_pm25 | 0.299±0.020 | **0.244**±0.009 | 0.254±0.005 | 0.356±0.015 | 0.418±0.008 | 0.354±0.006 | 0.345±0.007 | 0.358±0.008 | 0.410±0.007 |
| five_cities_chengdu_pm25 | 0.269±0.010 | **0.205**±0.006 | 0.214±0.006 | 0.327±0.004 | 0.378±0.008 | 0.315±0.004 | 0.304±0.005 | 0.301±0.006 | 0.326±0.006 |
| five_cities_guangzhou_pm25 | 0.401±0.014 | **0.317**±0.015 | 0.331±0.012 | 0.458±0.008 | 0.518±0.014 | 0.454±0.010 | 0.453±0.011 | 0.457±0.012 | 0.488±0.010 |
| five_cities_shanghai_pm25 | 0.318±0.014 | **0.229**±0.007 | 0.254±0.008 | 0.432±0.031 | 0.445±0.011 | 0.386±0.006 | 0.386±0.008 | 0.398±0.009 | 0.450±0.010 |
| five_cities_shenyang_pm25 | 0.333±0.019 | **0.283**±0.014 | 0.297±0.011 | 0.477±0.020 | 0.520±0.018 | 0.415±0.013 | 0.419±0.014 | 0.430±0.015 | 0.519±0.013 |
| gas_sensor_drift_class | 0.082±0.012 | 0.079±0.009 | **0.073**±0.010 | 0.090±0.010 | 0.079±0.008 | 0.121±0.005 | 0.128±0.008 | 0.132±0.008 | 0.139±0.008 |
| gas_sensor_drift_conc | 0.162±0.013 | 0.149±0.013 | **0.146**±0.011 | 0.171±0.015 | 0.170±0.019 | 0.168±0.015 | 0.172±0.016 | 0.173±0.013 | 0.173±0.014 |
| indoor_loc_alt | **0.099**±0.004 | 0.128±0.006 | 0.181±0.004 | 0.121±0.004 | 0.187±0.005 | 0.166±0.003 | 0.148±0.002 | 0.162±0.003 | 0.171±0.003 |
| indoor_loc_lat | **0.079**±0.004 | 0.092±0.005 | 0.110±0.004 | 0.108±0.005 | 0.112±0.004 | 0.109±0.004 | 0.097±0.004 | 0.109±0.004 | 0.105±0.004 |
| indoor_loc_long | **0.058**±0.004 | 0.072±0.004 | 0.083±0.002 | 0.079±0.006 | 0.080±0.002 | 0.084±0.003 | 0.070±0.003 | 0.085±0.003 | 0.074±0.003 |
| insurance_benchmark | 0.978±0.006 | 0.982±0.008 | 0.984±0.008 | **0.976**±0.007 | 0.982±0.007 | 0.980±0.007 | 0.985±0.002 | 0.986±0.003 | 1.078±0.013 |
| metro_interstate_traffic_volume_long | 0.465±0.003 | **0.289**±0.007 | 0.305±0.003 | 0.464±0.003 | 0.466±0.003 | 0.389±0.003 | 0.384±0.004 | 0.442±0.016 | 0.436±0.004 |
| metro_interstate_traffic_volume_short | 0.464±0.003 | **0.286**±0.004 | 0.299±0.004 | 0.461±0.004 | 0.464±0.003 | 0.384±0.003 | 0.375±0.004 | 0.385±0.004 | 0.434±0.005 |
| naval_propulsion_comp | 0.014±0.003 | **0.006**±0.001 | 0.028±0.004 | 0.086±0.004 | 0.059±0.003 | 0.060±0.002 | 0.063±0.002 | 0.064±0.003 | 0.079±0.005 |
| naval_propulsion_turb | 0.041±0.027 | **0.014**±0.001 | 0.039±0.003 | 0.109±0.010 | 0.086±0.006 | 0.096±0.006 | 0.097±0.005 | 0.097±0.005 | 0.115±0.005 |
| nursery | 0.085±0.003 | **0.079**±0.003 | 0.087±0.008 | 0.086±0.004 | 0.100±0.003 | 0.111±0.003 | 0.106±0.004 | 0.102±0.003 | 0.116±0.004 |
| online_news_popularity | 0.989±0.003 | 0.989±0.003 | 0.991±0.002 | 0.989±0.003 | **0.988**±0.003 | 1.000±0.002 | 0.998±0.003 | 0.999±0.001 | 1.035±0.023 |
| parking_birmingham | 0.292±0.004 | 0.294±0.004 | 0.298±0.004 | 0.301±0.004 | 0.303±0.004 | **0.283**±0.004 | 0.288±0.004 | 0.293±0.004 | 0.333±0.004 |
| parkinson_motor | 0.100±0.010 | **0.085**±0.009 | 0.095±0.008 | 0.197±0.026 | 0.408±0.012 | 0.182±0.011 | 0.168±0.010 | 0.164±0.006 | 0.195±0.011 |
| parkinson_total | 0.110±0.010 | **0.094**±0.009 | 0.105±0.010 | 0.210±0.014 | 0.423±0.025 | 0.180±0.009 | 0.163±0.010 | 0.158±0.010 | 0.181±0.007 |
| protein_tertiary_structure | 0.600±0.004 | **0.494**±0.003 | 0.502±0.004 | 0.602±0.004 | 0.579±0.004 | 0.581±0.002 | 0.576±0.002 | 0.576±0.002 | 0.593±0.002 |
| skill_craft | **0.627**±0.012 | 0.632±0.012 | 0.672±0.012 | 0.628±0.013 | 0.662±0.010 | 0.649±0.010 | 0.646±0.006 | 0.650±0.010 | 0.645±0.009 |
| sml2010_dining | 0.030±0.003 | **0.030**±0.002 | 0.040±0.002 | 0.084±0.006 | 0.085±0.008 | 0.074±0.002 | 0.091±0.002 | 0.101±0.003 | 0.132±0.003 |
| sml2010_room | **0.029**±0.002 | 0.030±0.002 | 0.041±0.004 | 0.079±0.004 | 0.082±0.005 | 0.076±0.003 | 0.089±0.004 | 0.098±0.004 | 0.129±0.004 |
| superconductivity | 0.293±0.007 | 0.295±0.006 | 0.294±0.008 | 0.309±0.008 | 0.300±0.006 | **0.281**±0.004 | 0.282±0.005 | 0.281±0.005 | 0.287±0.004 |
| travel_review_ratings | 0.518±0.018 | 0.528±0.012 | 0.523±0.013 | 0.519±0.011 | 0.530±0.013 | 0.483±0.013 | **0.480**±0.014 | 0.486±0.015 | 0.485±0.011 |
| wall_follow_robot_2 | 0.037±0.010 | 0.101±0.016 | 0.109±0.013 | 0.088±0.013 | 0.090±0.017 | 0.059±0.020 | 0.069±0.026 | 0.054±0.026 | **0.027**±0.020 |
| wall_follow_robot_24 | 0.199±0.018 | 0.313±0.013 | 0.307±0.025 | 0.172±0.025 | 0.303±0.017 | 0.103±0.014 | **0.090**±0.021 | 0.094±0.018 | 0.095±0.018 |
| wall_follow_robot_4 | 0.057±0.025 | 0.115±0.024 | 0.136±0.021 | 0.089±0.013 | 0.141±0.017 | 0.065±0.024 | 0.067±0.029 | 0.053±0.024 | **0.027**±0.020 |
| wine_quality_all | 0.765±0.008 | 0.734±0.010 | 0.732±0.011 | 0.777±0.008 | 0.777±0.010 | 0.712±0.011 | **0.710**±0.012 | 0.713±0.012 | 0.717±0.012 |
| wine_quality_white | 0.758±0.021 | 0.729±0.012 | 0.728±0.014 | 0.782±0.012 | 0.774±0.011 | 0.710±0.013 | **0.709**±0.014 | 0.710±0.014 | 0.714±0.011 |

Table D.4: nRMSE of *tuned* methods on datasets in $\mathcal{B}_{\text{reg}}^{\text{train}}$, averaged over ten train-validation-test splits. When we write $a \pm b$, $a$ is the mean error on the dataset and $[a - b, a + b]$ is an approximate 95% confidence interval for the mean in the #splits $\to \infty$ limit. The confidence interval is computed from the $t$-distribution using a normality assumption as in Appendix C.6. In each row, the lowest mean error is highlighted in bold, and errors whose confidence interval contains the lowest error are underlined.

| Dataset | RealMLP-HPO | MLP-PLR-HPO | ResNet-HPO | MLP-HPO | CatBoost-HPO | LGBM-HPO | XGB-HPO |
|---|---|---|---|---|---|---|---|
| air_quality_bc | **0.004**±0.000 | 0.012±0.003 | 0.039±0.003 | 0.026±0.006 | 0.029±0.004 | 0.029±0.004 | 0.026±0.006 |
| air_quality_co2 | 0.288±0.017 | 0.284±0.013 | 0.293±0.016 | 0.298±0.014 | **0.241**±0.013 | 0.247±0.016 | 0.245±0.014 |
| air_quality_no2 | 0.311±0.006 | 0.325±0.005 | 0.322±0.006 | 0.333±0.006 | **0.286**±0.003 | 0.291±0.007 | 0.287±0.004 |
| air_quality_nox | 0.280±0.016 | 0.287±0.013 | 0.283±0.014 | 0.281±0.015 | 0.244±0.015 | 0.252±0.015 | **0.239**±0.009 |
| appliances_energy | 0.724±0.019 | 0.715±0.011 | 0.778±0.009 | 0.791±0.009 | 0.702±0.006 | **0.674**±0.009 | 0.678±0.008 |
| bejing_pm25 | **0.309**±0.007 | 0.343±0.009 | 0.393±0.005 | 0.423±0.006 | 0.433±0.007 | 0.370±0.010 | 0.386±0.007 |
| bike_sharing_casual | **0.272**±0.006 | 0.284±0.007 | 0.287±0.008 | 0.288±0.008 | 0.283±0.006 | 0.280±0.008 | 0.284±0.006 |
| bike_sharing_total | **0.209**±0.007 | 0.217±0.007 | 0.259±0.006 | 0.228±0.004 | 0.217±0.006 | 0.213±0.006 | 0.215±0.005 |
| carbon_nanotubes_u | **0.007**±0.001 | 0.010±0.003 | 0.023±0.002 | 0.011±0.000 | 0.009±0.001 | 0.009±0.001 | 0.015±0.002 |
| carbon_nanotubes_v | **0.007**±0.001 | 0.011±0.004 | 0.022±0.001 | 0.011±0.000 | 0.009±0.000 | 0.009±0.001 | 0.014±0.001 |
| carbon_nanotubes_w | 0.049±0.014 | 0.049±0.013 | 0.053±0.012 | 0.051±0.012 | 0.051±0.012 | 0.050±0.012 | 0.052±0.012 |
| chess_krvk | **0.090**±0.005 | 0.117±0.005 | 0.135±0.004 | 0.109±0.005 | 0.340±0.002 | 0.266±0.024 | 0.410±0.031 |
| cycle_power_plant | 0.207±0.004 | 0.211±0.004 | 0.220±0.003 | 0.212±0.005 | 0.184±0.004 | **0.182**±0.005 | 0.186±0.007 |
| electrical_grid_stability_simulated | **0.144**±0.003 | 0.152±0.009 | 0.171±0.003 | 0.184±0.003 | 0.194±0.003 | 0.209±0.004 | 0.226±0.006 |
| facebook_comment_volume | 0.626±0.045 | 0.619±0.049 | 0.634±0.027 | 0.644±0.029 | **0.591**±0.037 | 0.607±0.039 | 0.598±0.045 |
| facebook_live_sellers_thailand_shares | 0.566±0.049 | 0.521±0.056 | 0.492±0.036 | 0.505±0.040 | 0.494±0.056 | 0.485±0.048 | **0.469**±0.043 |
| five_cities_beijing_pm25 | **0.277**±0.009 | 0.330±0.009 | 0.379±0.005 | 0.421±0.008 | 0.380±0.008 | 0.358±0.009 | 0.367±0.007 |
| five_cities_chengdu_pm25 | **0.261**±0.006 | 0.299±0.011 | 0.342±0.007 | 0.374±0.006 | 0.348±0.006 | 0.301±0.009 | 0.321±0.014 |
| five_cities_guangzhou_pm25 | **0.392**±0.013 | 0.441±0.011 | 0.502±0.011 | 0.519±0.008 | 0.498±0.008 | 0.458±0.014 | 0.474±0.012 |
| five_cities_shanghai_pm25 | **0.306**±0.012 | 0.361±0.011 | 0.397±0.011 | 0.415±0.012 | 0.438±0.008 | 0.397±0.012 | 0.398±0.014 |
| five_cities_shenyang_pm25 | **0.330**±0.022 | 0.400±0.017 | 0.469±0.016 | 0.507±0.019 | 0.452±0.012 | 0.427±0.015 | 0.442±0.016 |
| gas_sensor_drift_class | 0.079±0.010 | 0.087±0.009 | **0.073**±0.009 | 0.078±0.009 | 0.120±0.005 | 0.120±0.007 | 0.120±0.007 |
| gas_sensor_drift_conc | **0.147**±0.014 | 0.165±0.012 | 0.150±0.012 | 0.150±0.011 | 0.165±0.013 | 0.169±0.013 | 0.163±0.014 |
| indoor_loc_alt | **0.100**±0.004 | 0.105±0.006 | 0.172±0.004 | 0.185±0.004 | 0.181±0.004 | 0.137±0.003 | 0.159±0.006 |
| indoor_loc_lat | **0.079**±0.004 | 0.086±0.004 | 0.104±0.004 | 0.106±0.004 | 0.122±0.004 | 0.091±0.004 | 0.106±0.007 |
| indoor_loc_long | **0.060**±0.004 | 0.066±0.005 | 0.074±0.003 | 0.077±0.004 | 0.097±0.003 | 0.068±0.003 | 0.084±0.006 |
| insurance_benchmark | 0.977±0.008 | 0.979±0.008 | 0.982±0.006 | 0.980±0.007 | 0.976±0.007 | **0.972**±0.006 | 0.974±0.007 |
| metro_interstate_traffic_volume_long | 0.459±0.003 | 0.418±0.006 | 0.467±0.004 | 0.465±0.003 | 0.397±0.004 | **0.391**±0.008 | 0.392±0.008 |
| metro_interstate_traffic_volume_short | 0.457±0.004 | 0.418±0.007 | 0.466±0.003 | 0.465±0.003 | 0.393±0.004 | 0.387±0.010 | **0.386**±0.009 |
| naval_propulsion_comp | **0.005**±0.001 | 0.033±0.003 | 0.059±0.003 | 0.036±0.003 | 0.062±0.002 | 0.058±0.003 | 0.062±0.004 |
| naval_propulsion_turb | **0.014**±0.001 | 0.047±0.007 | 0.078±0.005 | 0.054±0.003 | 0.095±0.002 | 0.091±0.007 | 0.096±0.006 |
| nursery | **0.080**±0.004 | 0.080±0.002 | 0.083±0.002 | 0.086±0.004 | 0.125±0.003 | 0.113±0.005 | 0.120±0.006 |
| online_news_popularity | 0.997±0.008 | 0.993±0.014 | 0.990±0.004 | 0.990±0.004 | 0.990±0.003 | **0.988**±0.004 | 0.990±0.002 |
| parking_birmingham | 0.292±0.005 | 0.283±0.009 | 0.299±0.005 | 0.301±0.005 | 0.284±0.004 | 0.286±0.004 | **0.279**±0.004 |
| parkinson_motor | **0.098**±0.015 | 0.165±0.015 | 0.372±0.018 | 0.389±0.019 | 0.219±0.011 | 0.187±0.011 | 0.214±0.025 |
| parkinson_total | **0.114**±0.013 | 0.171±0.017 | 0.388±0.020 | 0.387±0.028 | 0.219±0.010 | 0.177±0.011 | 0.207±0.023 |
| protein_tertiary_structure | 0.567±0.003 | 0.591±0.007 | **0.566**±0.004 | 0.577±0.005 | 0.608±0.002 | 0.568±0.003 | 0.590±0.009 |
| skill_craft | **0.625**±0.011 | 0.627±0.014 | 0.663±0.014 | 0.662±0.009 | 0.627±0.010 | 0.633±0.015 | 0.635±0.011 |
| sml2010_dining | **0.029**±0.001 | 0.052±0.004 | 0.065±0.003 | 0.066±0.005 | 0.076±0.003 | 0.085±0.004 | 0.089±0.006 |
| sml2010_room | **0.029**±0.002 | 0.054±0.006 | 0.065±0.004 | 0.064±0.003 | 0.075±0.002 | 0.083±0.003 | 0.087±0.006 |
| superconductivity | 0.288±0.007 | 0.296±0.004 | 0.293±0.008 | 0.294±0.008 | 0.286±0.006 | **0.278**±0.007 | 0.281±0.005 |
| travel_review_ratings | 0.499±0.015 | 0.501±0.020 | 0.529±0.015 | 0.531±0.016 | 0.475±0.012 | 0.463±0.012 | **0.460**±0.011 |
| wall_follow_robot_2 | **0.044**±0.024 | 0.051±0.022 | 0.194±0.012 | 0.092±0.017 | 0.060±0.020 | 0.066±0.025 | 0.211±0.005 |
| wall_follow_robot_24 | 0.167±0.020 | 0.136±0.029 | 0.302±0.018 | 0.306±0.021 | 0.095±0.016 | 0.097±0.018 | **0.080**±0.016 |
| wall_follow_robot_4 | 0.047±0.026 | 0.059±0.021 | 0.213±0.012 | 0.126±0.014 | 0.055±0.019 | 0.065±0.024 | **0.043**±0.016 |
| wine_quality_all | 0.751±0.011 | 0.771±0.010 | 0.771±0.013 | 0.773±0.007 | 0.727±0.009 | **0.703**±0.012 | 0.707±0.010 |
| wine_quality_white | 0.736±0.011 | 0.775±0.015 | 0.768±0.011 | 0.775±0.012 | 0.722±0.013 | **0.704**±0.012 | 0.710±0.013 |

Table D.5: Classification error of *untuned* methods on datasets in $\mathcal{B}_{\text{class}}^{\text{test}}$, averaged over ten train-validation-test splits. When we write $a \pm b$, $a$ is the mean error on the dataset and $[a - b, a + b]$ is an approximate 95% confidence interval for the mean in the #splits $\to \infty$ limit. The confidence interval is computed from the $t$-distribution using a normality assumption as in Appendix C.6. In each row, the lowest mean error is highlighted in bold, and errors whose confidence interval contains the lowest error are underlined.

| Dataset | RealMLP-TD | RealTabR-D | TabR-S-D | MLP-PLR-D | MLP-D | CatBoost-TD | LGBM-TD | XGB-TD | RF-D |
|---|---|---|---|---|---|---|---|---|---|
| ada | 0.148±0.012 | 0.146±0.013 | 0.151±0.009 | 0.146±0.013 | 0.149±0.011 | **0.139**±0.008 | 0.141±0.011 | 0.140±0.011 | 0.144±0.007 |
| airlines | 0.335±0.001 | **0.331**±0.001 | 0.332±0.001 | 0.337±0.001 | 0.337±0.001 | 0.332±0.001 | 0.333±0.001 | 0.337±0.001 | 0.382±0.001 |
| amazon-commerce-reviews | 0.209±0.021 | 0.242±0.016 | 0.402±0.028 | 0.576±0.102 | 0.372±0.019 | **0.197**±0.017 | 0.285±0.020 | 0.290±0.020 | 0.407±0.022 |
| Bioresponse | 0.238±0.010 | 0.228±0.013 | 0.228±0.013 | 0.236±0.009 | 0.232±0.006 | 0.205±0.010 | **0.204**±0.004 | 0.211±0.007 | 0.205±0.009 |
| car | **0.008**±0.006 | 0.013±0.009 | 0.011±0.006 | 0.013±0.006 | 0.011±0.006 | 0.019±0.009 | 0.021±0.006 | 0.019±0.004 | 0.071±0.017 |
| christine | 0.293±0.012 | 0.293±0.009 | 0.289±0.007 | 0.271±0.011 | 0.284±0.015 | 0.269±0.012 | **0.266**±0.012 | 0.273±0.015 | 0.281±0.013 |
| churn | **0.044**±0.003 | 0.050±0.005 | 0.055±0.006 | 0.046±0.004 | 0.065±0.007 | 0.050±0.004 | 0.048±0.005 | 0.048±0.004 | 0.064±0.005 |
| cmc | 0.465±0.019 | 0.457±0.019 | 0.449±0.025 | 0.452±0.014 | **0.441**±0.015 | 0.460±0.017 | 0.457±0.017 | 0.467±0.018 | 0.472±0.014 |
| cnae-9 | 0.068±0.010 | 0.055±0.008 | 0.066±0.011 | 0.065±0.008 | **0.053**±0.011 | 0.076±0.010 | 0.308±0.017 | 0.091±0.012 | 0.087±0.012 |
| connect-4 | **0.130**±0.003 | 0.135±0.003 | 0.135±0.003 | 0.149±0.002 | 0.149±0.002 | 0.143±0.003 | 0.136±0.003 | 0.142±0.003 | 0.181±0.003 |
| covertype | 0.029±0.001 | **0.026**±0.000 | 0.029±0.001 | 0.056±0.002 | 0.069±0.002 | 0.105±0.000 | 0.058±0.001 | 0.072±0.000 | 0.055±0.001 |
| credit-g | 0.257±0.017 | 0.250±0.015 | 0.252±0.025 | 0.256±0.023 | 0.269±0.017 | **0.250**±0.018 | 0.252±0.019 | 0.255±0.013 | 0.256±0.025 |
| Diabetes130US | 0.402±0.003 | 0.399±0.003 | 0.401±0.003 | 0.396±0.002 | 0.400±0.003 | **0.383**±0.002 | 0.398±0.002 | 0.455±0.005 | 0.398±0.002 |
| dilbert | **0.010**±0.002 | 0.014±0.002 | 0.020±0.002 | 0.019±0.003 | 0.024±0.003 | 0.013±0.002 | 0.013±0.002 | 0.012±0.002 | 0.039±0.004 |
| dionis | **0.089**±0.002 | 0.093±0.001 | 0.099±0.002 | 0.129±0.002 | 0.114±0.001 | 0.199±0.008 | 0.128±0.023 | 0.435±0.003 | 0.123±0.002 |
| dna | 0.044±0.004 | 0.050±0.005 | 0.063±0.007 | 0.056±0.005 | 0.056±0.006 | 0.046±0.003 | **0.040**±0.004 | 0.041±0.003 | 0.050±0.005 |
| fabert | 0.312±0.009 | 0.314±0.008 | 0.354±0.009 | 0.367±0.010 | 0.370±0.009 | **0.285**±0.006 | 0.386±0.008 | 0.299±0.006 | 0.317±0.008 |
| Fashion-MNIST | 0.097±0.001 | 0.101±0.002 | 0.106±0.002 | 0.115±0.002 | 0.109±0.002 | 0.099±0.001 | **0.091**±0.001 | 0.092±0.002 | 0.122±0.002 |
| gina | 0.053±0.005 | 0.060±0.005 | 0.080±0.006 | 0.079±0.008 | 0.090±0.005 | **0.047**±0.005 | 0.053±0.005 | 0.061±0.005 | 0.077±0.008 |
| guillermo | 0.175±0.004 | 0.219±0.006 | 0.271±0.010 | 0.210±0.007 | 0.243±0.006 | **0.165**±0.004 | 0.171±0.004 | 0.179±0.004 | 0.197±0.005 |
| helena | 0.617±0.002 | **0.599**±0.003 | 0.602±0.002 | 0.634±0.002 | 0.623±0.002 | 0.631±0.003 | 0.638±0.003 | 0.718±0.003 | 0.647±0.002 |
| Higgs | 0.250±0.001 | **0.248**±0.001 | 0.255±0.001 | 0.261±0.002 | 0.253±0.001 | 0.257±0.001 | 0.259±0.001 | 0.260±0.001 | 0.271±0.001 |
| Internet-Advertisements | 0.024±0.003 | 0.026±0.004 | 0.026±0.004 | 0.026±0.005 | 0.026±0.005 | 0.024±0.005 | 0.025±0.003 | 0.025±0.005 | **0.020**±0.003 |
| jannis | 0.273±0.002 | **0.262**±0.003 | 0.271±0.002 | 0.276±0.002 | 0.291±0.002 | 0.282±0.002 | 0.282±0.002 | 0.285±0.002 | 0.302±0.002 |
| jasmine | 0.207±0.014 | 0.201±0.011 | 0.206±0.012 | 0.197±0.011 | 0.207±0.012 | **0.187**±0.011 | 0.190±0.011 | 0.195±0.012 | 0.189±0.008 |
| jungle_chess_2pcs_raw_endgame_complete | **0.004**±0.001 | 0.014±0.003 | 0.098±0.010 | 0.009±0.001 | 0.107±0.003 | 0.133±0.002 | 0.134±0.003 | 0.136±0.002 | 0.204±0.002 |
| kc1 | 0.140±0.007 | **0.139**±0.007 | 0.143±0.009 | 0.142±0.007 | 0.145±0.011 | 0.147±0.010 | 0.143±0.007 | 0.144±0.010 | 0.141±0.007 |
| KDDCup99 | 0.000±0.000 | 0.000±0.000 | 0.000±0.000 | 0.000±0.000 | 0.000±0.000 | **0.000**±0.000 | 0.002±0.000 | 0.000±0.000 | 0.000±0.000 |
| kick | 0.099±0.001 | 0.099±0.001 | 0.100±0.001 | 0.098±0.001 | 0.098±0.001 | **0.096**±0.001 | 0.097±0.001 | 0.138±0.008 | 0.098±0.001 |
| madeline | 0.258±0.013 | 0.269±0.021 | 0.425±0.022 | 0.261±0.015 | 0.413±0.012 | **0.136**±0.008 | 0.198±0.011 | 0.195±0.018 | 0.262±0.008 |
| mfeat-factors | **0.016**±0.004 | 0.023±0.004 | 0.024±0.005 | 0.026±0.004 | 0.029±0.005 | 0.021±0.004 | 0.028±0.004 | 0.030±0.005 | 0.031±0.006 |
| MiniBooNE | **0.050**±0.001 | 0.051±0.001 | 0.050±0.001 | 0.053±0.001 | 0.052±0.001 | 0.053±0.001 | 0.053±0.001 | 0.055±0.002 | 0.065±0.001 |
| numerai28.6 | 0.479±0.004 | **0.414**±0.003 | 0.421±0.002 | 0.481±0.002 | 0.480±0.002 | 0.480±0.003 | 0.481±0.003 | 0.483±0.004 | 0.489±0.003 |
| okcupid-stem | 0.253±0.004 | 0.246±0.003 | 0.247±0.003 | 0.248±0.004 | 0.249±0.004 | **0.243**±0.003 | 0.246±0.003 | 0.410±0.016 | 0.262±0.003 |
| pc4 | 0.095±0.012 | 0.105±0.014 | 0.101±0.019 | 0.098±0.009 | **0.094**±0.008 | 0.099±0.012 | 0.099±0.013 | 0.100±0.013 | 0.104±0.012 |
| philippine | 0.284±0.011 | 0.268±0.009 | 0.305±0.012 | 0.271±0.008 | 0.301±0.008 | **0.249**±0.012 | 0.251±0.011 | 0.253±0.009 | 0.254±0.010 |
| phoneme | 0.097±0.007 | 0.100±0.007 | 0.101±0.007 | 0.112±0.008 | 0.120±0.013 | **0.097**±0.007 | 0.100±0.007 | 0.102±0.007 | 0.098±0.006 |
| porto-seguro | 0.038±0.000 | 0.038±0.000 | 0.038±0.000 | **0.038**±0.000 | 0.038±0.000 | 0.038±0.000 | 0.038±0.000 | 0.038±0.000 | 0.038±0.000 |
| qsar-biodeg | 0.126±0.015 | 0.125±0.021 | 0.133±0.013 | 0.139±0.017 | **0.121**±0.016 | 0.139±0.014 | 0.137±0.016 | 0.131±0.012 | 0.136±0.016 |
| riccardo | **0.002**±0.000 | 0.002±0.001 | 0.004±0.001 | 0.011±0.001 | 0.006±0.001 | 0.003±0.001 | 0.003±0.001 | 0.003±0.001 | 0.048±0.003 |
| robert | 0.488±0.008 | 0.522±0.006 | 0.574±0.004 | 0.544±0.023 | 0.579±0.007 | 0.487±0.006 | **0.464**±0.006 | 0.471±0.008 | 0.570±0.008 |
| Satellite | 0.006±0.002 | 0.006±0.002 | 0.007±0.002 | 0.006±0.002 | 0.006±0.002 | 0.006±0.001 | **0.005**±0.002 | 0.005±0.002 | 0.006±0.002 |
| segment | 0.077±0.009 | 0.074±0.010 | 0.071±0.007 | 0.080±0.008 | 0.082±0.008 | **0.069**±0.008 | 0.071±0.006 | 0.070±0.006 | 0.072±0.006 |
| shuttle | 0.000±0.000 | 0.001±0.000 | 0.001±0.000 | 0.000±0.000 | 0.001±0.000 | 0.000±0.000 | 0.002±0.001 | **0.000**±0.000 | 0.000±0.000 |
| steel-plates-fault | 0.241±0.019 | 0.225±0.015 | 0.232±0.011 | 0.227±0.017 | 0.250±0.014 | 0.223±0.014 | 0.223±0.011 | **0.220**±0.011 | 0.241±0.013 |
| sylvine | 0.054±0.005 | **0.035**±0.006 | 0.060±0.007 | 0.058±0.006 | 0.075±0.006 | 0.052±0.005 | 0.057±0.006 | 0.058±0.005 | 0.067±0.005 |
| volkert | 0.282±0.003 | 0.228±0.004 | **0.223**±0.003 | 0.300±0.004 | 0.271±0.003 | 0.299±0.002 | 0.291±0.002 | 0.296±0.002 | 0.341±0.002 |
| yeast | 0.403±0.021 | 0.396±0.015 | 0.404±0.019 | 0.404±0.020 | 0.411±0.019 | 0.411±0.019 | 0.401±0.017 | 0.409±0.024 | **0.391**±0.017 |

Table D.6: Classification error of *tuned* methods on datasets in $\mathcal{B}_{\text{class}}^{\text{test}}$, averaged over ten train-validation-test splits. When we write $a \pm b$, $a$ is the mean error on the dataset and $[a - b, a + b]$ is an approximate 95% confidence interval for the mean in the #splits $\to \infty$ limit. The confidence interval is computed from the $t$-distribution using a normality assumption as in Appendix C.6. In each row, the lowest mean error is highlighted in bold, and errors whose confidence interval contains the lowest error are underlined.

| Dataset | RealMLP-HPO | MLP-PLR-HPO | ResNet-HPO | MLP-HPO | CatBoost-HPO | LGBM-HPO | XGB-HPO |
|---|---|---|---|---|---|---|---|
| ada | 0.147±0.008 | 0.140±0.008 | 0.148±0.012 | 0.150±0.010 | **0.138±0.012** | 0.141±0.013 | 0.140±0.012 |
| airlines | 0.334±0.001 | 0.334±0.001 | 0.334±0.001 | 0.334±0.001 | 0.331±0.001 | **0.329±0.001** | 0.329±0.001 |
| amazon-commerce-reviews | **0.207±0.022** | 0.437±0.068 | 0.280±0.018 | 0.336±0.032 | 0.216±0.015 | 0.264±0.022 | 0.300±0.021 |
| Bioresponse | 0.219±0.010 | 0.233±0.010 | 0.225±0.008 | 0.229±0.011 | 0.209±0.010 | 0.206±0.012 | **0.204±0.011** |
| car | **0.004±0.003** | 0.013±0.007 | 0.016±0.013 | 0.012±0.007 | 0.017±0.008 | 0.017±0.010 | 0.022±0.007 |
| christine | 0.280±0.014 | 0.274±0.011 | 0.284±0.010 | 0.277±0.012 | 0.270±0.009 | **0.268±0.012** | 0.270±0.013 |
| churn | **0.042±0.003** | 0.045±0.003 | 0.059±0.007 | 0.054±0.006 | 0.048±0.004 | 0.048±0.005 | 0.048±0.005 |
| cmc | 0.472±0.022 | 0.456±0.034 | 0.450±0.027 | **0.447±0.020** | 0.471±0.021 | 0.470±0.016 | 0.454±0.018 |
| cnae-9 | 0.079±0.020 | 0.066±0.010 | 0.064±0.012 | **0.053±0.011** | 0.066±0.009 | 0.079±0.013 | 0.095±0.015 |
| connect-4 | **0.132±0.002** | 0.143±0.003 | 0.136±0.002 | 0.141±0.003 | 0.139±0.003 | 0.136±0.002 | 0.145±0.001 |
| covertype | **0.028±0.001** | 0.036±0.001 | 0.038±0.001 | 0.040±0.001 | 0.062±0.001 | 0.033±0.001 | 0.040±0.003 |
| credit-g | 0.262±0.023 | 0.276±0.022 | 0.272±0.028 | 0.271±0.018 | **0.234±0.024** | 0.268±0.027 | 0.248±0.017 |
| Diabetes130US | 0.395±0.002 | 0.392±0.002 | 0.398±0.003 | 0.401±0.003 | **0.384±0.003** | 0.390±0.002 | 0.387±0.002 |
| dilbert | **0.007±0.001** | 0.019±0.003 | 0.016±0.002 | 0.026±0.002 | 0.014±0.002 | 0.014±0.002 | 0.022±0.004 |
| dionis | **0.088±0.001** | 0.126±0.009 | 0.090±0.001 | 0.108±0.005 | 0.104±0.002 | 0.109±0.003 | 0.122±0.003 |
| dna | 0.043±0.005 | 0.056±0.008 | 0.046±0.003 | 0.054±0.006 | 0.043±0.004 | **0.040±0.003** | 0.040±0.003 |
| fabert | 0.309±0.006 | 0.343±0.014 | 0.363±0.011 | 0.367±0.006 | **0.286±0.006** | 0.298±0.007 | 0.303±0.007 |
| Fashion-MNIST | 0.093±0.003 | 0.107±0.010 | 0.103±0.002 | 0.105±0.002 | 0.097±0.002 | **0.091±0.001** | 0.094±0.002 |
| gina | **0.046±0.006** | 0.077±0.006 | 0.073±0.006 | 0.086±0.006 | 0.053±0.005 | 0.050±0.005 | 0.058±0.005 |
| guillermo | **0.165±0.002** | 0.202±0.006 | 0.228±0.006 | 0.242±0.005 | 0.170±0.002 | 0.167±0.002 | 0.169±0.003 |
| helena | 0.614±0.003 | 0.627±0.005 | **0.603±0.003** | 0.620±0.006 | 0.622±0.002 | 0.624±0.003 | 0.626±0.002 |
| Higgs | 0.247±0.002 | 0.252±0.001 | **0.244±0.001** | 0.252±0.001 | 0.258±0.001 | 0.255±0.001 | 0.257±0.001 |
| Internet-Advertisements | 0.024±0.002 | **0.021±0.004** | 0.024±0.004 | 0.025±0.004 | 0.025±0.005 | 0.025±0.004 | 0.026±0.004 |
| jannis | **0.269±0.002** | 0.278±0.004 | 0.279±0.003 | 0.287±0.002 | 0.281±0.002 | 0.278±0.002 | 0.279±0.002 |
| jasmine | 0.213±0.012 | 0.205±0.014 | 0.208±0.011 | 0.218±0.011 | 0.202±0.011 | 0.196±0.016 | **0.188±0.006** |
| jungle_chess_2pcs_raw_endgame_complete | **0.003±0.001** | 0.008±0.001 | 0.115±0.005 | 0.032±0.005 | 0.133±0.002 | 0.133±0.003 | 0.134±0.002 |
| kc1 | 0.143±0.010 | 0.153±0.010 | **0.139±0.006** | 0.142±0.005 | 0.142±0.008 | 0.143±0.010 | 0.144±0.005 |
| KDDCup99 | 0.000±0.000 | 0.000±0.000 | 0.000±0.000 | 0.000±0.000 | **0.000±0.000** | 0.000±0.000 | 0.000±0.000 |
| kick | 0.098±0.001 | 0.098±0.001 | 0.097±0.001 | 0.098±0.001 | **0.096±0.001** | 0.096±0.001 | 0.097±0.001 |
| madeline | 0.166±0.012 | 0.215±0.018 | 0.411±0.011 | 0.406±0.016 | **0.150±0.013** | 0.153±0.010 | 0.162±0.014 |
| mfeat-factors | **0.015±0.003** | 0.029±0.005 | 0.019±0.003 | 0.030±0.006 | 0.022±0.004 | 0.029±0.005 | 0.034±0.006 |
| MiniBooNE | **0.049±0.001** | 0.051±0.001 | 0.049±0.001 | 0.051±0.001 | 0.053±0.001 | 0.052±0.001 | 0.053±0.001 |
| numerai28.6 | **0.479±0.004** | 0.479±0.003 | 0.481±0.003 | 0.480±0.003 | 0.480±0.003 | 0.479±0.004 | 0.481±0.002 |
| okcupid-stem | 0.250±0.004 | 0.247±0.003 | 0.248±0.003 | 0.247±0.003 | **0.242±0.003** | 0.245±0.003 | 0.254±0.004 |
| pc4 | 0.103±0.009 | 0.111±0.017 | **0.093±0.011** | 0.103±0.009 | 0.096±0.012 | 0.103±0.015 | 0.104±0.014 |
| philippine | 0.273±0.015 | 0.272±0.014 | 0.301±0.010 | 0.296±0.010 | 0.250±0.008 | **0.241±0.010** | 0.245±0.007 |
| phoneme | **0.098±0.007** | 0.099±0.006 | 0.116±0.009 | 0.107±0.007 | 0.102±0.004 | 0.102±0.008 | 0.112±0.006 |
| porto-seguro | 0.038±0.000 | **0.038±0.000** | 0.038±0.000 | 0.038±0.000 | 0.038±0.000 | 0.038±0.000 | 0.038±0.000 |
| qsar-biodeg | 0.129±0.014 | 0.134±0.014 | 0.126±0.016 | **0.122±0.016** | 0.135±0.011 | 0.136±0.012 | 0.134±0.020 |
| riccardo | **0.002±0.001** | 0.002±0.000 | 0.005±0.001 | 0.005±0.001 | 0.003±0.001 | 0.003±0.001 | 0.003±0.001 |
| robert | 0.478±0.007 | 0.496±0.007 | 0.544±0.006 | 0.555±0.008 | 0.486±0.008 | **0.467±0.005** | 0.475±0.008 |
| Satellite | 0.006±0.001 | 0.007±0.002 | 0.007±0.002 | 0.007±0.001 | 0.007±0.002 | 0.006±0.002 | **0.005±0.002** |
| segment | 0.079±0.007 | 0.081±0.008 | 0.077±0.007 | 0.082±0.009 | **0.070±0.007** | 0.072±0.008 | 0.073±0.006 |
| shuttle | **0.000±0.000** | 0.000±0.000 | 0.001±0.000 | 0.001±0.000 | 0.000±0.000 | 0.000±0.000 | 0.000±0.000 |
| steel-plates-fault | 0.239±0.014 | 0.244±0.010 | 0.247±0.012 | 0.248±0.014 | **0.222±0.012** | 0.223±0.008 | 0.232±0.017 |
| sylvine | **0.053±0.005** | 0.058±0.005 | 0.074±0.004 | 0.073±0.005 | 0.055±0.005 | **0.053±0.004** | 0.058±0.003 |
| volkert | 0.272±0.004 | 0.288±0.006 | **0.235±0.003** | 0.256±0.004 | 0.301±0.003 | 0.285±0.004 | 0.290±0.003 |
| yeast | 0.407±0.020 | 0.408±0.014 | 0.399±0.026 | 0.410±0.020 | 0.393±0.020 | 0.404±0.022 | **0.391±0.017** |

Table D.7: nRMSE of *untuned* methods on datasets in $\mathcal{B}_{\text{reg}}^{\text{test}}$, averaged over ten train-validation-test splits. When we write $a \pm b$, $a$ is the mean error on the dataset and $[a - b, a + b]$ is an approximate 95% confidence interval for the mean in the #splits $\to \infty$ limit. The confidence interval is computed from the $t$-distribution using a normality assumption as in Appendix C.6. In each row, the lowest mean error is highlighted in bold, and errors whose confidence interval contains the lowest error are underlined.

| Dataset | RealMLP-TD | RealTabR-D | TabR-S-D | MLP-PLR-D | MLP-D | CatBoost-TD | LGBM-TD | XGB-TD | RF-D |
|---|---|---|---|---|---|---|---|---|---|
| airfoil_self_noise | **0.180**±0.012 | 0.223±0.009 | 0.263±0.025 | 0.249±0.020 | 0.308±0.019 | 0.213±0.009 | 0.233±0.017 | 0.238±0.016 | 0.300±0.016 |
| Airlines_DepDelay_10M | **0.979**±0.000 | 0.983±0.000 | 0.983±0.000 | 0.984±0.001 | 0.983±0.001 | 0.981±0.001 | 0.985±0.000 | 1.000±0.000 | 1.013±0.002 |
| Allstate_Claims_Severity | **0.654**±0.006 | 0.665±0.008 | 0.667±0.006 | 0.662±0.006 | 0.663±0.005 | 0.658±0.008 | 0.662±0.008 | 0.951±0.158 | 0.686±0.007 |
| auction_verification | 0.197±0.018 | 0.065±0.013 | 0.107±0.029 | 0.160±0.015 | 0.196±0.029 | **0.064**±0.019 | 0.206±0.036 | 0.064±0.014 | 0.122±0.019 |
| black_friday | 0.692±0.001 | 0.690±0.002 | 0.688±0.002 | 0.694±0.002 | 0.694±0.001 | **0.679**±0.002 | 0.679±0.002 | 0.681±0.002 | 0.743±0.002 |
| brazilian_houses | 1.076±0.569 | 0.784±0.287 | 0.706±0.127 | 0.576±0.154 | 0.703±0.148 | 0.581±0.133 | 0.891±0.281 | 0.818±0.114 | **0.539**±0.074 |
| Buzzinsocialmedia_Twitter | 0.341±0.014 | **0.233**±0.012 | 0.247±0.013 | 0.357±0.020 | 0.337±0.028 | 0.323±0.016 | 0.296±0.016 | 0.361±0.018 | 0.234±0.014 |
| california_housing | 0.420±0.007 | **0.362**±0.008 | 0.363±0.008 | 0.428±0.007 | 0.432±0.009 | 0.400±0.007 | 0.402±0.008 | 0.409±0.008 | 0.439±0.008 |
| concrete_compressive_strength | 0.298±0.025 | 0.301±0.024 | 0.306±0.029 | 0.303±0.019 | 0.323±0.025 | 0.303±0.017 | **0.297**±0.025 | 0.298±0.024 | 0.330±0.022 |
| cps88wages | 0.834±0.016 | **0.834**±0.015 | 0.835±0.015 | 0.836±0.015 | 0.836±0.016 | 0.842±0.013 | 0.850±0.015 | 0.853±0.014 | 0.928±0.029 |
| cpu_activity | 0.127±0.004 | **0.121**±0.004 | 0.123±0.004 | 0.130±0.004 | 0.143±0.004 | 0.166±0.010 | 0.125±0.009 | 0.127±0.011 | 0.134±0.005 |
| diamonds | 0.137±0.002 | **0.132**±0.002 | 0.136±0.002 | 0.145±0.002 | 0.144±0.002 | 0.138±0.002 | 0.143±0.003 | 0.138±0.003 | 0.148±0.005 |
| elevators | 0.281±0.005 | **0.280**±0.004 | 0.724±0.013 | 0.557±0.160 | 0.747±0.012 | 0.323±0.007 | 0.334±0.007 | 0.346±0.007 | 0.421±0.011 |
| fifa | **0.460**±0.023 | 0.507±0.031 | 0.506±0.026 | 0.461±0.030 | 0.504±0.021 | 0.485±0.021 | 0.472±0.023 | 0.548±0.032 | 0.482±0.026 |
| fps_benchmark | **0.006**±0.002 | 0.021±0.004 | 0.026±0.004 | 0.032±0.003 | 0.036±0.003 | 0.070±0.018 | 0.092±0.020 | 0.033±0.004 | 0.093±0.009 |
| geographical_origin_of_music | 0.887±0.030 | **0.874**±0.017 | 0.901±0.022 | 0.923±0.039 | 0.903±0.016 | 0.881±0.015 | 0.880±0.017 | 0.875±0.015 | 0.884±0.014 |
| health_insurance | **0.775**±0.005 | 0.777±0.005 | 0.776±0.004 | 0.776±0.005 | 0.777±0.004 | 0.776±0.004 | 0.781±0.003 | 0.784±0.004 | 0.827±0.004 |
| house_16H | 0.570±0.010 | 0.573±0.011 | **0.564**±0.011 | 0.570±0.015 | 0.570±0.016 | 0.580±0.011 | 0.573±0.011 | 0.584±0.012 | 0.608±0.009 |
| house_prices_nominal | 0.378±0.039 | **0.370**±0.041 | 0.382±0.035 | 0.419±0.041 | 0.406±0.051 | 0.422±0.021 | 0.376±0.035 | 0.378±0.038 | 0.372±0.038 |
| house_sales | **0.319**±0.007 | 0.336±0.014 | 0.341±0.009 | 0.323±0.013 | 0.334±0.010 | 0.321±0.013 | 0.324±0.013 | 0.331±0.011 | 0.360±0.009 |
| kin8nm | **0.242**±0.003 | 0.258±0.003 | 0.300±0.006 | 0.276±0.005 | 0.302±0.005 | 0.347±0.006 | 0.425±0.005 | 0.452±0.007 | 0.560±0.008 |
| kings_county | 0.326±0.007 | 0.349±0.012 | 0.347±0.011 | **0.325**±0.011 | 0.336±0.008 | 0.336±0.008 | 0.323±0.013 | 0.328±0.013 | 0.360±0.010 |
| Mercedes_Benz_Greener_Manufacturing | 0.672±0.030 | 0.672±0.030 | 0.679±0.033 | **0.668**±0.031 | 0.682±0.031 | 0.672±0.028 | 0.693±0.027 | 0.990±0.039 | 0.718±0.028 |
| miami_housing | 0.278±0.007 | **0.272**±0.008 | 0.276±0.008 | 0.286±0.006 | 0.296±0.010 | 0.279±0.011 | 0.274±0.007 | 0.278±0.008 | 0.307±0.008 |
| MIP-2016-regression | 0.835±0.038 | 0.815±0.032 | **0.801**±0.027 | 0.852±0.037 | 0.825±0.023 | 0.809±0.032 | 0.814±0.043 | 0.809±0.036 | 0.837±0.025 |
| nyc-taxi-green-dec-2016 | 0.695±0.025 | 0.658±0.016 | 0.660±0.017 | 0.722±0.019 | 0.706±0.015 | 0.651±0.015 | 0.662±0.013 | 1.137±0.166 | **0.643**±0.010 |
| pol | 0.067±0.003 | **0.067**±0.003 | 0.144±0.005 | 0.074±0.004 | 0.133±0.008 | 0.102±0.004 | 0.103±0.005 | 0.109±0.007 | 0.120±0.007 |
| pumadyn32nh | **0.590**±0.007 | 0.590±0.007 | 0.644±0.011 | 0.593±0.006 | 0.652±0.007 | 0.593±0.008 | 0.605±0.007 | 0.608±0.007 | 0.602±0.008 |
| QSAR-TID-10980 | 0.593±0.012 | 0.600±0.013 | 0.596±0.011 | 0.672±0.014 | 0.617±0.009 | 0.593±0.010 | **0.589**±0.008 | 0.596±0.009 | 0.612±0.009 |
| QSAR-TID-11 | 0.522±0.016 | **0.511**±0.014 | 0.519±0.014 | 0.578±0.014 | 0.531±0.012 | 0.527±0.013 | 0.521±0.014 | 0.526±0.013 | 0.538±0.015 |
| quake | 1.006±0.005 | 1.007±0.008 | 1.004±0.011 | 1.004±0.008 | 1.002±0.010 | **1.000**±0.004 | 1.004±0.005 | 1.001±0.004 | 1.050±0.020 |
| Santander_transaction_value | 0.901±0.015 | 0.932±0.011 | 0.994±0.003 | 0.907±0.014 | 0.996±0.005 | 0.866±0.017 | **0.864**±0.021 | 0.871±0.018 | 0.883±0.022 |
| sarcos | 0.117±0.002 | **0.101**±0.002 | 0.107±0.002 | 0.141±0.006 | 0.132±0.003 | 0.125±0.002 | 0.129±0.002 | 0.134±0.002 | 0.172±0.003 |
| SAT11-HAND-runtime-regression | 0.475±0.040 | 0.481±0.039 | **0.465**±0.032 | 0.509±0.054 | 0.485±0.031 | 0.492±0.034 | 0.558±0.028 | 0.493±0.034 | 0.619±0.035 |
| socmob | 0.426±0.038 | 0.396±0.034 | 0.379±0.057 | 0.396±0.086 | 0.481±0.091 | **0.378**±0.028 | 0.436±0.056 | 0.402±0.062 | 0.490±0.055 |
| solar_flare | 0.982±0.055 | 0.976±0.055 | **0.963**±0.050 | 0.973±0.070 | 0.979±0.057 | 0.995±0.016 | 1.002±0.035 | 0.984±0.013 | 1.116±0.078 |
| space_ga | 0.555±0.024 | **0.496**±0.031 | 0.504±0.028 | 0.520±0.027 | 0.503±0.025 | 0.570±0.033 | 0.565±0.031 | 0.571±0.029 | 0.610±0.030 |
| topo_2_1 | **0.968**±0.004 | 0.970±0.004 | 0.970±0.009 | 0.973±0.007 | 0.969±0.006 | 0.975±0.006 | 0.977±0.005 | 0.978±0.006 | 0.983±0.009 |
| video_transcoding | **0.056**±0.004 | 0.073±0.006 | 0.080±0.007 | 0.110±0.004 | 0.095±0.005 | 0.057±0.004 | 0.067±0.005 | 0.067±0.006 | 0.115±0.006 |
| wave_energy | **0.003**±0.001 | 0.006±0.001 | 0.023±0.002 | 0.062±0.007 | 0.110±0.008 | 0.078±0.001 | 0.139±0.001 | 0.193±0.001 | 0.414±0.002 |
| Yolanda | 0.793±0.001 | 0.754±0.002 | **0.754**±0.001 | 0.804±0.002 | 0.796±0.001 | 0.804±0.001 | 0.806±0.001 | 0.806±0.001 | 0.845±0.001 |
| yprop_4_1 | 0.963±0.010 | 0.964±0.009 | **0.950**±0.007 | 0.967±0.009 | 0.965±0.007 | 0.960±0.007 | 0.959±0.008 | 0.967±0.015 | 0.963±0.019 |

Table D.8: nRMSE of *tuned* methods on datasets in $\mathcal{B}^{\text{test}}_{\text{reg}}$, averaged over ten train-validation-test splits. When we write $a \pm b$, $a$ is the mean error on the dataset and $[a-b, a+b]$ is an approximate 95% confidence interval for the mean in the #splits $\to \infty$ limit. The confidence interval is computed from the $t$-distribution using a normality assumption as in Appendix C.6. In each row, the lowest mean error is highlighted in bold, and errors whose confidence interval contains the lowest error are underlined.

| Dataset | RealMLP-HPO | MLP-PLR-HPO | ResNet-HPO | MLP-HPO | CatBoost-HPO | LGBM-HPO | XGB-HPO |
|---|---|---|---|---|---|---|---|
| airfoil_self_noise | **0.174**±0.011 | 0.210±0.011 | 0.329±0.015 | 0.233±0.017 | 0.227±0.011 | 0.241±0.015 | 0.248±0.012 |
| Airlines_DepDelay_10M | **0.979**±0.000 | 0.980±0.001 | 0.982±0.000 | 0.983±0.000 | 0.980±0.001 | 0.980±0.001 | 0.982±0.001 |
| Allstate_Claims_Severity | **0.651**±0.006 | 0.655±0.006 | 0.658±0.006 | 0.659±0.005 | 0.652±0.005 | 0.656±0.008 | 0.656±0.006 |
| auction_verification | 0.101±0.016 | 0.067±0.025 | 0.178±0.013 | 0.162±0.014 | **0.061**±0.020 | 0.130±0.030 | 0.088±0.010 |
| black_friday | 0.686±0.003 | 0.687±0.001 | 0.690±0.002 | 0.693±0.002 | 0.679±0.002 | **0.679**±0.001 | 0.681±0.001 |
| brazilian_houses | 0.788±0.345 | 0.742±0.109 | 1.623±1.223 | 0.606±0.147 | 0.711±0.369 | 0.878±0.340 | **0.565**±0.079 |
| Buzzinsocialmedia_Twitter | 0.266±0.015 | 0.254±0.018 | 0.286±0.025 | 0.275±0.017 | 0.320±0.018 | 0.289±0.018 | **0.222**±0.014 |
| california_housing | 0.413±0.008 | 0.427±0.008 | 0.426±0.010 | 0.435±0.008 | 0.402±0.007 | **0.398**±0.008 | 0.400±0.008 |
| concrete_compressive_strength | 0.290±0.029 | 0.295±0.024 | 0.314±0.028 | 0.314±0.029 | **0.271**±0.030 | 0.279±0.021 | 0.278±0.023 |
| cps88wages | 0.834±0.015 | **0.833**±0.016 | 0.835±0.015 | 0.834±0.015 | 0.835±0.015 | 0.835±0.016 | 0.834±0.016 |
| cpu_activity | 0.125±0.005 | 0.124±0.004 | 0.127±0.003 | 0.137±0.006 | 0.122±0.003 | 0.122±0.007 | **0.119**±0.006 |
| diamonds | **0.134**±0.002 | 0.138±0.003 | 0.141±0.002 | 0.141±0.002 | 0.136±0.002 | 0.138±0.003 | 0.135±0.002 |
| elevators | **0.276**±0.005 | 0.306±0.013 | 0.315±0.005 | 0.731±0.039 | 0.307±0.007 | 0.318±0.006 | 0.322±0.006 |
| fifa | **0.457**±0.025 | 0.466±0.027 | 0.494±0.028 | 0.512±0.026 | 0.465±0.026 | 0.466±0.027 | 0.486±0.021 |
| fps_benchmark | **0.004**±0.002 | 0.006±0.001 | 0.039±0.002 | 0.008±0.001 | 0.038±0.019 | 0.018±0.002 | 0.033±0.007 |
| geographical_origin_of_music | 0.899±0.038 | 0.934±0.033 | 0.919±0.039 | 0.906±0.030 | 0.871±0.017 | 0.869±0.024 | **0.861**±0.020 |
| health_insurance | 0.775±0.004 | 0.777±0.006 | 0.775±0.005 | 0.775±0.005 | 0.775±0.005 | 0.775±0.004 | **0.774**±0.005 |
| house_16H | 0.564±0.014 | 0.558±0.011 | **0.551**±0.012 | 0.570±0.018 | 0.573±0.011 | 0.571±0.013 | 0.575±0.014 |
| house_prices_nominal | 0.399±0.051 | 0.378±0.031 | 0.445±0.063 | 0.384±0.046 | **0.361**±0.022 | 0.383±0.030 | 0.374±0.039 |
| house_sales | 0.320±0.013 | 0.320±0.010 | 0.340±0.010 | 0.340±0.011 | **0.310**±0.011 | 0.319±0.008 | 0.324±0.007 |
| kin8nm | **0.238**±0.003 | 0.264±0.004 | 0.279±0.004 | 0.298±0.006 | 0.378±0.009 | 0.424±0.014 | 0.461±0.006 |
| kings_county | 0.327±0.010 | 0.317±0.008 | 0.341±0.013 | 0.339±0.009 | **0.309**±0.008 | 0.323±0.008 | 0.325±0.007 |
| Mercedes_Benz_Greener_Manufacturing | 0.668±0.032 | 0.670±0.030 | 0.680±0.031 | 0.677±0.030 | **0.664**±0.031 | 0.669±0.029 | 0.664±0.031 |
| miami_housing | 0.267±0.009 | 0.272±0.007 | 0.287±0.010 | 0.295±0.013 | **0.257**±0.006 | 0.269±0.008 | 0.272±0.011 |
| MIP-2016-regression | 0.843±0.037 | 0.829±0.032 | 0.835±0.030 | 0.846±0.043 | 0.788±0.034 | **0.788**±0.038 | 0.807±0.030 |
| nyc-taxi-green-dec-2016 | **0.613**±0.031 | 0.665±0.020 | 0.643±0.019 | 0.692±0.065 | 0.656±0.014 | 0.655±0.013 | 0.661±0.016 |
| pol | **0.059**±0.004 | 0.065±0.005 | 0.169±0.006 | 0.130±0.006 | 0.119±0.004 | 0.106±0.005 | 0.108±0.006 |
| pumadyn32nh | **0.586**±0.007 | 0.589±0.007 | 0.606±0.009 | 0.626±0.010 | 0.594±0.006 | 0.599±0.006 | 0.602±0.007 |
| QSAR-TID-10980 | 0.598±0.014 | 0.636±0.010 | 0.614±0.008 | 0.612±0.011 | 0.596±0.011 | **0.582**±0.010 | 0.590±0.010 |
| QSAR-TID-11 | 0.514±0.016 | 0.552±0.015 | 0.524±0.017 | 0.527±0.011 | 0.526±0.014 | **0.509**±0.015 | 0.517±0.015 |
| quake | 1.012±0.012 | 1.004±0.010 | 1.006±0.007 | 1.003±0.006 | 1.005±0.008 | **1.001**±0.007 | 1.005±0.010 |
| Santander_transaction_value | 0.879±0.024 | 0.856±0.026 | 0.934±0.016 | 0.921±0.012 | 0.873±0.017 | **0.843**±0.020 | 0.851±0.019 |
| sarcos | **0.102**±0.002 | 0.110±0.002 | 0.109±0.002 | 0.113±0.003 | 0.136±0.002 | 0.128±0.002 | 0.132±0.003 |
| SAT11-HAND-runtime-regression | **0.444**±0.053 | 0.489±0.035 | 0.477±0.031 | 0.464±0.038 | 0.515±0.032 | 0.501±0.029 | 0.531±0.041 |
| socmob | 0.383±0.054 | **0.299**±0.041 | 0.459±0.055 | 0.412±0.080 | 0.364±0.054 | 0.404±0.052 | 0.417±0.054 |
| solar_flare | 1.017±0.089 | 0.981±0.067 | 0.975±0.069 | 0.975±0.066 | 0.984±0.046 | **0.972**±0.054 | 1.009±0.156 |
| space_ga | 0.495±0.022 | 0.516±0.015 | **0.489**±0.021 | 0.499±0.019 | 0.548±0.026 | 0.546±0.026 | 0.564±0.025 |
| topo_2_1 | 0.968±0.005 | 0.972±0.005 | 0.970±0.005 | 0.968±0.004 | 0.970±0.004 | **0.964**±0.004 | 0.968±0.007 |
| video_transcoding | **0.052**±0.005 | 0.057±0.005 | 0.068±0.006 | 0.063±0.006 | 0.073±0.002 | 0.067±0.004 | 0.072±0.002 |
| wave_energy | **0.003**±0.001 | 0.007±0.001 | 0.044±0.002 | 0.029±0.004 | 0.049±0.001 | 0.081±0.004 | 0.095±0.009 |
| Yolanda | 0.786±0.001 | 0.791±0.002 | **0.786**±0.001 | 0.791±0.002 | 0.810±0.001 | 0.795±0.002 | 0.800±0.003 |
| yprop_4_1 | 0.965±0.007 | 0.965±0.004 | 0.963±0.009 | 0.965±0.009 | 0.963±0.008 | **0.949**±0.005 | 0.954±0.008 |

Table D.9: Classification error of *untuned* methods on datasets in $\mathcal{B}^{\text{Grinsztajn}}_{\text{class}}$, averaged over ten train-validation-test splits. When we write $a \pm b$, $a$ is the mean error on the dataset and $[a-b, a+b]$ is an approximate 95% confidence interval for the mean in the #splits $\to \infty$ limit. The confidence interval is computed from the $t$-distribution using a normality assumption as in Appendix C.6. In each row, the lowest mean error is highlighted in bold, and errors whose confidence interval contains the lowest error are underlined.

| Dataset | RealMLP-TD | RealTabR-D | TabR-S-D | MLP-PLR-D | MLP-D | CatBoost-TD | LGBM-TD | XGB-TD | RF-D |
|---|---|---|---|---|---|---|---|---|---|
| albert | 0.348±0.003 | 0.350±0.001 | 0.349±0.001 | **0.346**±0.001 | 0.348±0.002 | 0.347±0.002 | 0.347±0.002 | 0.363±0.005 | 0.353±0.001 |
| bank-marketing | 0.206±0.008 | 0.198±0.009 | 0.196±0.004 | 0.201±0.006 | 0.207±0.006 | **0.193**±0.009 | 0.196±0.008 | 0.195±0.006 | 0.200±0.006 |
| Bioresponse | 0.240±0.011 | 0.233±0.007 | 0.236±0.011 | 0.249±0.013 | 0.240±0.008 | 0.228±0.013 | **0.227**±0.010 | 0.229±0.006 | 0.232±0.008 |
| california | 0.114±0.003 | **0.090**±0.004 | 0.092±0.003 | 0.113±0.003 | 0.122±0.003 | 0.095±0.002 | 0.097±0.002 | 0.097±0.003 | 0.111±0.002 |
| compas-two-years | **0.325**±0.009 | 0.332±0.008 | 0.326±0.007 | 0.328±0.007 | 0.326±0.005 | 0.325±0.004 | 0.329±0.006 | 0.331±0.009 | 0.377±0.009 |
| covertype | 0.122±0.002 | **0.096**±0.001 | 0.101±0.001 | 0.145±0.004 | 0.144±0.002 | 0.138±0.001 | 0.143±0.002 | 0.146±0.003 | 0.153±0.001 |
| credit | 0.227±0.005 | 0.224±0.005 | 0.226±0.007 | 0.224±0.007 | 0.225±0.006 | **0.223**±0.007 | 0.226±0.007 | 0.227±0.008 | 0.235±0.005 |
| default-of-credit-card-clients | **0.281**±0.005 | 0.284±0.005 | 0.286±0.005 | 0.284±0.004 | 0.285±0.006 | 0.286±0.005 | 0.284±0.005 | 0.287±0.005 | 0.292±0.003 |
| Diabetes130US | 0.397±0.002 | 0.396±0.001 | 0.397±0.001 | 0.396±0.001 | 0.396±0.001 | **0.395**±0.001 | 0.399±0.002 | 0.398±0.001 | 0.438±0.001 |
| electricity | 0.170±0.007 | 0.154±0.015 | **0.110**±0.007 | 0.161±0.002 | 0.170±0.004 | 0.117±0.002 | 0.115±0.001 | 0.111±0.001 | 0.137±0.002 |
| heloc | 0.284±0.007 | 0.284±0.009 | 0.284±0.007 | 0.276±0.008 | 0.282±0.009 | 0.281±0.008 | 0.282±0.007 | 0.284±0.006 | 0.283±0.006 |
| Higgs | **0.288**±0.002 | 0.292±0.002 | 0.307±0.001 | 0.288±0.001 | 0.310±0.002 | 0.290±0.001 | 0.289±0.001 | 0.294±0.002 | 0.300±0.001 |
| house_16H | 0.116±0.004 | 0.119±0.002 | 0.115±0.003 | **0.113**±0.003 | 0.116±0.005 | 0.116±0.004 | 0.115±0.004 | 0.118±0.004 | 0.121±0.003 |
| jannis | 0.222±0.002 | 0.226±0.001 | 0.257±0.002 | **0.221**±0.001 | 0.249±0.002 | 0.222±0.001 | 0.224±0.001 | 0.230±0.001 | 0.235±0.001 |
| MagicTelescope | 0.137±0.004 | 0.131±0.004 | **0.130**±0.003 | 0.136±0.004 | 0.139±0.004 | 0.136±0.004 | 0.139±0.005 | 0.138±0.005 | 0.144±0.004 |
| MiniBooNE | **0.063**±0.001 | 0.065±0.001 | 0.067±0.001 | 0.066±0.001 | 0.064±0.001 | 0.064±0.001 | 0.063±0.001 | 0.064±0.001 | 0.078±0.001 |
| pol | **0.012**±0.002 | 0.014±0.001 | 0.032±0.002 | 0.017±0.002 | 0.037±0.004 | 0.014±0.001 | 0.015±0.001 | 0.014±0.001 | 0.016±0.002 |
| road-safety | 0.233±0.002 | 0.232±0.002 | **0.230**±0.001 | 0.238±0.004 | 0.241±0.002 | 0.233±0.001 | 0.237±0.001 | 0.242±0.001 | 0.243±0.001 |

Table D.10: Classification error of *tuned* methods on datasets in $\mathcal{B}_{\text{class}}^{\text{Grinsztajn}}$, averaged over ten train-validation-test splits. When we write $a \pm b$, $a$ is the mean error on the dataset and $[a-b, a+b]$ is an approximate 95% confidence interval for the mean in the #splits $\to \infty$ limit. The confidence interval is computed from the $t$-distribution using a normality assumption as in Appendix C.6. In each row, the lowest mean error is highlighted in bold, and errors whose confidence interval contains the lowest error are underlined.

| Dataset | RealMLP-HPO | TabR-HPO | MLP-PLR-HPO | FTT-HPO | ResNet-HPO | MLP-HPO | CatBoost-HPO | LGBM-HPO | XGB-HPO | RF-HPO |
|---|---|---|---|---|---|---|---|---|---|---|
| albert | 0.349±0.002 | 0.348±0.002 | 0.346±0.002 | 0.346±0.002 | 0.348±0.002 | 0.347±0.001 | **0.344**±0.001 | 0.347±0.003 | 0.348±0.002 | 0.346±0.002 |
| bank-marketing | 0.202±0.004 | **0.193**±0.006 | 0.199±0.005 | 0.199±0.003 | 0.204±0.007 | 0.207±0.007 | 0.194±0.008 | 0.194±0.005 | 0.193±0.006 | 0.199±0.009 |
| Bioresponse | 0.235±0.010 | 0.244±0.010 | 0.250±0.008 | 0.245±0.011 | 0.232±0.010 | 0.236±0.010 | 0.234±0.009 | 0.229±0.007 | 0.229±0.013 | 0.223±0.008 |
| california | 0.111±0.003 | **0.090**±0.002 | 0.114±0.003 | 0.109±0.003 | 0.116±0.003 | 0.123±0.002 | 0.095±0.002 | 0.095±0.004 | 0.097±0.002 | 0.108±0.003 |
| compas-two-years | **0.325**±0.007 | 0.330±0.010 | 0.329±0.010 | 0.332±0.011 | 0.330±0.008 | 0.325±0.006 | 0.327±0.008 | 0.327±0.008 | 0.329±0.007 | 0.329±0.004 |
| covertype | 0.120±0.002 | **0.096**±0.001 | 0.140±0.002 | 0.125±0.002 | 0.142±0.003 | 0.145±0.002 | 0.142±0.001 | 0.135±0.002 | 0.147±0.006 | 0.144±0.003 |
| credit | 0.225±0.006 | 0.225±0.007 | 0.225±0.006 | 0.224±0.007 | 0.225±0.006 | 0.223±0.005 | **0.222**±0.007 | 0.226±0.007 | 0.224±0.007 | 0.226±0.006 |
| default-of-credit-card-clients | 0.285±0.005 | 0.284±0.007 | 0.286±0.004 | 0.285±0.005 | 0.285±0.007 | 0.286±0.007 | 0.281±0.005 | 0.285±0.005 | 0.281±0.004 | **0.281**±0.005 |
| Diabetes130US | 0.398±0.002 | 0.397±0.001 | 0.396±0.001 | 0.397±0.003 | 0.397±0.002 | 0.398±0.003 | 0.396±0.002 | 0.395±0.001 | 0.395±0.001 | **0.395**±0.001 |
| electricity | 0.162±0.003 | **0.063**±0.003 | 0.160±0.004 | 0.160±0.002 | 0.170±0.004 | 0.167±0.002 | 0.113±0.002 | 0.112±0.002 | 0.119±0.003 | 0.126±0.002 |
| heloc | 0.284±0.010 | 0.279±0.010 | **0.277**±0.008 | 0.279±0.008 | 0.280±0.009 | 0.283±0.008 | 0.277±0.006 | 0.280±0.007 | 0.283±0.008 | 0.282±0.008 |
| Higgs | **0.286**±0.001 | 0.288±0.001 | 0.290±0.003 | 0.291±0.002 | 0.299±0.003 | 0.303±0.002 | 0.290±0.002 | 0.289±0.002 | 0.289±0.002 | 0.295±0.002 |
| house_16H | 0.119±0.004 | 0.115±0.004 | **0.114**±0.003 | 0.115±0.004 | 0.115±0.004 | 0.116±0.003 | 0.114±0.004 | 0.115±0.005 | 0.116±0.003 | 0.121±0.004 |
| jannis | 0.221±0.002 | **0.220**±0.002 | 0.222±0.002 | 0.225±0.004 | 0.234±0.002 | 0.243±0.003 | 0.222±0.001 | 0.222±0.002 | 0.224±0.002 | 0.227±0.002 |
| MagicTelescope | 0.134±0.005 | **0.130**±0.005 | 0.137±0.008 | 0.137±0.004 | 0.137±0.005 | 0.140±0.004 | 0.138±0.007 | 0.139±0.005 | 0.140±0.004 | 0.142±0.005 |
| MiniBooNE | **0.061**±0.001 | 0.063±0.001 | 0.063±0.001 | 0.065±0.001 | 0.061±0.001 | 0.062±0.001 | 0.064±0.001 | 0.064±0.001 | 0.064±0.001 | 0.074±0.001 |
| pol | **0.013**±0.002 | 0.015±0.002 | 0.014±0.002 | 0.015±0.002 | 0.031±0.003 | 0.032±0.004 | 0.014±0.002 | 0.015±0.001 | 0.016±0.002 | 0.018±0.002 |
| road-safety | 0.229±0.002 | **0.223**±0.002 | 0.234±0.002 | 0.229±0.001 | 0.229±0.002 | 0.234±0.002 | 0.235±0.001 | 0.234±0.001 | 0.237±0.001 | 0.241±0.001 |

Table D.11: nRMSE of *untuned* methods on datasets in $\mathcal{B}_{\text{reg}}^{\text{Grinsztajn}}$, averaged over ten train-validation-test splits. When we write $a \pm b$, $a$ is the mean error on the dataset and $[a-b, a+b]$ is an approximate 95% confidence interval for the mean in the #splits $\to \infty$ limit. The confidence interval is computed from the $t$-distribution using a normality assumption as in Appendix C.6. In each row, the lowest mean error is highlighted in bold, and errors whose confidence interval contains the lowest error are underlined.

| Dataset | RealMLP-TD | RealTabR-D | TabR-S-D | MLP-PLR-D | MLP-D | CatBoost-TD | LGBM-TD | XGB-TD | RF-D |
|---|---|---|---|---|---|---|---|---|---|
| abalone | 0.668±0.014 | **0.647**±0.012 | 0.649±0.013 | 0.666±0.012 | 0.666±0.015 | 0.686±0.011 | 0.687±0.013 | 0.692±0.011 | 0.688±0.012 |
| Ailerons | 0.396±0.006 | 0.394±0.007 | 0.397±0.007 | 0.397±0.006 | 0.403±0.007 | **0.383**±0.007 | 0.389±0.006 | 0.408±0.006 | 0.402±0.006 |
| Airlines_DepDelay_1M | 0.979±0.001 | 0.979±0.001 | 0.981±0.001 | 0.980±0.001 | 0.980±0.001 | 0.979±0.000 | 0.982±0.000 | 0.984±0.001 | 1.011±0.001 |
| Allstate_Claims_Severity | 0.707±0.003 | 0.697±0.001 | 0.699±0.001 | **0.692**±0.001 | 0.698±0.001 | 0.695±0.001 | 0.694±0.001 | 0.839±0.026 | 0.728±0.002 |
| analcatdata_supreme | 0.142±0.015 | **0.136**±0.011 | 0.142±0.013 | 0.144±0.015 | 0.141±0.014 | 0.141±0.013 | 0.144±0.013 | 0.145±0.013 | 0.145±0.013 |
| Bike_Sharing_Demand | **0.228**±0.005 | 0.232±0.006 | 0.237±0.006 | 0.242±0.003 | 0.244±0.005 | 0.228±0.005 | 0.231±0.004 | 0.243±0.016 | 0.258±0.005 |
| Brazilian_houses | 0.068±0.026 | 0.064±0.028 | 0.074±0.022 | 0.065±0.015 | 0.068±0.011 | 0.067±0.020 | **0.059**±0.021 | 0.067±0.021 | 0.074±0.025 |
| cpu_act | 0.129±0.005 | **0.121**±0.004 | 0.123±0.004 | 0.127±0.004 | 0.144±0.004 | 0.168±0.011 | 0.125±0.009 | 0.127±0.011 | 0.134±0.005 |
| delays_zurich_transport | **0.966**±0.002 | 0.966±0.001 | 0.967±0.001 | 0.967±0.001 | 0.969±0.001 | 0.967±0.001 | 0.968±0.001 | 0.971±0.001 | 1.068±0.003 |
| diamonds | 0.096±0.002 | **0.088**±0.001 | 0.092±0.001 | 0.102±0.002 | 0.102±0.003 | 0.092±0.001 | 0.097±0.001 | 0.094±0.001 | 0.115±0.004 |
| elevators | **0.280**±0.005 | 0.280±0.015 | 0.728±0.413 | 0.667±0.138 | 0.745±0.014 | 0.323±0.006 | 0.334±0.007 | 0.345±0.006 | 0.420±0.011 |
| house_16H | 0.698±0.024 | 0.701±0.019 | 0.685±0.021 | 0.672±0.013 | 0.680±0.011 | 0.682±0.014 | 0.685±0.018 | 0.693±0.015 | **0.667**±0.019 |
| house_sales | 0.324±0.003 | **0.312**±0.003 | 0.320±0.003 | 0.328±0.003 | 0.341±0.003 | 0.324±0.003 | 0.327±0.003 | 0.334±0.002 | 0.354±0.003 |
| houses | 0.411±0.006 | **0.362**±0.005 | 0.362±0.004 | 0.415±0.004 | 0.419±0.004 | 0.391±0.003 | 0.394±0.004 | 0.399±0.003 | 0.419±0.005 |
| medical_charges | **0.144**±0.000 | 0.144±0.000 | 0.145±0.001 | 0.145±0.001 | 0.148±0.002 | 0.146±0.000 | 0.150±0.000 | 0.154±0.000 | 0.153±0.001 |
| Mercedes_Benz_Greener_Manufacturing | 0.675±0.030 | **0.672**±0.030 | 0.675±0.029 | 0.673±0.032 | 0.674±0.031 | 0.677±0.029 | 0.694±0.027 | 0.691±0.026 | 0.736±0.023 |
| MiamiHousing2016 | 0.262±0.003 | **0.246**±0.005 | 0.252±0.005 | 0.269±0.005 | 0.280±0.005 | 0.260±0.004 | 0.267±0.004 | 0.271±0.004 | 0.295±0.006 |
| nyc-taxi-green-dec-2016 | 0.704±0.009 | **0.664**±0.002 | 0.677±0.003 | 0.750±0.028 | 0.707±0.003 | 0.677±0.002 | 0.669±0.003 | 0.695±0.003 | 0.668±0.002 |
| particulate-matter-ukair-2017 | 0.581±0.002 | **0.566**±0.004 | 0.568±0.004 | 0.582±0.002 | 0.587±0.001 | 0.566±0.001 | 0.572±0.001 | 0.579±0.001 | 0.597±0.001 |
| pol | **0.067**±0.003 | 0.067±0.003 | 0.142±0.006 | 0.074±0.004 | 0.141±0.009 | 0.102±0.004 | 0.103±0.005 | 0.109±0.007 | 0.120±0.007 |
| seattlecrime6 | 0.906±0.002 | 0.904±0.001 | 0.905±0.001 | 0.905±0.001 | 0.906±0.001 | **0.903**±0.001 | 0.905±0.001 | 0.910±0.001 | 0.914±0.001 |
| SGEMM_GPU_kernel_performance | **0.014**±0.000 | 0.014±0.000 | 0.022±0.003 | 0.032±0.002 | 0.032±0.003 | 0.017±0.000 | 0.016±0.000 | 0.017±0.000 | 0.015±0.000 |
| sulfur | 0.427±0.063 | **0.376**±0.038 | 0.402±0.051 | 0.429±0.057 | 0.438±0.051 | 0.414±0.056 | 0.424±0.058 | 0.415±0.065 | 0.439±0.049 |
| superconduct | 0.305±0.005 | 0.308±0.004 | 0.304±0.004 | 0.319±0.004 | 0.308±0.005 | 0.291±0.005 | 0.290±0.004 | **0.289**±0.004 | 0.295±0.003 |
| topo_2_1 | **0.969**±0.005 | 0.970±0.005 | 0.971±0.004 | 0.969±0.004 | 0.970±0.003 | 0.975±0.007 | 0.978±0.009 | 0.979±0.007 | 0.984±0.011 |
| visualizing_soil | 0.009±0.001 | 0.007±0.001 | 0.020±0.009 | 0.020±0.001 | 0.027±0.002 | 0.004±0.001 | 0.005±0.001 | 0.005±0.001 | **0.004**±0.001 |
| wine_quality | 0.762±0.014 | 0.736±0.010 | 0.737±0.011 | 0.783±0.011 | 0.778±0.015 | 0.716±0.010 | **0.711**±0.013 | 0.711±0.012 | 0.717±0.012 |
| yprop_4_1 | 0.968±0.008 | 0.958±0.005 | **0.957**±0.005 | 0.966±0.003 | 0.969±0.009 | 0.962±0.005 | 0.965±0.008 | 0.969±0.004 | 0.968±0.007 |

Table D.12: nRMSE of *tuned* methods on datasets in $\mathcal{B}_{\text{reg}}^{\text{Grinsztajn}}$, averaged over ten train-validation-test splits. When we write $a \pm b$, $a$ is the mean error on the dataset and $[a-b, a+b]$ is an approximate 95% confidence interval for the mean in the #splits $\to \infty$ limit. The confidence interval is computed from the $t$-distribution using a normality assumption as in Appendix C.6. In each row, the lowest mean error is highlighted in bold, and errors whose confidence interval contains the lowest error are underlined.

| Dataset | RealMLP-HPO | TabR-HPO | MLP-PLR-HPO | FTT-HPO | ResNet-HPO | MLP-HPO | CatBoost-HPO | LGBM-HPO | XGB-HPO | RF-HPO |
|---|---|---|---|---|---|---|---|---|---|---|
| abalone | 0.661±0.013 | **0.653**±0.009 | 0.664±0.011 | 0.665±0.011 | 0.656±0.015 | 0.666±0.010 | 0.679±0.011 | 0.675±0.011 | 0.672±0.010 | 0.675±0.012 |
| Ailerons | 0.385±0.009 | 0.390±0.008 | 0.397±0.007 | 0.388±0.006 | 0.400±0.007 | 0.402±0.007 | **0.380**±0.006 | 0.385±0.007 | 0.411±0.006 | 0.402±0.006 |
| Airlines_DepDelay_1M | 0.978±0.000 | 0.978±0.001 | 0.978±0.001 | 0.978±0.001 | 0.980±0.001 | 0.982±0.001 | 0.978±0.001 | 0.977±0.001 | **0.977**±0.000 | 0.979±0.000 |
| Allstate_Claims_Severity | 0.691±0.001 | 0.692±0.001 | 0.689±0.001 | 0.689±0.002 | 0.698±0.001 | 0.695±0.001 | 0.685±0.001 | **0.685**±0.001 | 0.753±0.012 | 0.713±0.002 |
| analcatdata_supreme | 0.144±0.018 | 0.148±0.015 | 0.142±0.018 | 0.142±0.017 | 0.149±0.016 | 0.144±0.016 | 0.142±0.014 | 0.143±0.016 | **0.141**±0.018 | 0.146±0.013 |
| Bike_Sharing_Demand | 0.229±0.005 | **0.227**±0.004 | 0.237±0.005 | 0.238±0.007 | 0.276±0.005 | 0.245±0.007 | 0.234±0.004 | 0.234±0.005 | 0.241±0.014 | 0.257±0.005 |
| Brazilian_houses | **0.053**±0.015 | 0.072±0.022 | 0.057±0.014 | 0.058±0.014 | 0.063±0.010 | 0.062±0.014 | 0.067±0.016 | 0.056±0.023 | 0.063±0.017 | 0.089±0.031 |
| cpu_act | 0.125±0.004 | **0.115**±0.004 | 0.125±0.004 | 0.120±0.005 | 0.128±0.004 | 0.137±0.004 | 0.123±0.003 | 0.122±0.007 | 0.120±0.005 | 0.132±0.005 |
| delays_zurich_transport | 0.965±0.001 | 0.966±0.001 | 0.966±0.001 | 0.966±0.001 | 0.968±0.001 | 0.969±0.001 | 0.964±0.001 | 0.963±0.001 | **0.963**±0.001 | 0.963±0.000 |
| diamonds | 0.091±0.001 | **0.090**±0.001 | 0.095±0.002 | 0.095±0.001 | 0.107±0.003 | 0.105±0.002 | 0.092±0.001 | 0.093±0.002 | 0.093±0.001 | 0.107±0.001 |
| elevators | **0.276**±0.005 | 0.285±0.005 | 0.315±0.013 | 0.485±0.137 | 0.318±0.007 | 0.744±0.017 | 0.308±0.007 | 0.319±0.005 | 0.328±0.008 | 0.428±0.013 |
| house_16H | 0.714±0.032 | 0.694±0.013 | 0.680±0.025 | 0.694±0.036 | 0.680±0.014 | 0.684±0.018 | 0.674±0.015 | 0.680±0.021 | 0.673±0.016 | **0.665**±0.015 |
| house_sales | 0.320±0.003 | **0.313**±0.003 | 0.323±0.003 | 0.321±0.003 | 0.331±0.003 | 0.338±0.002 | 0.320±0.003 | 0.323±0.003 | 0.323±0.003 | 0.353±0.003 |
| houses | 0.402±0.004 | **0.357**±0.005 | 0.418±0.005 | 0.405±0.005 | 0.420±0.005 | 0.421±0.006 | 0.392±0.004 | 0.391±0.004 | 0.395±0.006 | 0.418±0.004 |
| medical_charges | **0.143**±0.000 | 0.144±0.000 | 0.144±0.000 | 0.144±0.000 | 0.147±0.002 | 0.145±0.001 | 0.145±0.000 | 0.145±0.000 | 0.147±0.000 | 0.147±0.001 |
| Mercedes_Benz_Greener_Manufacturing | 0.671±0.032 | 0.672±0.030 | 0.670±0.030 | 0.668±0.031 | 0.677±0.032 | 0.669±0.030 | 0.669±0.029 | 0.666±0.030 | **0.665**±0.031 | 0.668±0.030 |
| MiamiHousing2016 | 0.260±0.004 | **0.245**±0.004 | 0.262±0.005 | 0.261±0.004 | 0.268±0.006 | 0.280±0.006 | 0.254±0.003 | 0.258±0.004 | 0.258±0.005 | 0.280±0.005 |
| nyc-taxi-green-dec-2016 | 0.670±0.002 | 0.656±0.012 | 0.688±0.004 | 0.715±0.022 | 0.691±0.004 | 0.701±0.004 | 0.668±0.004 | 0.665±0.003 | 0.697±0.004 | **0.655**±0.002 |
| particulate-matter-ukair-2017 | 0.578±0.002 | 0.564±0.004 | 0.575±0.001 | 0.579±0.003 | 0.583±0.002 | 0.586±0.001 | **0.563**±0.001 | 0.563±0.001 | 0.563±0.002 | 0.577±0.001 |
| pol | **0.059**±0.004 | 0.066±0.004 | 0.064±0.003 | 0.066±0.003 | 0.166±0.007 | 0.127±0.008 | 0.118±0.004 | 0.107±0.005 | 0.109±0.004 | 0.117±0.005 |
| seattlecrime6 | 0.904±0.001 | **0.903**±0.001 | 0.904±0.001 | 0.904±0.001 | 0.909±0.001 | 0.905±0.001 | 0.903±0.001 | 0.903±0.001 | 0.903±0.001 | 0.904±0.001 |
| SGEMM_GPU_kernel_performance | **0.014**±0.000 | 0.015±0.001 | 0.016±0.001 | 0.017±0.001 | 0.039±0.002 | 0.017±0.000 | 0.017±0.000 | 0.016±0.000 | 0.016±0.000 | 0.014±0.000 |
| sulfur | **0.361**±0.057 | 0.372±0.040 | 0.397±0.068 | 0.410±0.056 | 0.428±0.057 | 0.404±0.055 | 0.398±0.036 | 0.423±0.060 | 0.416±0.058 | 0.416±0.044 |
| superconduct | 0.299±0.006 | 0.299±0.004 | 0.304±0.005 | 0.315±0.005 | 0.304±0.006 | 0.305±0.004 | 0.294±0.003 | 0.286±0.004 | **0.285**±0.004 | 0.291±0.006 |
| topo_2_1 | 0.968±0.006 | 0.968±0.004 | 0.970±0.004 | 0.968±0.005 | 0.971±0.004 | 0.967±0.007 | 0.972±0.004 | 0.967±0.006 | 0.966±0.005 | **0.964**±0.003 |
| visualizing_soil | 0.005±0.001 | 0.006±0.001 | 0.009±0.001 | 0.011±0.001 | 0.026±0.001 | 0.010±0.001 | 0.006±0.000 | **0.005**±0.001 | 0.024±0.006 | 0.006±0.002 |
| wine_quality | 0.752±0.010 | 0.739±0.012 | 0.776±0.012 | 0.781±0.008 | 0.781±0.008 | 0.780±0.012 | 0.727±0.008 | **0.703**±0.012 | 0.709±0.009 | 0.709±0.013 |
| yprop_4_1 | 0.971±0.007 | 0.953±0.004 | 0.967±0.007 | 0.968±0.008 | 0.966±0.008 | 0.960±0.004 | 0.982±0.030 | 0.953±0.006 | 0.954±0.005 | **0.949**±0.006 |

# E   Broader Impact

We present NN models with an improved speed-accuracy tradeoff and hope that this can reduce the resource consumption of tabular models in applications and further benchmarks. While tabular ML has many potential applications, we feel that none must be particularly highlighted here.

