# OpenReview forum: "Better by default: Strong pre-tuned MLPs and boosted trees on tabular data"
_NeurIPS.cc/2024/Conference — NeurIPS 2024 poster_

### Official Review · Reviewer_vGM4 · 2024-06-20

**Soundness:** 2
**Presentation:** 3
**Contribution:** 3
**Rating:** 4
**Confidence:** 4

**Summary:**

The authors introduce RealMLP, an improved multilayer perceptron (MLP), alongside improved default parameters for GBDTs and RealMLP. The authors tune RealMLP on a meta-train benchmark with 71 classification and 47 regression datasets and compare them to hyperparameter-optimized versions on a disjoint meta-test benchmark with 48 classification and 42 regression datasets, as
well as the GBDT-friendly benchmark by Grinsztajn et al. (2022), claiming that RealMLP offers a better time-accuracy tradeoff than other neural nets and is competitive with GBDTs.

**Strengths:**

1. The inclusion of the standard recent benchmark of Grinsztajn et al. (2022) is welcome, and gives more credence to the authors' claims.
2. The description of the authors' improvements in Sec. 3 is unusually good, containing sufficient detail that, if needed, someone could most likely recreate the method from scratch.
3. The goal of improving the performance of MLPs on tabular data is both useful and interesting.

**Weaknesses:**

MAJOR

1. The support for a key claim ("RealMLP offers a better time-accuracy tradeoff than other neural nets and is competitive with GBDTs") is, at best, inconsistent. In Fig. 2, utilizing the authors' own benchmark suite, RealMLP is consistently less performant than the best GBDT. The claim that it outperforms other neural nets is not strongly supported by Fig. 2, since only a handful of the baseline NN methods from [1] are included. Particularly notable is the exclusion of [2], despite the fact that the authors reference it in their prior work. [2] requires less than a second to train and there exist methods in the literature to extend it to datasets of arbitrary size. As for Fig. 3, it seemingly contains no confidence intervals. However, the authors' method appears to perform comparably to FT-Transformer and Tab-R for classification tasks; on regression tasks, it is faster, but less performant, than FT-Transformer.
2. Sec 2.1 explains how the new benchmark was assembled, but not why it was necessary to do so. [1], [2] and Grinsztajn et al. have already introduced robust tabular benchmark suites into the literature. The authors also mention, but utilize only parts of, the AutoML benchmark and the OpenML-CTR23 regression benchmark.
3. While the concerns raised in Sec. 2.2 about some previously used metrics are reasonable, the choice of shifted geometric mean error, which adds noise to extremely low error scores, is also problematic. A better approach would have been to report multiple metrics.

MINOR

1. The 95% confidence intervals in the figures are very difficult to see, and should be made clearer.

[1] When do neural nets outperform boosted trees on tabular data?
[2] TabPFN: A Transformer That Solves Small Tabular Classification Problems in a Second

**Questions:**

QUESTIONS

* Why was it necessary to create a new benchmark suite? Why is this the fairest choice of benchmark for these methods?
* Why report only one non-standard metric?
* Why are so few deep tabular methods included in the core experiments?

SUMMARY

While the detailed description of the authors method is helpful and the ideas contained therein could be of interest to the NeurIPS audience, the experimental results are not yet robust enough to warrant publication at NeurIPS.

**Limitations:**

The authors have adequately addressed the limitations and potential negative societal impact of their work.

---

> ### Author Rebuttal · Authors · 2024-08-07
>
> We thank the reviewer for the constructive feedback.
>
> Major weaknesses:
>
> 1.
> - We meant to write “is competitive with GBDTs in terms of benchmark scores”, we will update this sentence. Of course, GBDTs are faster on most datasets with our settings. But note that lower is better in terms of benchmark scores in Figure 2 and RealMLP is better than GBDTs on all benchmarks in terms of scores in Figure 2.
> - The simple “TabPFN+subsampling” version from [1] is outperformed by TabR and FT-Transformer in the recent large benchmark in https://arxiv.org/pdf/2407.00956, both of which are included in our Figure 3 and partially in Figure 2. Regarding versions of TabPFN that scale to large datasets, many recent scaling works are concurrent works. There is TuneTables from February, but still it (1) will be substantially slower than 1 second when scaled to larger datasets, and (2) is limited to classification. TabPFN-based results will probably be outdated very soon anyways since TabPFNV2 is supposed to be out soon.
> - Figure 3 contains no confidence intervals because we used the codebase of Grinsztajn et al. to compute results and it does not provide confidence intervals with the same meaning as the confidence intervals in Figure 2 (in general, producing such confidence intervals for normalized benchmark scores as done in Figure 3 would be statistically questionable, as the normalization of the metric depends on the benchmark results itself.)
> - Your last claim is incorrect: RealMLP outperforms FT-Transformer in terms of speed and benchmark scores for regression on the Grinsztajn et al. benchmark (and performs slightly better for classification as well).
>
> 2.
> - Regarding the meta-train benchmark: The benchmark consists of datasets that were available to us when we started experimenting. While it is not established, we cannot change it post-hoc since we already used it to design our methods.
> - Regarding the meta-test benchmark: The benchmark consists of the AutoML benchmark and the CTR23 benchmarks, which are well-curated tabular benchmarks. The AutoML benchmark is well-established in AutoML, and [1, 2] also use it for their dataset collection. The contained datasets span a large range of sizes and dimensionalities, allowing us to test meta-generalization of tuned defaults. We removed some datasets according to well-defined criteria in Appendix C.3.2, mainly datasets that are already included in the meta-train benchmark or are too small.
> - The benchmark suite of [1] is interesting but was released concurrently to when we started experimenting with the meta-test benchmark, and is mainly covering classification. As far as we can see, [1] doesn’t provide clear criteria by which they have selected their datasets, and their TabZilla subset has selection criteria prefering datasets that are hard for GBDTs, which would be unfairly favorable to neural networks such as our RealMLP.
> - [2] contains many datasets but mostly small ones, which we do not design our methods for, and which result in noisier evaluation due to small test sets. They also use some datasets from the AutoML benchmark, which we already use.
> - We use the Grinsztajn et al. benchmark datasets separately in Figure 3, to have a common benchmark where many baselines can run. We chose not to use these datasets for the meta-test benchmark since there are less of them (less than half of what we have now after excluding the meta-train datasets), and they are more limited in terms of number of features etc.
>
> 3. We **do report alternative aggregation metrics** in Figure B.7 – B.15 and reference some of them from Section 5.2. We also report normalized accuracies on the Grinsztajn et al. (2022) benchmark (Figure 3), to conform with the original benchmark. We agree that all aggregation strategies have some upsides and downsides.
>
> Minor weaknesses:
> - We increased the linewidth of the error bars, see the rebuttal PDF.
>
>
> Questions:
> - See above.
> - See above, we do report multiple aggregation metrics.
> - Because the meta-test benchmark is so expensive to run, see the global response. We already have the Grinsztajn et al benchmark for this purpose. Another issue is the lack of easily usable interfaces for most other published deep learning methods, including details such as how to pass a separate validation set, how to change the early stopping metric, how to pass categorical data, how to select the correct GPU device, etc. In contrast, we make scikit-learn interfaces available that allow configuring all of this easily.

---

> > ### Comment · Reviewer_vGM4 · 2024-08-11
> >
> > I would like to thank the authors for their response; I will not change my score at this time.

---

> > > ### Author Response · Authors · 2024-08-13
> > >
> > > We thank the reviewer for acknowledging our response, but are surprised to see no change of score given that we addressed most concerns:
> > > - Weakness 3 was incorrect.
> > > - Weakness 2 is unjustified in our opinion since we do include a standard benchmark and additional benchmarks *on top of this* with well-defined dataset inclusion criteria. Again, we used all datasets from the AutoML and CTR23 benchmarks except those that were excluded for a good reason based on exclusion criteria defined in Appendix C.3.2.
> > > - RealMLP performed better in terms of benchmark scores than claimed by the reviewer in Weakness 1.
> > > - We did add more baselines in the rebuttal, and we already have most of the well-performing baselines from [1] in Figure 3 except TabPFN, and in addition to [1] we have TabR and MLP-PLR which we believe to be more relevant since we are improving standard NNs and not PFNs, which are currently hard to compare against (How do you fairly compare to a method that doesn’t use a validation set? What do you do about limitations in #features/#classes? Which of the recent methods do you use to scale it to large datasets?)

---

### Official Review · Reviewer_J8gT · 2024-07-09

**Soundness:** 4
**Presentation:** 4
**Contribution:** 4
**Rating:** 7
**Confidence:** 4

**Summary:**

In the paper "Better by default: Strong pre-tuned MLPs and boosted trees on tabular data", the authors make two major contributions: (i) they propose a multi-layer perceptron configurations that is tuned on a set of training datasets and (ii) investigate in a large scale empirical study how different default parameterizations (library vs tuned on a set of training datasets) compares to optimized hyperparameters on a set of test datasets. The study reveals interesting results regarding the performance of default parameterizations set in standard ML libraries, better default parameterizations exist (at least for the scope of the benchmark), and that the proposed MLP often performs competitive to gradient-boosted decision trees when it comes to a Pareto-optimal tradeoff between accuracy and training time.

**Strengths:**

- The paper is very well written and easy to follow. Sufficient details on the architecture of RealMLP and the implementation of tuning the default parameterization or optimizing the hyperparameters are given.
- Generally speaking, the authors are very eager to provide all the details to make their work reproducible which is of high importance for such a paper.
- In principle, I like the 2D-comparisons but a summary plot how the D vs TD vs HPO behaves for the different methods would be very much appreciated.

**Weaknesses:**

- Intuitively one would expect that the performance of TD lies in between D and HPO but I could also imagine that TD is oftentimes very close to HPO or might even exceed HPO's performance due to overtuning. But this comparison is not considered as an overview.
- It is not clear why random search has been used as HPO technique in the paper, especially, since the authors even mention more sophisticated techniques such as SMAC that are based on Bayesian optimization.
- The style of writing in Section 3 could be improved as there are various content-wise enumerations in the form of sentences starting with "We".

**Questions:**

- How would a CASH approach compare here if we would first select the learning algorithm according to the (tuned) default parameterization and then optimize its hyperparameters? As it is done in Mohr, F., Wever, M. Naive automated machine learning. Mach Learn 112, 1131–1170 (2023). https://doi.org/10.1007/s10994-022-06200-0
- Why was random search used for hyperparameter optimization?
- Is there any intuition why using the last of tied epochs as the final model as stated on line 161?
- Why are hyperparameter values rounded as stated on line 170?

**Limitations:**

Limitations have properly been described in the paper.

---

> ### Author Rebuttal · Authors · 2024-08-07
>
> We thank the reviewer for the constructive feedback.
>
> Weaknesses:
> - Thank you for the suggestion. We have something like this in Figures B.12-B.15, D.2, D.6, D.7, or do you have a specific suggestion for an overview plot?
> - Random search 1) is used in the Grinsztajn et al benchmark, 2) is more convenient to run since all steps can be run in parallel and resource constraints (RAM usage) can be adjusted per parameter configuration, 3) allows to change the optimization metric post-hoc as we do for AUROC in Figure B.4 since the sampled configurations do not depend on the optimization metric. Random search has also been used in the benchmark of McElfresh et al. (2023), for example. We have a comparison to TPE (in hyperopt) for GBDTs in Figure B.6 and the differences are rather small.
> - Thank you for the comment, we will try to improve this in the next version.
>
> Questions:
> - Thanks for the question, we also looked at this. It seems that the results are about halfway between Best-TD and Best-HPO (whose difference is quite small already). We were not sure if including this would complicate the message too much.
> - See above.
> - Not a lot, but a reason could be that when the MLP continues to improve slowly but the last few validation accuracies are identical, the last one is still more promising for the test set.
> - To make them look nicer and more memorable in paper/code. (We checked that it doesn’t significantly hurt the performance.)

---

> > ### Comment · Reviewer_J8gT · 2024-08-12
> > **Response to Rebuttal**
> >
> > Thank you very much for the on-point responses and the clarification.
> >
> > Regarding the use of random search: I am well aware of the benefits of random search in empirical studies, but it should be mentioned in the paper to justify the choice and explain the benefits to the reader.
> >
> > Regarding the overview plot, I was rather thinking of some spider chart where the axes are given by the learner. TD, D, and HPO are then data rows for that kind of chart so that one can see the area spanned by the respective types for each base learner or something in that direction. Does that make sense?

---

> > > ### Author Response · Authors · 2024-08-14
> > >
> > > Thank you for your answer and your suggestions!
> > >
> > > > Regarding the use of random search: I am well aware of the benefits of random search in empirical studies, but it should be mentioned in the paper to justify the choice and explain the benefits to the reader.
> > >
> > > We will discuss this choice in more detail in the revised manuscript.
> > >
> > > >Regarding the overview plot, I was rather thinking of some spider chart where the axes are given by the learner. TD, D, and HPO are then data rows for that kind of chart so that one can see the area spanned by the respective types for each base learner or something in that direction. Does that make sense?
> > >
> > > Thank you for this insightful suggestion, we will experiment with spider charts and add it to the revised version if it gives a clear picture of the situation. We are not sure the D versions of different algorithms are directly comparable, but for TD and HPO at least it should be interesting.

---

### Official Review · Reviewer_eZ6k · 2024-07-09

**Soundness:** 3
**Presentation:** 3
**Contribution:** 2
**Rating:** 4
**Confidence:** 5

**Summary:**

The work investigates/provides better default hyperparameter configurations for MLPs and gradient-boosted decision trees. Additionally, it proposes an augmented neural network with a series of enhancements. Experimental results are provided showing that the neural network is on par with gradient-boosted decision trees. Moreover, the authors argue that the neural network and the gradient-boosted decision trees with the new defaults are competitive with their HPO-tuned per-dataset counterparts and motivate practitioners to use the different methods with the new defaults and perform ensembling.

**Strengths:**

- The work is written well.
- The work uses a wide collection of benchmarks that are well-known in the community.
- The work includes results with default, tuned-per-benchmark, and tuned-per-dataset hyperparameter configurations for the main methods investigated.

**Weaknesses:**

- The novelty of the paper is limited in my opinion. The majority of the components that are proposed in Figure 1. C are not novel and are used in AutoML methods. Moreover, it is not clear what subset of the proposed components could actually benefit the other methods and not only the MLP.
- The baselines used are not consistent, for example, FT-Transformer and SAINT are included for the Grinsztajn et al. benchmark, but not for the other benchmarks.
- The protocol is not fully consistent, as for example, for the results of the meta-train and meta-test benchmark, HPO is not applied for the ResNet method or Tab-R.
- Figure 2 shows results for accuracy, where RealMLP-TD strongly outperforms gradient-boosted decision tree methods for the meta-test classification and regression benchmark. However, AUROC is a metric that considers class imbalance and provides a better view of the method performances, Figure B.3 which reports results for AUROC shows a different picture, where, gradient-boosted decision trees outperform the RealMLP-TD. As such, I would propose to place Figure B.3 in the main paper and Figure 2 in the Appendix.

**Questions:**

- Could the authors provide results for the missing baselines and for the baselines that do not feature HPO results?
- How does this work fair in comparison to using a portfolio of configurations using dataset meta-features and selecting the default configuration per dataset based on a similarity distance between the portfolio of configurations and the context dataset?

**Limitations:**

- I believe the authors have adequately addressed the limitations.

---

> ### Author Rebuttal · Authors · 2024-08-07
>
> We thank the reviewer for the constructive feedback.
>
> Weaknesses:
>
> 1. What is the source for your claim? Which AutoML tool would these be in? Are you referring to some of the color-coded improvements or only to the gray ones? There are some non-novel components in Figure 1 (c) since we wanted to start from a “vanilla MLP”, but as we color-coded in the figure, there are many new and unusual components. We also tried out the two NNs in AutoGluon and their default settings performed worse than MLP-D on the meta-train benchmark. Regarding your second point, please see the global response.
>
> 2. While we understand the desire to have every method on every benchmark, unfortunately the meta-test benchmark is much more expensive to run than the other two benchmarks since it contains larger and higher-dimensional datasets. For example, FT-Transformer runs into out-of-memory issues on some datasets, trying to allocate, e.g., 185 GB of RAM. We did not try SAINT since it’s even more expensive.
>
> 3. Continuing on the point above, TabR-HPO would potentially take 5-20 GPU-months (based on extrapolating the runtime of TabR-S-D) to run on meta-test on RTX 3090 GPUs. We are now running it on the Grinsztajn et al. benchmark to have it on at least one benchmark. We did not run ResNet-HPO before since in most papers and Fig. 3 it is quite similar to MLP-HPO while being slower (takes around two GPU-weeks on our benchmarks in Fig. 2). We are currently running it and provide preliminary results in the uploaded PDF, which are confirming our expectations.
>
> 4. We included the AUROC experiments after the main experiments were finished to include an imbalance-sensitive metric. However, since the whole meta-learning and model development process was performed with accuracy as the target metric, and the best-epoch selection was also performed with accuracy as the target metric, we think that the results in Figure 2 are more suited for the main paper as they are more appropriate to demonstrate the effect of meta-learning default parameters. We do refer to the AUROC experiments in the main paper, though, and will try to emphasize these experiments more in the next version.
>
> Questions:
> - See above.
> - This is an interesting question, but would deserve its own paper. This would also be challenging to do without too much overfitting, given the limited number of meta-train datasets. TabRepo, perhaps the most large-scale study on portfolio learning, still only considers static portfolios.

---

> ### Comment · Reviewer_eZ6k · 2024-08-13
> **Author Rebuttal Response**
>
> I would like to thank the authors for the reply. I have carefully read the other reviews and all the responses from the authors.
>
> - Regarding: **"What is the source for your claim? Which AutoML tool would these be in? Are you referring to some of the color-coded improvements or only to the gray ones? There are some non-novel components in Figure 1 (c) since we wanted to start from a “vanilla MLP”, but as we color-coded in the figure, there are many new and unusual components. We also tried out the two NNs in AutoGluon and their default settings performed worse than MLP-D on the meta-train benchmark. Regarding your second point, please see the global response."**
>
>    The gray components would be an example, but additionally, the highlighted label smoothing is not something surprising. As mentioned by another reviewer, the paper provides a set of results that give intuition on the different method performances. As such, it would be beneficial for techniques that can be applied to the MLP to be added to the other baselines. It might be that the components hurt performance, it might be that the components improve performance.
>
>    One simple test would be to reuse the defaults for the MLP for another deep baseline (for what component can be applied), while this might not be conclusive as the hyperparameters probably need to be tuned per method, it would provide some insights.
>
> - Regarding: **"While we understand the desire to have every method on every benchmark, unfortunately, the meta-test benchmark is much more expensive to run than the other two benchmarks since it contains larger and higher-dimensional datasets. For example, FT-Transformer runs into out-of-memory issues on some datasets, trying to allocate, e.g., 185 GB of RAM. We did not try SAINT since it’s even more expensive."**
>
>    In my perspective, the already considered list of methods is extensive. For example, running TabNet from my perspective is not interesting, the method is outperformed significantly from many methods. However, the already included baselines in the comparisons should be present in all results, since they leave "holes" in the results.
>
>    One might worry that some methods might achieve more competitive performance and that is the reason why the authors do not include them. I personally do not think the authors are hiding results, however, the concerns should be addressed, especially considering the work lies towards the benchmarking and experimental side.
>
>    If a method fails on certain datasets, the authors can point it out in the work.
>
> - Regarding: **"We included the AUROC experiments after the main experiments were finished to include an imbalance-sensitive metric. However, since the whole meta-learning and model development process was performed with accuracy as the target metric, and the best-epoch selection was also performed with accuracy as the target metric, we think that the results in Figure 2 are more suited for the main paper as they are more appropriate to demonstrate the effect of meta-learning default parameters. We do refer to the AUROC experiments in the main paper, though, and will try to emphasize these experiments more in the next version."**
>
>    While that is true, the main attention is invested in the main manuscript, as such, the results can be potentially misleading. AUROC is a better metric that considers class imbalance. Personally, I find the work interesting, whether it beats CatBoost or not.
>
> - As a last question, was TabR run with embeddings or without? My understanding is the latter. If that is the case, it is questionable, since the main version of TabR contains embeddings. Additionally, the MLP is run with embeddings.

---

> > ### Author Response · Authors · 2024-08-13
> >
> > We thank the reviewer for the detailed response.
> >
> > - The label smoothing is highlighted as “unusual”, and we would argue that it is indeed unusual in the tabular context. We tried TabR-S-D with our preprocessing + beta_2=0.95 + scaling layer + parametric Mish + label smoothing on meta-train-class and it was slightly better than with just the preprocessing, which is in turn better than with the original preprocessing (as we showed in the Appendix). At this point the message of this investigation is unclear and would require more ablations, tuning, etc., opening a big rabbit hole.
> > - We understand that some readers might be concerned by the non-matching baselines, and we will explicitly mention the reasons for this in the next version of the paper. For baselines that fail on some datasets, we have the additional problem that not all aggregation metrics support missing results.
> > - In case of acceptance, we will try to use the extra page to add Figure B.3 or B.4, which contain the AUC results, to the main paper.
> > - TabR-S-D was run without numerical embeddings, since this is the version that was proposed in the paper for default hyperparameters. While this might deteriorate performance, it also improves the training time, which we also report. The new TabR-HPO results accidentally didn’t use numerical embeddings, we will rerun them.

---

> ### Comment · Reviewer_eZ6k · 2024-08-13
> **Response to Authors**
>
> I thank the authors for the response.
>
> - Regarding: **"The label smoothing is highlighted as “unusual”, and we would argue that it is indeed unusual in the tabular context. We tried TabR-S-D with our preprocessing + beta_2=0.95 + scaling layer + parametric Mish + label smoothing on meta-train-class and it was slightly better than with just the preprocessing, which is in turn better than with the original preprocessing (as we showed in the Appendix). At this point the message of this investigation is unclear and would require more ablations, tuning, etc., opening a big rabbit hole."**
>
>    I am not sure about the practitioners, but personally, I have used label smoothing in my experiments. Nevertheless, this is a paper for the community and I would need to review the other papers in the literature to confirm this very minor point. A recipe from the authors could be beneficial for the community. I agree that it is more complicated in the scenario the authors are pointing out for other baselines. One could maybe frame it as a general recipe or set of defaults for deep learning methods but it is not trivial and I am not sure if that is the message the authors want to convey.
>
> - Regarding: **"TabR-S-D was run without numerical embeddings, since this is the version that was proposed in the paper for default hyperparameters. While this might deteriorate performance, it also improves the training time, which we also report. The new TabR-HPO results accidentally didn’t use numerical embeddings, we will rerun them."**
>
>    I understand, in that case, I would agree with the authors on using TabR-S-D for the default case and the one with numerical embeddings for the HPO case.
>
> In my perspective, the work is in a good position, it needs a bit more work regarding the inconsistent baseline usage, but overall it has extensive results. Based on this, I will increase my score from 3 to 4.

---

> > ### Author Response · Authors · 2024-08-14
> >
> > We thank the reviewer for the response and the score increase.
> >
> > We now ran TabR-S-D with the above modifications (except label smoothing) on our meta-train regression benchmark as well (took ~5 GPU-hours) and see the exact same picture as for classification. We are wondering whether we should try a version of TabR-HPO in Figure 3 that includes some of our tricks into the search space. If this helps, then of course it is unclear which tricks help and what this means for defaults.
> >
> > Regarding the inconsistent baseline usage, most other papers evaluate their methods on a single benchmark. Therefore, we would argue that the fact that we have an additional very diverse meta-test benchmark, even though we cannot run all methods due to large and high-dimensional datasets, should not be seen as a weakness but a strength of our paper.

---

### Official Review · Reviewer_RnHK · 2024-07-11

**Soundness:** 3
**Presentation:** 3
**Contribution:** 2
**Rating:** 4
**Confidence:** 5

**Summary:**

The paper introduces RealMLP, an enhanced Multilayer Perceptron (MLP) designed for classification and regression tasks on tabular data. It also proposes optimized default parameters for both RealMLP and Gradient Boosted Decision Trees (GBDTs). Through benchmarking on diverse datasets, the authors demonstrate that RealMLP achieves a superior balance between time efficiency and accuracy compared to other neural networks, while remaining competitive with GBDTs. The integration of RealMLP and GBDTs using optimized defaults shows promising performance on medium-sized tabular datasets, alleviating the need for extensive hyperparameter tuning.

**Strengths:**

1. The study includes comprehensive benchmarking on a meta-train dataset comprising 71 classification and 47 regression datasets, and a meta-test dataset comprising 48 classification and 42 regression datasets.

2. Detailed experimental settings and hyperparameters are explicitly specified in the paper and appendices, ensuring reproducibility.

3. The experiments establish statistical significance, presenting error bars and critical-difference diagrams.

4. RealMLP demonstrates superior efficiency by offering a better time-accuracy tradeoff compared to other neural networks.

5. The optimized combination of RealMLP and GBDTs with improved default parameters achieves outstanding performance without the need for hyperparameter tuning.

**Weaknesses:**

1. The finding presented in this work is not entirely novel. Similar conclusions have been reached by Tree-hybrid MLPs `[1]`, although the methods are not quite the same. Moreover, ensemble different methods, including combinations like MLP + GBDTs, have long been recognized as effective approaches in Kaggle competitions and do not necessitate rediscovery.

2. While mentioning good performance by default, it's worth noting that Excelformer achieves one of the best performances under default settings. However, a direct comparison was not included, which would enhance the clarity of the findings. Please include the comparison in the rebuttal.

3. Despite emphasizing the applicability to medium-sized datasets, practical applications often involve diverse tabular data with varying feature lengths and training data sizes. The small application scope could potentially limit the impact of this study. Besides, most of tabular data sizes are often small.

4. The compared methods are evasive. Since the authors want to claim the speed-performance balance, the effective approaches Net-DNF `[2]`, TabNet `[3]`, and TabCaps `[4]` are not compared. The compared models FT-Transformer and SAINT, especially SAINT, are heavy.

**REF**

`[1]` Team up GBDTs and DNNs: Advancing Efficient and Effective Tabular Prediction with Tree-hybrid MLPs

`[2]` Net-DNF: Effective Deep Modeling of Tabular Data

`[3]` Tabnet: Attentive interpretable tabular learning

`[4]` Tabcaps: A capsule neural network for tabular data classification with bow routing

**Questions:**

The effective default performance achieved by MLP + GBDTs may not be suitable for all scenarios, such as those requiring tabular pre-training or zero-shot scenarios, where pure tabular models might be more appropriate. How does the proposed method address applications in tabular pre-training scenarios?

---

> ### Author Rebuttal · Authors · 2024-08-07
>
> We thank the reviewer for the constructive feedback.
>
> Weaknesses:
>
> 1. **[1] was uploaded on 13th of July 2024**, so this paper cannot be used to question our novelty. (Besides, it also uses a mix of architectures, while we consider algorithm selection / ensembling.) Regarding ensembles, we do not claim to rediscover ensembles. Rather, we try to see how meta-learning better defaults affects the trade-off between ensembling / algorithm selection and HPO.
> While the ExcelFormer paper claimed to have very good defaults, ExcelFormer performed poorly in the experiments at https://github.com/pyg-team/pytorch-frame. So either the original results were problematic or the method is difficult to use correctly, both of which don’t make it attractive for us to compare to ExcelFormer. Moreover, as a transformer with inter-feature attention, ExcelFormer would likely result in out-of-memory errors on some high-dimensional datasets of the meta-test benchmark (as FT-Transformer does). We compared to TabR-S-D since their paper also claimed to have better defaults than GBDTs. We can, however, acknowledge the missing ExcelFormer comparison as a limitation.
>
> 2. We consider dataset sizes of 1K to 500K samples and up to 10K features, which is quite a large range, larger than the Grinsztajn et al. benchmark, for example. Smaller datasets are also interesting but hard to evaluate due to noisy test scores, and might be much more affected by details in early stopping etc. In addition, they are easier to handle for TabPFN or LLM-based methods, necessitating different baselines and making MLPs less attractive.
>
> 3. We agree that additional efficient baselines can help to substantiate our claim, but we think that the mentioned methods are not particularly promising based on recent results from the literature (see below). Instead, we evaluate MLP-PLR [5], an MLP with numerical embeddings that was presented as better than FT-Transformer from the group that created FT-Transformer, and performed well in recent benchmarks [6, 7]. MLP-PLR is faster than RealMLP but does not match RealMLP’s benchmark scores. Here are some reasons why we think that the mentioned NNs are not very promising:
> - TabNet performs poorly both in recent benchmarks as well as in our preliminary experiments. In McElfresh et al. (2023), TabNet performs much worse than MLP or FT-T while also being slower than both of them. In the FT-Transformer and SAINT papers, TabNet is also worse than MLP. The same holds for https://arxiv.org/abs/2407.09790 and https://arxiv.org/abs/2407.00956
> - Net-DNF performed slightly worse than TabNet in Shwartz-Ziv et al. (2022). It did perform a bit better than MLP on 4 datasets in Borisov et al. (2022). It was also outperformed by TabCaps in the TabCaps paper.
> - TabCaps performs worse than MLP in a recent large benchmark (https://arxiv.org/abs/2407.00956), see Tables 19 and 20 therein. The TabCaps paper also doesn’t contain a lot of datasets, so their own claims are probably not too reliable.
>
> Questions:
> - We do not aim to address tabular pre-training or zero-shot scenarios. In cases where one can pre-train on a dataset with the same columns, RealMLP might be a good option.
>
> [5] https://proceedings.neurips.cc/paper_files/paper/2022/hash/9e9f0ffc3d836836ca96cbf8fe14b105-Abstract-Conference.html
>
> [6] https://arxiv.org/abs/2407.02112
>
> [7] https://arxiv.org/abs/2406.19380

---

> ### Comment · Reviewer_RnHK · 2024-08-11
>
> Thank you for your reply.
>
> I’m aware that [1] was uploaded on July 13th, 2024, and that it is completely different from your paper. As I mentioned in my initial comment, it was only an example to illustrate that this paper revisits an old conclusion. I’m not questioning the novelty of your work based on this example.
>
> Regarding the performance of Excelformer, I believe you should use the official code. I’ve tried using the code from PYG, but it seems to have many bugs, and the hyperparameters were clearly misused.
>
> Additionally, in my experience, Net-DNF often outperforms both TabNet and NODE.
>
> But, the performance of previous works and the similarity to [1] are not the main points! We all know that a paper should aim to present new scientific findings.
>
> You said: "we try to see how meta-learning better defaults affects the trade-off between ensembling / algorithm selection and HPO". Essentially, integrating different methods (such as MLP + GBDT) has long been recognized as an effective approach in Kaggle tabular prediction competitions. Besides, achieving good results in these competitions typically does not require extremely rigorous HPO (that may leads to over-fitting), which are well understood by tabular data prediction experts.

---

> > ### Author Response · Authors · 2024-08-12
> >
> > We thank the reviewer for the insights on various NN models. Does the reviewer have insights on the performance of Net-DNF vs. more modern / well-studied methods like MLP/ResNet from RTDL that are used in many more recent benchmarks?
> >
> > Regarding the main point “a paper should aim to present new scientific findings”:
> > - The general benefits of combining GBDTs and NNs are of course widely known both on Kaggle as well as established scientifically through AutoML tools as well as Shwartz-Ziv et al. (2022). However, we specifically study whether (1) ensembling is preferable to HPO under time constraints and whether (2) meta-learning defaults shifts this balance. While some Kagglers may have intuitions about this, we doubt that there would be a consensus, for example, by looking at the posts of Kaggle Grandmaster Bojan Tunguz (“XGBoost is all you need”). We invite the reviewer to provide concrete evidence to the contrary if they disagree.
> > - Even though practitioners and Kaggle experts have some important intuitive knowledge not found in the literature, we believe that verifying this knowledge on large scale benchmarks and making it accessible to the scientific community is important.
> > - Even if the reviewer does not find our results on this aspect particularly interesting, there are other contributions of our work: we also provide an easily accessible improved MLP which performs very well both with default HP and after HPO, find better hyperparameters for standard GBDT libraries, and provide numerous insights in our ablations.

---

### Official Review · Reviewer_5ngr · 2024-07-12

**Soundness:** 3
**Presentation:** 3
**Contribution:** 2
**Rating:** 4
**Confidence:** 4

**Summary:**

The paper proposes an enhanced version of the tabular MLP model -- RealMLP. By using multiple tricks over simple MLP, the proposed method becomes much more competitive with GBDTs than a simple MLP. Moreover, the authors provide strong "tuned default" configurations for GBDTs and RealMLP. These configs considerably reduce training time in exchange for arguably small test error increase compared to hyperparameter optimization procedures.

**Strengths:**

* The paper addresses two important problems in tabular DL:  1) lack of an efficient strong DL baseline  2) high cost of extensive hyperparameter tuning for a good performance
* Extensive evaluation of the proposed method. Appendix A and B are also provide important ablations on RealMLP.
* Great experimental setup in terms of meta-train and meta-test splits that allow to find tuned-defaults.
* RealMLP is considerably better than MLP.
* Tuned-defaults for GBDTs.
* Everything necessary to reproduce the results is provided.

**Weaknesses:**

1. One of the main contributions of the paper is the different DL techniques from Figure 1 that have been added to MLP. Although there are enough useful experiments and ablations on these techniques for RealMLP (Figure 1 and Appendix B), these tricks can be applied to any DL model. From Figure 3 we observe that TabR-S-D and FT-Transformer-D are much better than MLP-D and slightly worse than RealMLP-TD. So, I think it is essential to ablate the mentioned techniques to improve these models and compare the results with RealMLP. I expect that the techniques improve stronger models much less compared to MLP, and it would be a beneficial result. It will show that RealMLP-TD is a good baseline while TabR-S-D-with-tricks is only slightly better than TabR-S-D. Otherwise, if improved TabR or FT-T considerably outperform RealMLP, the results will be even closer to GBDTs' which is also a positive outcome. I think that the results in Table B.5, which show an advantage of RC+SF preprocessing for TabR-S-D , also support my idea.
2. Table B.1. from Appendix: changing Adam's $\beta_2$ from 0.95 to 0.999 significantly increases the error rate while changing Adam's $\beta_2$ from 0.999 to 0.95 on Figure 1 has almost no effect. I would make a conclusion that the resulting RealMLP-TD architecture with all tricks is very sensitive to hyperparameters.
3. Also Table B.1: Strangely, PLR embedding is much worse than PL one. The only difference is ReLU if I am not mistaken.
4. No non-parametric DL baselines in Figure 2. Only TabR-S-D is presented. Also, it would be interesting to compare tuned MLP-PLR [1]. Authors from [1] use extensive HPO and do not fix embedding size, so a comparison with RealMLP-TD would be interesting to see if a properly tuned MLP-PLR is better/worse than RealMLP with tricks and fixed embedding size.

**Questions:**

1. Please, see weaknesses.
2. Probably I missed it in the text but how many HPO iterations have you used to obtain -TD configs?
3. Have you measured inference efficiency? I think MLP is usually a bit faster than GBDTs but I wonder if parametric activations or proposed numerical embeddings make MLP slower.

**Limitations:**

Limitations are addressed.

---

> ### Author Rebuttal · Authors · 2024-08-07
>
> We thank the reviewer for the constructive feedback.
>
> Weaknesses:
>
> 1. While we agree that such experiments would be interesting, they would be very costly and are not central to the research question we want to answer in the paper (how good can we make MLPs and GBDTs with default parameters?). We would argue that the existing results should already be interesting to readers and that our paper's ability to pose many interesting questions and potential follow-up studies should be viewed as a strength rather than a weakness.
>
> 2. We can mention this in the main paper. We already mentioned in Appendix B that this is likely due to the very high weight decay value (we did an earlier ablation without weight decay for regression and the differences were much smaller). So we would argue that the sensitivity to some hyperparameters is not a big issue as it can be fixed with lower weight decay or by just using our provided defaults.
>
> 3. We double-checked our implementation and we also checked MLP-PLR and MLP-PL with the original libraries and observed the same effect for defaults on meta-train (we didn’t run MLP-PL-HPO). So it appears that PLR is better than PL on the 11 datasets of its own paper with HPO, but is worse with fixed hyperparameters on our 118 meta-train datasets. The PLR library documentation also mentions that PL can allow for a lower embedding dimension (perhaps because there is no information loss due to the ReLU). So this seems to be a weakness of PLR embeddings and not of our paper.
>
> 4. TabR-S-D is a non-parametric baseline. Unfortunately, TabR-HPO would be way too expensive to run on the meta-test benchmark (easily 5-20 GPU-months on RTX 3090 GPUs extrapolated from the runtime of TabR-S-D). We did also include TabR-HPO on the Grinsztajn et al. benchmark in the rebuttal. Since other non-parametric methods like SAINT are even more expensive, we cannot afford to run them in Figure 2 (but SAINT is in Figure 3). We did include MLP-PLR now. The defaults are on the level of MLP-D (the paper does not propose defaults, but the library does). MLP-PLR-HPO comes close to RealMLP-HPO on meta-test-reg but underperforms RealMLP-TD on the other three meta-benchmarks. Given the results, optimizing the embedding dimension could be interesting for RealMLP-HPO as well, although the larger embedding dimensions appear to considerably increase CPU runtimes. (MLP-PLR-HPO has about equal runtime to RealMLP-HPO, despite using early stopping.)
>
> Questions:
>
> 1. See weaknesses.
>
> 2. We did not mention this in the paper since it is a one-time cost, but: Around 100-300 steps for GBDTs (could have probably matched this with 20-30 manual tuning steps). For RealMLP, we did not use automated tuning, since we repeatedly implemented new components and tried them, so it is hard to say but several hundred configurations were tried.
>
> 3. We have measured inference efficiency but did not want to report it since it could be quite sensitive to implementation details (inference batch size, data conversions, etc.). For example, we heard from AutoGluon developers that they could greatly improve AutoGluon’s inference times just by running its preprocessing in numpy instead of pandas. Nonetheless, here are some of our measurements (inference time per 1K samples):
> - CatBoost-D_CPU: 0.00273057 s
> - XGB-D_CPU: 0.00297026 s
> - RF-SKL-D_CPU: 0.016169 s
> - RealMLP-TD_CPU: 0.00501239 s
> - RealMLP-TD-S_CPU: 0.00377973 s
> - MLP-D_CPU: 0.0187927 s
> - TabR-S-D_CPU: 0.125827 s
>
> As you can see, RealMLP-TD is a bit slower than GBDTs but still fast, while for MLP-D there seems to be a suboptimal implementation (we don’t know why it would be so slow otherwise) and TabR is very slow.

---

> ### Comment · Reviewer_5ngr · 2024-08-13
> **Response to Authors**
>
> I carefully read the rebuttal and would like to thank the authors! Also, I apologise for the late reply.
>
> 1. In my opinion, the idea of making MLP as strong as possible is really valuable for the field but the lack of "fair" comparison is a big weakness of the paper. ("fair comparison" = other models with the proposed techniques). Though I understand the high cost of the tuning, I think even results for TabR-S-D+some_tricks would be beneficial for the paper.
> 2. Thanks for the answer. It should be added to the main text since stability to hyperparameters is important.
> 3. I think this "issue" could also be related to optimization stability (or sensitivity to hyperparameters).
> 4. Sorry, I made a typo, TabR is the non-parametric baseline. I meant that the paper lacks a strong parametric baseline since FT-T is quite expensive, and ResNet is just bad. MLP-PLR is a good baseline thanks for the provided results.
> 5. Inference time seems reasonable, thanks for the results.
>
> Overall, in my opinion, the main weakness (the first one) remains and, thus, I keep my score unchanged.

---

> > ### Author Response · Authors · 2024-08-13
> >
> > We thank the reviewer for taking the time to consider our response.
> >
> > Regarding the fairness of our comparison between our MLP and other neural networks, we do not strive to be fair between different architectures, and state this in the paragraph on limitations. This paper is not about comparing architectures, but rather offering a good model, and measuring the importance of meta-learning good default hyperparameters.
> >
> > We tried some of our tricks on top of TabR-S-D on our meta-train-class benchmark. The result is slightly better than the TabR-S-D version with only our preprocessing, which is already better than TabR-S-D. We are not sure how to turn this into a fair comparison with a reasonable amount of effort.

---

### Author Rebuttal · Authors · 2024-08-07

Dear Reviewers,
thank you for the constructive feedback. We identified two main points raised by multiple reviewers:
- 4/5 reviewers asked for more baselines. In total, ten baselines were mentioned (MLP-PLR, ExcelFormer, Net-DNF, TabNet, TabCaps, versions of TabPFN, TabR-HPO, ResNet-HPO, SAINT, FT-Transformer). In the **attached PDF**, we **add MLP-PLR** as a related baseline with good speed/accuracy trade-off, as well as **TabR-HPO** on the Grinsztajn et al. benchmark and **ResNet-HPO** on the meta-benchmarks. However, we hope that the reviewers understand that we cannot run all possible baselines for reasons of computational cost, implementation/verification effort and cluttering of the paper/figures. In particular, we should have emphasized more that the meta-test benchmark is very expensive to run (ca 4 GPU-weeks for RealMLP-HPO, ca 2 GPU-weeks for ResNet-HPO, probably >5 GPU-months for TabR-HPO given that the runtime of TabR-S-D is >4 GPU-days). In addition, some methods (Transformer-based/TabPFN-based) fail due to the high numbers of features (up to 10K). The diversity in size and dimensions of the meta-test datasets allow for a broad study of meta-generalization. We have already included the Grinsztajn et al. benchmark in Figure 3 in order to have a common and less expensive benchmark with more deep learning baselines. Details on our decisions can be found in our individual replies.
- 2/5 reviewers asked whether our tricks would help other architectures as well. While we agree that this is an interesting question, we would argue that it is not relevant to our central question, namely how well MLPs and GBDTs can perform with tuned default settings. Moreover, due to the much higher training cost of models like TabR or FT-Transformer, such experiments would require a lot of extra effort. We therefore do not want to explore this question further in this paper beyond the preprocessing experiments that we already have in Table B.5.

We kindly ask the reviewers to consider raising their score, given that our benchmarking efforts are already more extensive than what can be found in most other method-papers.

---

> ### Author Response · Authors · 2024-08-08
>
> We just noticed that our pdf uses an older version of the meta-train/meta-test plots where ResNet-HPO is not included. The results for ResNet-HPO are similar to the Grinsztajn et al. benchmark: a bit better than MLP-HPO for classification, a bit worse for regression, and about twice as slow.

---

### Decision · Program_Chairs · 2024-09-25

**Decision:**

Accept (poster)

**Comment:**

While the work is not entirely novel, there are extensive benchmarks, and the methods provided are easily applicable for practitioners.
The main concern of the reviewers seems to be the breadth of the comparison. However, the reviewers do provide a wide array of benchmarks and baseline methods.